# Elucidating Flow Matching ODE Dynamics via Data Geometry and Denoisers

Zhengchao Wan [* 1]   Qingsong Wang [* 2]   Gal Mishne [2]   Yusu Wang [2]

## Abstract

Flow matching (FM) models extend ODE sampler based diffusion models into a general framework, significantly reducing sampling steps through learned vector fields. However, the theoretical understanding of FM models, particularly how their sample trajectories interact with underlying data geometry, remains underexplored. A rigorous theoretical analysis of FM ODE is essential for sample quality, stability, and broader applicability. In this paper, we advance the theory of FM models through a comprehensive analysis of sample trajectories. Central to our theory is the discovery that the denoiser, a key component of FM models, guides ODE dynamics through attracting and absorbing behaviors that adapt to the data geometry. We identify and analyze the three stages of ODE evolution: in the initial and intermediate stages, trajectories move toward the mean and local clusters of the data. At the terminal stage, we rigorously establish the convergence of FM ODE under weak assumptions, addressing scenarios where the data lie on a low-dimensional submanifold—cases that previous results could not handle. Our terminal stage analysis offers insights into the memorization phenomenon and establishes equivariance properties of FM ODEs. These findings bridge critical gaps in understanding flow matching models, with practical implications for optimizing sampling strategies and architectures guided by the intrinsic geometry of data.

## 1. Introduction

Diffusion-based generative models have become the de facto method for the task of image generation (Sohl-Dickstein et al., 2015; Ho et al., 2020; Song & Ermon, 2019). Compared to previous generative models (e.g., GANs (Goodfellow et al., 2014)), diffusion models are easier to train but suffer from long sampling times due to the sequential nature of the sampling process. ODE-based samplers were introduced to address this limitation, where the sampling process is done by integrating an ODE. With its efficiency, ODE-based samplers have become the dominant approach in diffusion models (Song et al., 2021; Lu et al., 2022; Karras et al., 2022). Recently, the ODE-based viewpoint of diffusion models has been extended to a general framework known as **flow matching** (FM) (Lipman et al., 2022; Albergo & Vanden-Eijnden, 2023; Liu et al., 2023), which uses an ODE to interpolate between a prior and a target data distribution. FM models learn a vector field $u_t$, similar to the score function in diffusion models. During sampling, a data sample $x_1$ is generated by integrating the ODE starting from some $x_0 \in \mathbb{R}^d$ sampled from a prior distribution:

$$\frac{dx_t}{dt} = u_t(x_t), \ t \in [0, 1).$$

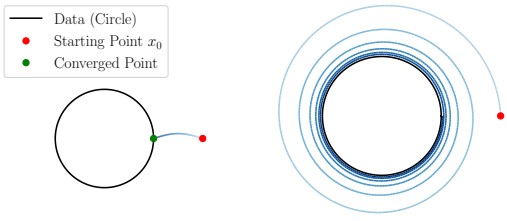

| — | Data (Circle) |
| :-: | :-- |
| • | Starting Point $x_0$ |
| • | Converged Point |

(a) Convergent trajectory    (b) Winding trajectory

*Figure 1.* **Comparing two ODE trajectory behaviors.**

Various versions of the FM model have gained popularity, such as the rectified flow model (Liu et al., 2023), which is utilized in commercial image generation software (Esser et al., 2024). Furthermore, the succinct and deterministic formulation of the FM model also makes theoretical analysis potentially easier. Despite the empirical success, critical theoretical questions remain insufficiently addressed: How does the data geometry (e.g., clusters, manifold structure) influence and guide individual sampling trajectories? Are these trajectories guaranteed to converge toward the data distribution as $t \to 1$, especially when the data lies on a low-dimensional subspace or manifold? This convergence is

*Equal contribution [1]Department of Mathematics, University of Missouri, Columbia, Missouri, USA [2]Halıcıoğlu Data Science Institute, University of California San Diego, La Jolla, California, USA. Correspondence to: Gal Mishne <gmishne@ucsd.edu>, Yusu Wang <yusuwang@ucsd.edu>.

*Proceedings of the 42nd International Conference on Machine Learning*, Vancouver, Canada. PMLR 267, 2025. Copyright 2025 by the author(s).

critical for the generative model's performance, as suggested by Loaiza-Ganem et al. (2024).

These questions regarding *per-sample* trajectories have both theoretical and practical significance for the sampling process because (1) robust sampling requires trajectory convergence in terminal time (Figure 1a), avoiding undesirable behaviors like winding around the data manifold (Figure 1b). Such convergence provides the theoretical foundation for distilling the trajectory into a one-step generative model like the consistency model (Song et al., 2023); (2) understanding the relationship between data geometry and ODE trajectories can motivate geometry-based steering of the sampling process or modification of the latent space for improved generation quality.

**Our approach.** We conduct a thorough investigation of per-sample FM ODE trajectories by focusing our analysis on the **denoiser**—the conditional mean of the data given noise (Karras et al., 2022). The denoiser emerges as the only data-dependent component of the flow vector field $u_t$, fundamentally determining FM ODE dynamics. Interestingly, by examining how the denoiser interacts with the data geometry, we demonstrate that the FM ODE exhibits two key properties: (1) **Attracting**—trajectories are drawn toward a specific set, and (2) **Absorbing**—once within a certain set, trajectories remain confined near it. With these properties, we quantitatively elucidate FM ODE trajectories across three stages (Figure 2): *initial*, *intermediate*, and *terminal*. The initial stage is characterized by trajectories moving toward the mean of the data distribution, while the intermediate stage is shaped by coarse-scale data geometry, with trajectories attracted to and absorbed into local clusters. The terminal stage is marked by the trajectory converging to the data support, where the attracting and absorbing dynamics ensure the convergence (see Section 5.1 for more details). While there are prior works on sampling evolution of diffusion models (Biroli et al., 2024; Li & Chen, 2024), they have mainly focused on *distribution-level* analysis of stochastic samplers using simplified settings like Gaussian mixtures. In contrast, our work reveals how data geometry—both coarse-scale clustering and fine-scale structure (discrete vs. manifold)—manifests in and guides individual ODE trajectories.

**Contributions.** In Section 3, we introduce the fundamentals of the denoiser and present our meta attracting and absorbing theorems (Theorems 3.1 and 3.2), which demonstrate how the properties of the denoiser can be leveraged to analyze FM ODE trajectories. These theorems provide a unifying framework to qualitatively study the behavior of trajectories during the initial and intermediate stages (Section 4) as well as the terminal stage (Section 5). Specifically, in Section 4, we first establish the well-posedness of the FM ODE trajectory on $[0, 1)$ for general data distributions

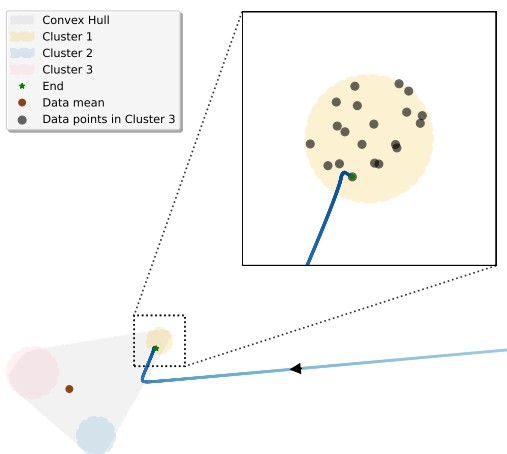

*Figure 2.* **Three stages of an FM ODE trajectory with synthetic data.** The curve with blue progression shows an FM ODE trajectory, with an arrow indicating direction. The shaded region indicates the convex hull of data. Three stages are visible: initially, the trajectory aligns with the data mean (brown point); next, it is attracted to a local cluster (yellow cluster); finally, it converges to a data point (green star). See Appendix J.1 for more details.

(Theorem 4.1) and establish rigorously how an FM ODE trajectory will initially move toward the data mean (Proposition 4.2) and later toward local clusters (Proposition 4.4). In Section 5, we establish the convergence for FM ODE as $t \to 1$ under mild assumptions (Theorem 5.3). To the best of our knowledge, this is the first result that accommodates data distributions supported on submanifolds. This convergence result allows us to study the properties of flow maps, leading to our establishment of equivariance of flow maps with respect to geometric transformations (Proposition 5.7). We also delve into the case of discrete measures, showing that terminal time training plays a critical role in addressing memorization phenomena (Propositions 5.9 and 5.10). See Figure 4 in Appendix A for a roadmap of our main theoretical results.

Due to space constraints, all proofs are deferred to the appendix. A **significant number of additional results** and observations, which could be of independent interest, are also presented in the appendix. For instance, we identify that the FM ODE vector field exhibits singularities and blows up when the data distribution lacks full support (cf. Appendix C.2), and we derive precise rates of convergence of posterior distributions/denoisers as $t \to 1$ depending on the data geometry, as detailed in Appendix D.

## 2. Background and Related Work

**Notations.** For any subset $\Omega \subset \mathbb{R}^d$, we let $d_\Omega(x) := \inf_{y \in \Omega} \|x - y\|$ denote the distance to $\Omega$. Let $B_r(\Omega) := \{x \in \mathbb{R}^d | d_\Omega(x) < r\}$. Let $\overline{\Omega}$ and $\partial\Omega$ denote the closure

and boundary of $\Omega$, respectively. The *medial axis* of $\Omega$ is denoted

$$\Sigma_\Omega := \left\{ x \in \mathbb{R}^d : \#\{\arg\min_{y \in \Omega} \|x - y\| > 1\} \right\},$$

where $\#A$ denotes the cardinality of a set $A$. For any $x \notin \Sigma_\Omega$, its *projection* onto $\Omega$

$$\mathrm{proj}_\Omega(x) := \arg\min_{y \in \Omega} \|x - y\|$$

is unique and hence well defined when $\Omega$ is closed. The *reach* of $\Omega$ is defined as $\tau_\Omega := \inf_{x \in \Omega} d_{\Sigma_\Omega}(x)$ which in a sense quantifies the smoothness of the set $\Omega$—a larger reach rules out tight bottlenecks and sharp bends (see, e.g., (Federer, 1959) for more details).

We use $\delta_x$ to denote the Dirac delta measure at $x$. We let $\mathcal{N}(\mu, \Sigma)$ denote the Gaussian distribution with mean $\mu$ and covariance $\Sigma$. Let $s \in [1, \infty]$, and we use $d_{\mathrm{W},s}(\nu_1, \nu_2)$ to denote $s$-Wasserstein distance for two probability measures $\nu_1$ and $\nu_2$.

See Appendix A for a table of symbols used in this paper.

### 2.1. Background on Flow Matching

Flow matching (FM) models (Lipman et al., 2022; Albergo & Vanden-Eijnden, 2023; Liu et al., 2023) are a class of generative models whose *training process* consists of learning a vector field $u_t$ that generates a probability path $(p_t)_{t \in [0,1]}$ interpolating a prior $p_0 = p_{\mathrm{prior}}$ and a target data distribution $p_1 = p$ and whose *sampling process* consists of integrating an ODE from an initial point $\boldsymbol{Z} \sim p_{\mathrm{prior}}$ to obtain a terminal point $\boldsymbol{X} \sim p$. More precisely, the interpolating path $(p_t)_{t \in [0,1]}$ in FM model is constructed as follows:

$$p_t(dx_t) := \int p_t(dx_t | \boldsymbol{X} = x) p(dx), \quad (1)$$

where the conditional distribution $p_t(\cdot | \boldsymbol{X} = x)$ satisfies that $p_0(\cdot | \boldsymbol{X} = x) = p_{\mathrm{prior}}$ and $p_1(\cdot | \boldsymbol{X} = x) = \delta_x$. *We assume that the prior $p_{prior}$ is the standard Gaussian $\mathcal{N}(0, I)$ throughout this paper.* Then, $p_t(\cdot | \boldsymbol{X} = x)$ are specified as

$$p_t(\cdot | \boldsymbol{X} = x) := \mathcal{N}(\alpha_t x, \beta_t^2 I),$$

where $\alpha_t$ and $\beta_t$ are *scheduling functions* satisfying $\alpha_0 = \beta_1 = 0$ and $\alpha_1 = \beta_0 = 1$ and are often monotonic. Common choices include linear scheduling $\alpha_t = t$ and $\beta_t = 1-t$ used in the rectified flow model (Liu et al., 2023; Esser et al., 2024) and those arising from noise scheduling in diffusion models. In this paper, we assume that $\alpha_t, \beta_t$ are smooth functions of $t$ on the closed interval $[0, 1]$. It is worth noting that $p_t$ is the law of the random variable $\boldsymbol{X}_t := \alpha_t \boldsymbol{X} + \beta_t \boldsymbol{Z}$, assuming $\boldsymbol{X}$ and $\boldsymbol{Z}$ are independent.

The FM model then designs a vector field $u_t$ such that the ODE trajectory below generates $(p_t)_{t \in [0,1]}$, i.e., the result-

ing flow map $\Psi_t$ satisfies $p_t = (\Psi_t)_\# p_0$:

$$\frac{dx_t}{dt} = u_t(x_t). \quad (2)$$

To construct $u_t$, the FM model marginalizes over the *conditional vector field* $u_t(x|x_1)$:

$$u_t(x) = \int u_t(x|x_1) p(dx_1 | \boldsymbol{X}_t = x), \quad (3)$$

where $p(dx_1 | \boldsymbol{X}_t = x)$ represents the *posterior distribution*:

$$p(dx_1 | \boldsymbol{X}_t = x) = \frac{\exp\left(-\frac{\|x - \alpha_t x_1\|^2}{2\beta_t^2}\right)}{\int \exp\left(-\frac{\|x - \alpha_t x_1'\|^2}{2\beta_t^2}\right) p(dx_1')} p(dx_1).$$

Importantly, if $u_t(x|x_1)$ takes the following simple form:

$$u_t(x|x_1) = \frac{\dot{\beta}_t}{\beta_t} x + \frac{\dot{\alpha}_t \beta_t - \alpha_t \dot{\beta}_t}{\beta_t} x_1, \quad (4)$$

where the dot denotes differentiation with respect to $t$, then Liu et al. (2023, Theorem 3.3) and Lipman et al. (2022, Theorem 1) demonstrated that $u_t$ generates the probability path $(p_t)_{t \in [0,1]}$, assuming the ODE trajectory of Equation (2) exists on $[0, 1]$. This existence was rigorously established in Gao et al. (2024) under restrictive assumptions, excluding cases where $p$ is supported on a low-dim submanifold. For more general cases, see our results in Sections 4.1 and 5.1.

It turns out that the closed form of the conditional vector field $u_t(x|x_1)$ allows one to train a neural network to learn the vector field $u_t$ by minimizing the following loss function whose **unique** minimizer is $u_t(x)$ (Lipman et al., 2022):

$$\mathbb{E}_{\substack{t \in [0,1), \\ \boldsymbol{Z} \sim p_{\mathrm{prior}}, \boldsymbol{X} \sim p}} \left\| u_t^\theta(\alpha_t \boldsymbol{X} + \beta_t \boldsymbol{Z}) - \dot{\alpha}_t \boldsymbol{X} - \dot{\beta}_t \boldsymbol{Z} \right\|^2. \quad (5)$$

**Noise-to-signal ratio.** FM model with different scheduling functions can be unified through the *noise-to-signal ratio* (Shaul et al., 2024; Chen et al., 2024). We find it useful in our analysis as it simplifies the ODE dynamics and allows us to present our results more cleanly. Proofs of results in this section can be found in Appendix F.

Let $\alpha_t, \beta_t$ be strictly monotonic scheduling functions. The *noise-to-signal ratio* $\sigma_t := \beta_t / \alpha_t$ is defined for $t \in (0, 1]$. By monotonicity, $\sigma_t$ is invertible with inverse $t(\sigma)$. As $t$ increases from 0 to 1, $\sigma_t$ decreases from $\infty$ to 0. For $\sigma \in [0, \infty)$, we define $q_\sigma$ as the convolution of $p$ with the Gaussian distribution $\mathcal{N}(0, \sigma^2 I)$:

$$q_\sigma := p * \mathcal{N}(0, \sigma^2 I) = \int \mathcal{N}(\cdot | y, \sigma^2 I) p(dy). \quad (6)$$

Then, we have the following result.

**Proposition 2.1.** *For any $t \in (0, 1]$, define $A_t : \mathbb{R}^d \to \mathbb{R}^d$ by sending $x$ to $x/\alpha_t$. Then, $q_{\sigma_t} = (A_t)_\# p_t$.*

The probability path $q_{\sigma_t}$ satisfies the following ODE in $\sigma$.

**Proposition 2.2.** *For any $[a, b] \subset (0, 1]$, let $(x_t)_{t \in [a,b]}$ denote an ODE trajectory of Equation (2). Then, $(x_\sigma := x_{t(\sigma)}/\alpha_{t(\sigma)})_{\sigma \in (\sigma_b, \sigma_a]}$ satisfies the following ODE:*

$$\frac{dx_\sigma}{d\sigma} = -\sigma \nabla \log q_\sigma(x_\sigma), \qquad (7)$$

*where $q_\sigma(x)$ denotes the probability density of $q_\sigma$.*

The ODE model in Equation (7) generates a probability path $q_\sigma$ for $\sigma \in (0, \infty)$. During sampling, the ODE integrates backwards over $\sigma \in (0, \sigma_T]$ with end condition $x_{\sigma_T} = x$. By expressing our results in terms of the noise-to-signal ratio $\sigma$ rather than time $t$, we obtain a unified framework independent of specific scheduling functions, allowing for a more general and concise theoretical analysis.

## 2.2. Related Work

Chen et al. (2024) connects FM sampling to the mean shift algorithm (Comaniciu & Meer, 2002) via the denoiser, focusing on algorithmic strategies to identify high-curvature regions for better sampling. Pidstrigach (2022); Permenter & Yuan (2024) show that the denoiser converges to the projection when near the data support. We establish a general convergence result for almost every point and provide a characterization of trajectory evolution across different stages, going beyond prior interpretations of the sampling process as an approximate projection to data in Permenter & Yuan (2024). Gao & Li (2024) show that FM can only sample from the data support for discrete measures and analyzes local cluster absorption. However, their proof implicitly assumes ODE convergence and their local absorption analysis requires bounded prior support, which is not applicable to the common FM setting. We provide a rigorous proof of ODE convergence and analysis for full trajectory evolution.

The concurrent work by Baptista et al. (2025) analyzes the dynamical mechanisms underlying memorization in diffusion models with empirical measures. Their analysis of the ODE dynamics shares some similarities with our approach to discrete data distributions, e.g., the use of Voronoi diagrams. Their work also proposes some regularization techniques to mitigate memorization.

## 3. Denoiser and ODE Dynamics

It turns out that the vector field $u_t$ is fully determined by the so-called *denoiser*—the mean of the posterior distribution $p(\cdot|\mathbf{X}_t = x)$ (Karras et al., 2022). In this section, we first describe some basic properties of the denoiser, then illustrate a general attracting and absorbing dynamics of the ODE. Proofs and missing details can be found in Appendix C.

### 3.1. Basics of the Denoiser

By plugging Equation (4) into Equation (3), we have that

$$u_t(x) = \dot{\beta}_t/\beta_t \cdot x + (\dot{\alpha}_t \beta_t - \alpha_t \dot{\beta}_t)/\beta_t \cdot \mathbb{E}[\mathbf{X}|\mathbf{X}_t = x], \quad (8)$$

where $\mathbf{X} \sim p$, and $\mathbb{E}[\mathbf{X}|\mathbf{X}_t = x]$ is called the *denoiser* with the following form (with existence proved in Appendix C.1):

$$\mathbb{E}[\mathbf{X}|\mathbf{X}_t = x] = \int \frac{\exp\left(-\frac{\|x - \alpha_t y\|^2}{2\beta_t^2}\right) y}{\int \exp\left(-\frac{\|x - \alpha_t y'\|^2}{2\beta_t^2}\right) p(dy')} p(dy). \quad (9)$$

For brevity, we write $m_t(x) := \mathbb{E}[\mathbf{X}|\mathbf{X}_t = x]$. Since $m_t$ fully determines $u_t$, instead of learning $u_t$ directly, one can train a neural network $m_t^\theta$ to learn the denoiser $m_t$:

$$\mathbb{E}_{t \in [0,1), \mathbf{Z} \sim p_{\text{prior}}, \mathbf{X} \sim p} \left\| m_t^\theta(\alpha_t \mathbf{X} + \beta_t \mathbf{Z}) - \mathbf{X} \right\|^2. \quad (10)$$

Training with this loss can be more stable than with Equation (5) since for any $x \in \mathbb{R}^d$, $m_t(x)$ remains bounded while $u_t(x)$ can blow up to $\infty$ as $t \to 1$ (cf. Appendix C.2).

By direct computation, the ODE in $\sigma$ can also be expressed through the denoiser $m_\sigma(x) := \mathbb{E}[\mathbf{X}|\mathbf{X}_\sigma = x]$ as follows where $\mathbf{X}_\sigma := \mathbf{X} + \sigma \mathbf{Z}$:

$$\frac{dx_\sigma}{d\sigma} = -\sigma \nabla \log q_\sigma(x_\sigma) = -\frac{1}{\sigma}\left(m_\sigma(x_\sigma) - x_\sigma\right). \quad (11)$$

Notably, the ODE in $\sigma$ can be interpreted as **moving toward the denoiser $m_\sigma(x_\sigma)$**. For any $x$, one can explicitly write $m_\sigma(x)$ as follows.

$$m_\sigma(x) = \int \frac{\exp\left(-\frac{\|x - y\|^2}{2\sigma^2}\right) y}{\int \exp\left(-\frac{\|x - y'\|^2}{2\sigma^2}\right) p(dy')} p(dy). \quad (12)$$

### 3.2. Attracting and Absorbing

In Section 4.1, we will rigorously establish the existence of FM model ODE trajectory in $[0, 1)$ for any data distribution $p$ with a finite 2-moment. This sets the foundation for discussing the properties of the ODE trajectories.

Note that Equation (11) suggests that the trajectory moves toward the denoiser $m_\sigma(x_\sigma)$, which itself evolves along the trajectory, complicating the ODE dynamics. We address this by analyzing the geometric relationship between $m_\sigma(x)$ and the projection $\text{proj}_\Omega(x)$ onto certain closed sets $\Omega$. This reveals that the ODE trajectory exhibits two key properties: an *attracting property*—drawing trajectories toward $\Omega$, and an *absorbing property*—keeping trajectories within neighborhoods of $\Omega$. We characterize how the sampling process unfolds into distinct stages by identifying appropriate closed sets $\Omega$ with these properties. Below, we formulate these properties into two meta-theorems.

**Attracting toward sets.** Let $(x_\sigma)_{\sigma \in (\sigma_2, \sigma_1]}$ be an ODE trajectory. The distance $d_\Omega(x_\sigma)$ to a closed set $\Omega$ will decrease if the trajectory direction forms an *acute angle* with the projection direction (see e.g. Corollary B.15):

$$\langle m_\sigma(x_\sigma) - x_\sigma, \text{proj}_\Omega(x_\sigma) - x_\sigma \rangle > 0. \qquad (13)$$

See Figure 3 below for an illustration.

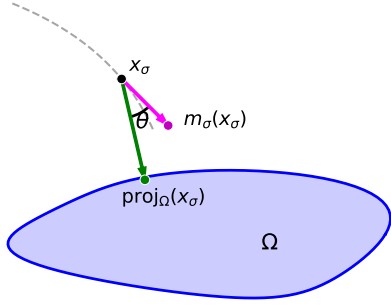

*Figure 3.* **Illustration of the acute angle condition.**

With some quantitative bound on the acute angle condition above, the ODE trajectory will be attracted toward $\Omega$.

**Theorem 3.1** (Informal - Attracting toward sets)**.** *Let $(x_\sigma)_{\sigma \in (\sigma_2, \sigma_1]}$ be an ODE trajectory of Equation (11) starting from some $x_{\sigma_1}$. Assume that the trajectory avoids the medial axis $\Sigma_\Omega$ of some closed $\Omega$ and satisfies the acute angle condition in a quantitative manner, then $d_\Omega(x_\sigma)$ will decrease along the trajectory, and $\lim_{\sigma \to 0} d_\Omega(x_\sigma) = 0$.*

See the formal version Theorem C.7 in Appendix C.3.

The requirement of avoiding the medial axis of $\Omega$ is trivially satisfied when $\Omega$ is convex (its medial axis is empty). More generally, one can rely on the absorbing property (discussed below) to ensure the trajectory stays off the medial axis.

**Absorbing by sets.** Given a set $\Omega$ in $\mathbb{R}^d$, we say $\Omega$ is *absorbing* for a FM ODE in $(\sigma_2, \sigma_1]$ if for any $x \in \Omega$, the ODE trajectory $(x_\sigma)_{\sigma \in (\sigma_2, \sigma_1]}$ starting at $x_{\sigma_1} = x$ will remain in $\Omega$ for all $\sigma \in (\sigma_2, \sigma_1]$. It turns out that the acute angel condition can essentially also guarantee that small neighborhoods of $\Omega$ are absorbing.

**Theorem 3.2** (Absorbing by sets)**.** *For any closed set $\Omega$ and $r > 0$, we consider the open neighborhood $B_r(\Omega)$.*

1. *Let $\overline{B_r(\Omega)}$ denote the closure of the set $B_r(\Omega)$. If $\overline{B_r(\Omega)} \cap \Sigma_\Omega = \emptyset$ and suppose for any $x \in \partial B_r(\Omega)$ and any $\sigma \in (\sigma_2, \sigma_1)$, one has $\langle m_\sigma(x) - x, \text{proj}_\Omega(x) - x \rangle > 0$, then $B_r(\Omega)$ is absorbing in $(\sigma_2, \sigma_1]$.*

2. *If there exists some $r_0 > 0$ such that $B_r(\Omega)$ is absorbing in $(\sigma_2, \sigma_1]$ for all $r \in (0, r_0)$, then $\Omega$ is absorbing in $(\sigma_2, \sigma_1]$ as well.*

*Remark* 3.3. Note that the absorbing property only depends on the denoiser's behavior in a fixed region rather than requiring a priori knowledge of how the denoiser evolves along an ODE trajectory. Once established, the absorbing property can propagate the acute angle condition to trajectories within this region, thereby enabling attracting property.

Next, we apply the meta-theorems to provide a broad description of FM ODE dynamics.

**Attracting and absorbing to the convex hull of data support.** By definition, the denoiser $m_\sigma(x)$ always lies in the convex hull of the support of the posterior distribution $p(\cdot | \boldsymbol{X}_\sigma = x)$ which is the same as $\text{conv}(\text{supp}(p))$. Due to convexity, $m_\sigma(x)$ always satisfies the acute angle condition with respect to $\text{conv}(\text{supp}(p))$. Consequently, the ODE trajectory is attracted toward and ultimately absorbed by $\text{conv}(\text{supp}(p))$.

**Proposition 3.4.** *Assume $p$ has a bounded support. For any $\sigma_1 > 0$, let $(x_\sigma)_{\sigma \in (0, \sigma_1]}$ be a flow matching ODE trajectory follows Equation (11). Then we have the following results for any $\sigma \in (0, \sigma_1]$:*

1. *If $x_{\sigma_1} \in \text{conv}(\text{supp}(p))$, then $x_\sigma \in \text{conv}(\text{supp}(p))$;*

2. *If $x_{\sigma_1} \notin \text{conv}(\text{supp}(p))$, then $x_\sigma$ moves toward $\text{conv}(\text{supp}(p))$ with the following decay guarantee:*

$$d_{\text{conv}(\text{supp}(p))}(x_\sigma) \le d_{\text{conv}(\text{supp}(p))}(x_{\sigma_1}) \cdot \sigma/\sigma_1.$$

See Appendix C.3.1 for the proof and a slight generalization to the case where $p$ is a Gaussian-smoothed bounded distribution. A more refined analysis of the trajectory for the initial and intermediate stages (focusing on the data mean and local clusters, respectively) is provided in Section 4. The terminal stage analysis in Section 5 requires more sophisticated techniques, as we need to develop absorbing properties that effectively avoid the medial axis of the data support—a key technical challenge for establishing convergence (cf. Appendix C.2).

## 4. Pre-Terminal Trajectory Analysis

As illustrated in Figure 2, the FM ODE trajectory overall moves toward the convex hull of the data support (Proposition 3.4). Furthermore, trajectory dynamics unfold in distinct stages. In this section, we first establish the well-posedness of the FM ODE to ensure the existence of trajectories. Then, we apply attracting and absorbing properties—toward the mean and local clusters—to provide a detailed analysis of the pre-terminal trajectory dynamics.

### 4.1. Well-posedness of FM ODEs for $t \in [0, 1)$

The following result establishes the existence and uniqueness of solutions to the FM ODE in $[0, 1)$ under very weak

assumptions, significantly expanding on previous work (Lipman et al., 2022; Gao et al., 2024) to contain cases where $p$ is supported on subspaces or submanifolds.

**Theorem 4.1.** *Assume $p$ has a finite 2-moment, then for every $x_0 \in \mathbb{R}^d$, there exists a unique solution $(x_t)_{t \in [0,1)}$ to Equation (2). Furthermore, the flow map $\Psi_t$ is continuous and satisfies that $(\Psi_t)_{\#} p_{prior} = p_t$ for all $t \in [0,1)$.*

The proof utilizes a careful analysis of the posterior covariance $\mathrm{Cov}[\boldsymbol{X}|\boldsymbol{X}_t = x]$ to establish local Lipschitzness and integrability of $u_t$. In addition, we show that the denoiser $m_t(x)$ grows at most linearly in $x$—a non-trivial bound obtained under the mild assumption that the data distribution $p$ has only a finite 2-moment. With these properties, we then apply the theory of continuity equations (see Ambrosio et al. (2008, Section 8.1)) to conclude the proof.

This result trivially extends to the $\sigma$ parameter and forms the foundation for analyzing pre-terminal trajectory properties.

### 4.2. Initial Stage of the Sampling Process

When $t = 0$, we have $m_0(x) \equiv \mathbb{E}[\boldsymbol{X}]$, suggesting the trajectory will initially approach the mean of data distribution. In this subsection, we quantitatively validate this intuition for a broad class of distributions.

**Proposition 4.2.** *Let $\delta \geq 0$ and $p_b$ be a distribution on $\mathbb{R}^d$ with a bounded support $\Omega := \mathrm{supp}(p_b)$. Let $p := p_b * \mathcal{N}(0, \delta^2 I)$. For a point $x_0$ with $|x_0 - \mathbb{E}[\boldsymbol{X}]| = R_0$ where $\boldsymbol{X} \sim p$, and for any parameter $0 < \zeta < 1$, define*

$$\sigma_{init}(\Omega, \zeta, R_0) := \sqrt{2R_0 \mathrm{diam}(\Omega)/\log(1 + \zeta R_0/\mathrm{diam}(\Omega))}.$$

*Then for all $\sigma_1 > \sqrt{\sigma_{init}(\Omega, \zeta, R_0)^2 + \delta^2}$, a trajectory $(x_\sigma)_{\sigma \in (\sqrt{\sigma_{init}(\Omega, \zeta, R_0)^2 + \delta^2}, \sigma_1]}$ starting from $x_{\sigma_1} = x_0$ will approach $\mathbb{E}[\boldsymbol{X}]$ with the rate:*

$$\|x_\sigma - \mathbb{E}[\boldsymbol{X}]\| < R_0 (\sigma^2 + \delta^2)^{\frac{1-\zeta}{2}} / (\sigma_1^2 + \delta^2)^{\frac{1-\zeta}{2}}.$$

Note that as $\zeta$ approaches 1, $\sigma_{\mathrm{init}}(\Omega, \zeta, R_0)$ decreases, extending the range of $\sigma$ that is applicable. However, the rate weakens as $\zeta$ gets closer to 1.

### 4.3. Intermediate Stage of the Sampling Process

As $\sigma$ decreases, trajectory behavior starts being influenced by coarse-scale geometry of the data, particularly its local clusters. When the trajectory lies close to a local cluster, it is attracted toward (and eventually absorbed by) the convex hull of that cluster——indicating robust feature emergence in the FM model. We formalize this notion via an assumption that characterizes a well-separated local cluster.

**Local Cluster Assumption.** Let $p$ be a probability measure on $\mathbb{R}^d$ and we say a set $S$ is a *local cluster* of $\Omega := \mathrm{supp}(p)$ if the following conditions hold:

1. $S$ is closed, bounded, and $\mathrm{diam}(S) = D < \infty$.
2. For all $x \in \Omega \backslash S$, $d_{\mathrm{conv}(S)}(x) > 2D$.

Then, the denoiser $m_\sigma(y)$ will be close to $\mathrm{conv}(S)$ for $y$ near $\mathrm{conv}(S)$ and when $\sigma$ is not too large.

**Proposition 4.3.** *Assume that $S$ is a local cluster of a probability measure $p$ satisfying the Local Cluster Assumption and $a_S := p(S) > 0$. Then, for any $x \in \mathbb{R}^d$ such that $d_{\mathrm{conv}(S)}(x) \leq D/2 - \epsilon$, we have that*

$$d_{\mathrm{conv}(S)}(m_\sigma(x)) \leq \mathrm{diam}(\Omega) \sqrt{(1 - a_S)/a_S} \, e^{-\frac{3D\epsilon}{2\sigma^2}}.$$

**Proposition 4.4.** *With the same assumptions as in Proposition 4.3, let $C_\epsilon^S := \frac{D/2 - \epsilon}{\mathrm{diam}(\Omega)\sqrt{\frac{1-a_S}{a_S}}}$, and define*

$$\sigma_0(S, \epsilon) := \begin{cases} \left(\frac{-3D\epsilon}{2\log(C_\epsilon^S)}\right)^{\frac{1}{2}}, & \text{if } C_\epsilon^S < 1, \\ \infty, & \text{otherwise.} \end{cases}$$

*Then for any $\sigma_1 < \sigma_0(S, \epsilon)$, $\overline{B_{D/2-\epsilon}(\mathrm{conv}(S))}$ is absorbing in $(0, \sigma_1]$ and any ODE trajectory starting from $x_{\sigma_1} \in \overline{B_{D/2-\epsilon}(\mathrm{conv}(S))}$ converges to $\mathrm{conv}(S)$ as $\sigma \to 0$.*

Note that when the weight $a_S$ of the local cluster $S$ is large, we do not necessarily have that $C_\epsilon^S < 1$. In this case, the cluster $S$, in fact, exhibits a stronger attracting force and the above result holds for all $\sigma_1 > 0$.

In Appendix J.1, we detail the synthetic data used in Figure 2 to validate the above Propositions 4.2 and 4.4. We also observe in Appendix J.2 that despite the theoretical constants not being tight (as is common with worst-case bounds), the qualitative behavior of an initial mean-attraction phase consistently holds in practice. Although Proposition 4.4 is developed under the local-cluster assumption, real-world datasets seldom satisfy it exactly. Empirically, however, we find that ODE trajectories still gravitate toward locally dense regions—even when clusters overlap—indicating that the dynamics are more robust than the assumption suggests (see Appendix J.3). We provide a partial theoretical explanation in Corollary H.1: for a distribution $p$ obtained by convolving a Gaussian with any measure that does satisfy the local-cluster assumption, we prove that dense regions remain attracting and absorbing for the flow. Together, the analysis and experiments point to a broader validity of the ODE dynamics beyond the confines of our assumptions.

While the above results are theoretical, they reveal how data geometry fundamentally shapes FM ODE dynamics—particularly, the cluster absorption results suggest a "locking" property where trajectories are systematically absorbed into local clusters. This property provides a theoretical foundation for why FM models achieve effective feature separation and mode coverage, as observed empirically in

Georgiev et al. (2023). It further suggests that embedding data into latent spaces with clear geometric structure (e.g., by object categories or visual attributes) could enhance robust feature learning.

Finally, as $\sigma$ approaches 0, trajectories are drawn to their nearest data points, with the nature of this attraction strongly governed by the fine-scale geometry of the data support——whether discrete or manifold-structured. We analyze this terminal convergence behavior in the next section.

## 5. Terminal Trajectory Analysis

The well-posedness of the FM ODE in $[0, 1)$ ensures the probability path under flow map $\Psi_t$ approaches the data distribution. However, distributional convergence alone does not guarantee trajectory convergence, as pathological cases like winding paths around data points may exist (see Figure 1b). The convergence of ODE trajectories—equivalently, the existence of flow map $\Psi_1$ at $t = 1$—is crucial for generating samples stably and for models like the consistency model (Song et al., 2023), which learns the flow map $\Psi_1$.

In this section, we establish the convergence of ODE trajectories at $t = 1$ for a broad class of data distributions. With this result, we analyze the equivariance of $\Psi_1$ under geometric transformations and discuss implications for memorization behavior.

### 5.1. Convergence of ODE Trajectories at $t = 1$

The convergence of ODE trajectory at $t = 1$, when expressed in terms of $\sigma$, requires the integration of $\frac{dx_\sigma}{d\sigma} = -\frac{1}{\sigma}(m_\sigma(x_\sigma) - x_\sigma)$ to remain convergent as $\sigma \to 0$. The divergence of $\int_0^T \frac{1}{\sigma} d\sigma$ creates a potential singularity that must be counteracted by a rapid diminishing of $\|m_\sigma(x_\sigma) - x_\sigma\|$. We study the denoiser's terminal behavior at a fixed point $x$ to understand when diminishing may occur.

**Theorem 5.1** (Convergence of denoiser to projection). *Let $p$ be a probability distribution with a finite 2-moment and support $\Omega$. Then for all $x \in \mathbb{R}^d \backslash \Sigma_\Omega$:*

$$\lim_{\sigma \to 0} m_\sigma(x) = \mathrm{proj}_\Omega(x),\,[1]$$

*where $\Sigma_\Omega$ denotes the medial axis of $\Omega$.*

Since $\mathrm{proj}_\Omega(x) = x$ precisely when $x \in \Omega$, the term $m_\sigma(x_\sigma) - x_\sigma$ diminishes if $x_\sigma$ is attracted to $\Omega$. The convergence only holds for $x \in \mathbb{R}^d \backslash \Sigma_\Omega$ since projection is not well-defined at the medial axis, which also causes the denoiser's Lipschitz constant to blow up (see Appendix C.2), preventing the use of standard ODE theory like the Picard-Lindelöf theorem.

---

[1]We also establish the convergence of $m_t$ in Corollary D.3 which is a nontrivial consequence of the convergence of $m_\sigma$.

We address these issues by a refined denoiser's convergence result with rate guarantee, which occurs at an $O(\sigma^\zeta)$ rate for any $0 < \zeta < 1$. See Appendix D for the proof and details—for example, the convergence rate is $\sqrt{m}\sigma + O(\sigma^2)$ for distributions on an $m$-dimensional submanifold and exponential for discrete distributions. This convergence rate yields a strengthened version of the absorbing result in Section 3.2, ensuring that trajectories are: (1) absorbed near $\Omega$ to avoid the medial axis, and (2) attracted to $\Omega$ rapidly enough to ensure convergence. Our theoretical analysis works for data distributions satisfying:

**Assumption 5.2** (Regularity assumptions for data distribution). Let $p$ be a probability measure on $\mathbb{R}^d$ with a finite 2-moment satisfying the following properties:

1. The reach $\tau_\Omega$ of the support $\Omega := \mathrm{supp}(p)$ is positive[2];

2. There exist constants $k \geq 0$ and $c > 0$ such that for any radius $R > 0$, there is a constant $C_R > 0$ satisfying the following: for any small radius $0 \leq r < c$ and any $x \in B_R(0) \cap \Omega$, we have $p(B_r(x)) \geq C_R r^k$.

Any discrete distribution satisfies the assumptions with $k = 0$. Moreover, $p$ satisfies the assumptions with $k = m$ when supported on an $m$-dimensional linear subspace or compact submanifold with positive reach, provided $p$ has a finite 2-moment and a non-vanishing density (cf. Lemma I.1). We emphasize that the positive reach condition is not restrictive and is a common assumption to ensure that the data manifold has no "sharp turns" in $R^d$ (Niyogi et al., 2008; Fefferman et al., 2016). This condition holds for common smooth compact submanifolds (Lieutier & Wintraecken, 2024) like spheres and tori.

We now state our main result:

**Theorem 5.3.** *For $p$ satisfying Assumption 5.2, we have*

1. *$\Psi_1(x) := \lim_{t \to 1} \Psi_t(x)$ exists for $x \in \mathbb{R}^d$ a.e.*

2. *$\Psi_1$ is a measurable map and $(\Psi_1)_{\#}p_{prior} = p$.*

*Furthermore, we have the following estimate of the convergence rate of the flow map. Recall that $\sigma_t := \beta_t/\alpha_t$, then, we have that for any fixed $0 < \zeta < 1$,*

$$\|\Psi_1(x) - \Psi_t(x)\| = O(\sigma_t^{\zeta/2}).$$

This theorem establishes the existence and convergence of the flow map for general data distributions. The convergence rate can be further refined for specific cases:

**Theorem 5.4.** *When $p$ is supported on a submanifold or a discrete set, we have the following convergence results:*

***Manifold.*** *Let $M \subset \mathbb{R}^d$ be an $m$-dim closed submanifold with positive reach and bounded second fundamental form*

---

[2]The support $\Omega$ can be $\mathbb{R}^d$ and in this case the reach $\tau = \infty$.

*up to its first-order derivatives. Assume that $p$ is supported on $M$, has a finite 2-moment, and has a density given by $p(dx) = \rho(x)\text{vol}_M(dx)$, where $\rho : M \to \mathbb{R}$ is smooth and nonvanishing. Then, for a.e. $x \in \mathbb{R}^d$, we have that*

$$\|\Psi_1(x) - \Psi_t(x)\| = O(\sqrt{\sigma_t});$$

**Discrete.** *If $p = \sum_{i=1}^{N} a_i x_i$ denotes a discrete probability measure, then for a.e. $x \in \mathbb{R}^d$, we have that*

$$\|\Psi_1(x) - \Psi_t(x)\| = O(\sigma_t).$$

We examine a tractable special case to assess our convergence rate bounds: a standard Gaussian distribution on a subspace, where closed-form solutions exist.

*Example* 5.5 (Data supported on subspaces). For any $0 < m \le d$, consider the subspace $\mathbb{R}^m \subset \mathbb{R}^d$. We express any point $\boldsymbol{x} \in \mathbb{R}^d$ as $\boldsymbol{x} = (x, y)$, where $x \in \mathbb{R}^m$ and $y \in \mathbb{R}^{d-m}$. Assume that the probability measure $p$ is supported on $\mathbb{R}^m$ and satisfies Assumption 5.2. One can show that FM ODE trajectories allow a dimension reduction in the following manner. For any initial point $\boldsymbol{x}_0 = (x_0, y_0) \in \mathbb{R}^d$, the FM ODE trajectory is given by

$$\boldsymbol{x}_t = (x_t, \beta_t y_0), \text{ for any } t \in [0, 1], \tag{14}$$

where $(x_t)_{t \in [0,1]}$ is the trajectory of the FM ODE on $\mathbb{R}^m$ with initial point $x_0$, and with $p$ regarded as a distribution on $\mathbb{R}^m$; see Proposition E.2 for a proof.

Let $\alpha_t = t$ and $\beta_t = 1-t$, and let $p$ be the standard Gaussian on $\mathbb{R}^m$. By Lemma E.1 and Equation (14), we have that

$$\boldsymbol{x}_t = \left( \sqrt{(1-t)^2 + t^2} x_0, (1-t)y_0 \right).$$

Also, when $m = 0$, $p = \delta_{\boldsymbol{0}}$ is supported on a single point. Then, the denoiser is always $\boldsymbol{0}$ and the ODE trajectory is given by $(\boldsymbol{x}_t = (1-t)\boldsymbol{x}_0)_{t \in [0,1]}$. In both cases, $\|\boldsymbol{x}_1 - \boldsymbol{x}_t\| = \Theta(1 - t) = \Theta(\sigma_t)$.

This example demonstrates our rate's optimality for discrete distributions, while suggesting potential improvement for manifolds: the current $O(\sqrt{\sigma_t})$ rate might be improved to $O(\sigma_t)$. This conjecture is supported by the linear convergence rate $O(\sigma)$ of the ODE's distribution path $q_\sigma \to p$ (see proposition below), which suggests the trajectory should converge at the same rate.

**Proposition 5.6.** *For any probability measure $p$ with a finite 2-moment, we have that $d_{\text{W},2}(q_\sigma, p) = O(\sigma)$.*

An important practical implication of the convergence rates in Theorems 5.3 and 5.4 is that, during the terminal stage, the ODE trajectory exhibits minor movements suggesting one can use fewer sampling steps to generate samples without sacrificing quality.

**Equivariance under geometric transformations.** Having established the existence of the flow map $\Psi_1 : \mathbb{R}^d \to \mathbb{R}^d$ under mild assumptions, we now investigate how ambient space geometry affects the flow maps through their behavior under geometric transformations. This analysis has practical implications for stability under data augmentation and reveals important equivariance properties.

We examine how FM flow maps transform under similarity transformations $T : \mathbb{R}^d \to \mathbb{R}^d$ of the form $T(x) = \gamma(\boldsymbol{O}x + b)$, where $\gamma > 0$ is a scaling factor, $\boldsymbol{O}$ is an orthogonal matrix, and $b$ is a translation vector. These transformations include any combination of scaling, rotation and translation.

For a data distribution $p$ satisfying Assumption 5.2, let $\bar{p} := T_{\#}p$ denote the transformed distribution. To relate the flow maps $\overline{\Psi}_1$ (for $\bar{p}$) and $\Psi_1$ (for $p$), we identify that the flow for the transformed data $\bar{p}$ can be obtained from the flow for the original data $p$ by choosing appropriate scheduling functions $\bar{\alpha}_t$ and $\bar{\beta}_t$ with respect to the original functions $\alpha_t$ and $\beta_t$. Specifically, taking $\bar{\alpha}_t := s_t \alpha_t / \gamma$ and $\bar{\beta}_t := s_t \beta_t$ where $s_t$ is any positive smooth function with $s_0 = 1$, $s_1 = \gamma$ (or simply $s_t \equiv 1$ when $\gamma = 1$), we establish:

**Proposition 5.7** (Equivariance under similarity transformations). *For any $x \in \mathbb{R}^d$ and $t \in [0, 1)$, we have that*

$$\overline{\Psi}_t(\boldsymbol{O}x) = s_t(\boldsymbol{O}\Psi_t(x) + \alpha_t b).$$

*Whenever $\Psi_1(x)$ exists (this holds for a.e. $x \in \mathbb{R}^d$ by Theorem 5.3), we have that $\overline{\Psi}_1(\boldsymbol{O}x)$ exists and satisfies*

$$\overline{\Psi}_1(\boldsymbol{O}x) = \gamma(\boldsymbol{O}\Psi_1(x) + b).$$

*Remark* 5.8 (Data distribution on affine subspaces). By Proposition 5.7, we can generalize Example 5.5 to cases where $p$ is supported on any affine subspace $A \subset \mathbb{R}^d$ (e.g., a point translated away from origin or a shifted linear subspace). The idea is simple: first apply a rigid transformation to map $A$ to $\mathbb{R}^{\dim(A)} \subset \mathbb{R}^d$, then apply the FM model there. This suggests that for data distribution supported on an affine subspace, we can reduce computation by first projecting onto that subspace, training an FM model there, and extending it back to ambient space via this result.

### 5.2. Terminal Absorbing Behavior and Memorization

In this subsection, we focus on the case when $p = \sum_{i=1}^{n} a_i \, \delta_{x_i}$ is supported on a discrete set, as this represents an important scenario corresponding to empirical target distributions derived from training data. We provide a detailed characterization of the terminal stage and show that each point $x_i$ exhibits strong attracting behavior during this stage, which is connected to the memorization in diffusion models (Carlini et al., 2023; Wen et al., 2024).

We let $\Omega = \{x_1, \ldots, x_n\}$ denote the support of $p$. For any small $\epsilon > 0$, we define the $\epsilon$-*shrunk Voronoi cells* as

$$V_i^\epsilon := \left\{ x : \|x - x_i\|^2 \le \|x - x_j\|^2 - \epsilon^2, \, \forall x_j \ne x_i \in \Omega \right\}.$$

Note that $V_i^\epsilon$ is convex and as $\epsilon \to 0$, $V_i^\epsilon$ expands to the classical Voronoi cell $V_i$ of $x_i$ with $\cup_{i=1}^n V_i = \mathbb{R}^d$.

We introduce $\sigma_0(V_i^\epsilon)$ such that $V_i^\epsilon$ is absorbing for $(0, \sigma_0(V_i^\epsilon))$. Specifically, let $\text{sep}(x_i) := d_{\Omega \setminus \{x_i\}}(x_i)$ denote the separation of $x_i$ and we introduce a constant

$$C_{i,\epsilon}^\Omega = \frac{2\,\text{sep}(x_i)}{\text{sep}^2(x_i) - \epsilon^2} \cdot \sqrt{\frac{1 - a_i}{a_i}} \cdot \text{diam}(\Omega)$$

for any parameter $\epsilon \in (0, \text{sep}(x_i)/2)$, where $a_i$ is the weight of $x_i$ in $p$. Then the time $\sigma_0(V_i^\epsilon)$ is defined as

$$\sigma_0(V_i^\epsilon) = \begin{cases} \infty, & \text{if } C_{i,\epsilon}^\Omega \le 1, \\ \frac{\epsilon}{2}\left(\log(C_{i,\epsilon}^\Omega)\right)^{-1/2}, & \text{if } C_{i,\epsilon}^\Omega > 1. \end{cases}$$

The constant $\sigma_0(V_i^\epsilon)$ is the time when the ODE trajectory is attracted to $V_i^\epsilon$ and with larger weight $a_i$ or higher separation $\text{sep}(x_i)$, the time $\sigma_0(V_i^\epsilon)$ is larger, showing strong attraction early on.

The following result shows that after approaching the mean and then being attracted to a local cluster, the ODE trajectory will eventually be attracted to the nearest data point.

**Proposition 5.9.** *Fix an arbitrary $0 < \sigma_1 < \sigma_0(V_i^\epsilon)$. Then, for any $y \in V_i^\epsilon$, the ODE trajectory $(x_\sigma)_{\sigma \in (0, \sigma_0]}$ starting from $x_{\sigma_0} = y$ will stay inside $V_i^\epsilon$, i.e., $x_\sigma \in V_i^\epsilon$. Furthermore, $(x_\sigma)_{\sigma \in (0, \sigma_0]}$ will converge to $x_i$ as $\sigma \to 0$.*

**Discussion on memorization.** Memorization occurs when a model perfectly fits training data and fails to generalize. This is relevant to FM models since the unique solution in Equation (5) or Equation (10) regarding empirical data only reproduces training data. The constant $\sigma_0(V_i^\epsilon)$ indicates attraction strength of each training sample—higher values (from larger weights $a_i$ or more isolated points) suggest increased memorization risk. This explains empirical findings of increased memorization for duplicate samples in Somepalli et al. (2023), as duplicates raise $a_i$ in the empirical distribution. For CIFAR-10 with $\epsilon = 1.0$, the mean $\sigma_0(V_i^\epsilon)$ across training images is approximately 0.17, corresponding to the final quarter of EDM's sampling steps (Karras et al., 2022). This suggests training in this critical final stage should not target optimality to avoid memorization.

We now provide a more formal connection between the terminal behavior of the ODE trajectory and the memorization phenomenon. For a neural network denoiser $m_\sigma^\theta$, the corresponding ODE trajectory is:

$$\frac{dx_\sigma^\theta}{d\sigma} = -\frac{1}{\sigma}(m_\sigma^\theta(x_\sigma) - x_\sigma^\theta). \tag{15}$$

The following result shows that an asymptotically optimally trained denoiser $m_\sigma^\theta$ merely reproduces the training data.

**Proposition 5.10** (Memorization of asymptotically optimal denoiser). *Let $p = \sum_{i=1}^n a_i \delta_{x_i}$, and let $m_\sigma^\theta : \mathbb{R}^d \to \mathbb{R}^d$*

*be a smooth map. Assume there exists a function $\phi(\sigma)$ with $\lim_{\sigma \to 0} \phi(\sigma) = 0$ such that $\|m_\sigma^\theta(x) - m_\sigma(x)\| \le \phi(\sigma)$ for all $x \in \mathbb{R}^d$. Then, for any $i = 1, \ldots, n$, there exists $\sigma_0(V_i^\epsilon, \phi) > 0$ such that for all $0 < \sigma_0 < \sigma_0(V_i^\epsilon, \phi)$ and any $y \in V_i^\epsilon$, the ODE trajectory $(x_\sigma^\theta)_{\sigma \in (0, \sigma_0]}$ for Equation (15), starting from $x_{\sigma_0} = y$, converges to $x_i$ as $\sigma \to 0$.*

*If further, both limits $\lim_{\sigma \to 0} x_\sigma^\theta$ and $\lim_{\sigma \to 0} m_\sigma^\theta(x_\sigma^\theta)$ known to exist, then $\lim_{\sigma \to 0} \|m_\sigma^\theta(x_\sigma^\theta) - x_\sigma^\theta\| = 0$.*

This proposition shows that for an FM model to be capable of generalization, the near-terminal denoiser itself must generalize—i.e., it must approximate projection onto the true underlying data manifold rather than simply projecting onto the training points. This insight motivates careful tuning of denoiser training near the terminal time. We validate these insights empirically in both synthetical (Appendix J.1) and image dataset (Appendix J.2).

## 6. Discussion

Our study significantly enhances the theoretical foundation for FM models by establishing a connection between data geometry and FM ODE dynamics. This leads to interesting practical implications; for example: (1) The FM ODE trajectory direction's initial alignment with the mean and its terminal time convergence suggest one can use more sparse sampling resources in these stages and reallocate more resources to the intermediate stage where the denoiser evolves more significantly which aligns with empirical findings in Esser et al. (2024). (2) The interaction between flow trajectories and data geometry through attracting and absorbing behavior reveals how the same dataset can exhibit distinct sampling trajectories when embedded in different spaces. This could be utilized to optimize latent space for improved generation and facilitate stable fine-tuning through careful adaptation when integrating new data. (3) Identifying the importance of terminal stages in memorization suggests targeted regularization in training such as regularizing the Jacobian of denoiser to avoid collapsing to locally constant maps; see more discussion in Remark C.3. Looking ahead, we aim to explore these directions to develop more understanding of memorization versus generalization, as well as more efficient diffusion models with steerable generation.

Theoretically, our analysis assumes that $\boldsymbol{X} \sim p$ and $\boldsymbol{Z} \sim p_{\text{prior}}$ are independent when constructing the probability path $(p_t)_{t \in [0,1]}$. It will be intriguing in future work to investigate whether this analysis can be extended to settings where $\boldsymbol{X}$ and $\boldsymbol{Z}$ are dependent. Such cases naturally arise, for example, when applying rectification techniques as in (Liu et al., 2023) or when employing known coupling methods to enhance flow matching, as explored in (Pooladian et al., 2023; Tong et al., 2024).

## Impact Statement

This paper studies the theoretical properties of the flow-matching model, a novel generative model that has been widely adopted in practice. Our investigation on the memorization phenomenon could potentially help to design better variants that do not leak private information and have a positive societal impact.

## Acknowledgements

This work is partially supported by NSF grants CCF-2112665, CCF-2217058, CCF-2310411 and CCF-2403452.

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

# Appendix

# Table of Contents for Appendix

## A. Table of Symbols and Roadmap of Theoretical Results

In this section, we provide a table of symbols used in the paper and a roadmap of the theoretical results in Figure 4.

| Symbol | Meaning / Description |
|---|---|
| $\mathbb{R}^d$ | $d$-dimensional Euclidean space. |
| $\|\cdot\|$ | Euclidean norm in $\mathbb{R}^d$. |
| $\text{supp}(p)$ | Support of the probability measure $p$. |
| $\boldsymbol{X} \sim p$ | Random variable $\boldsymbol{X}$ with distribution $p$. |
| $\mathsf{M}_2(p)$ | Second moment of $p$, i.e., $\int \|x\|^2 p(dx)$. |
| $\text{diam}(\Omega)$ | Diameter of a set $\Omega \subset \mathbb{R}^d$. |
| $\overline{\Omega}$ | Closure of a set $\Omega$. |
| $d_\Omega(x)$ | Point-to-set distance from $x$ to $\Omega$, i.e., $d_\Omega := \inf_{y \in \Omega} \|x - y\|$. |
| $\text{proj}_\Omega(x)$ | Projection of $x$ onto a closed set $\Omega \subset \mathbb{R}^d$. |
| $\text{conv}(\Omega)$ | Convex hull of a set $\Omega$. |
| $\Sigma_\Omega$ | Medial axis of a closed set $\Omega \subset \mathbb{R}^d$. |
| $\text{lfs}_\Omega(x)$ | Local feature size of $x$ regarding set $\Omega$, i.e. distance to medial axis $\Sigma_\Omega$. |
| $\tau_\Omega$ | Reach of a set $\Omega \subset \mathbb{R}^d$. |
| $\text{Inj}(M)$ | Injectivity radius of a manifold $M$. |
| $\text{sep}(x_i)$ | Separation scale w.r.t. a discrete point $x_i$. |
| $\delta_x$ | Dirac delta measure at $x$. |
| $\mathcal{N}(\mu, \Sigma)$ | Gaussian distribution with mean $\mu$ and covariance $\Sigma$. |
| $p * q$ | Convolution of probability measures $p$ and $q$. |
| $q_\sigma = p * \mathcal{N}(0, \sigma^2 I)$ | The blurred (or noisy) distribution at noise level $\sigma$. |
| $\text{NN}_\Omega(x)$, $\widehat{p}_{\text{NN}(x)}$ | Nearest-neighbor set of $x$ in a discrete set $\Omega$, and the measure restricted to that set. |
| $x_t$ | A state (trajectory point) indexed by time $t \in [0, 1]$ in the FM ODE. |
| $x_\sigma$ | A state (trajectory point) indexed by noise level $\sigma \in (0, \infty)$ in the FM ODE. |
| $\boldsymbol{X}_t$ | The random variable $\alpha_t \boldsymbol{X} + \beta_t \boldsymbol{Z}$, where $\boldsymbol{Z} \sim \mathcal{N}(0, I)$. |
| $\boldsymbol{X}_\sigma$ | The random variable $\boldsymbol{X} + \sigma \boldsymbol{Z}$, where $\boldsymbol{Z} \sim \mathcal{N}(0, I)$. |
| $p(\cdot|\boldsymbol{X}_t = x)$, $p(\cdot|\boldsymbol{X}_\sigma = x)$ | Posterior distributions of $\boldsymbol{X}$ given $\boldsymbol{X}_t = x$ and given $\boldsymbol{X}_\sigma = x$, respectively. |
| $m_\sigma(x)$ | The denoiser in $\sigma$ parameter, i.e. $\mathbb{E}[\boldsymbol{X} \mid \boldsymbol{X}_\sigma = x]$. |
| $m_t(x)$ | The denoiser in $t$ parameter (implicitly dependent on $\alpha_t, \beta_t$); i.e. $\mathbb{E}[\boldsymbol{X} \mid \boldsymbol{X}_t = x]$. |
| $\nabla \log q_\sigma$ | Score function of $q_\sigma$. |
| $d_{\text{W},s}$ | $s$-Wasserstein distance. |
| $u_t(x)$ | Vector field in the FM ODE for $t \in [0, 1)$. |
| $\Psi_t$ | The flow map of the FM ODE. |
| $p_t = (\Psi_t)_\# p_0$ | The pushforward distribution of an initial distribution $p_0$ under $\Psi_t$. |

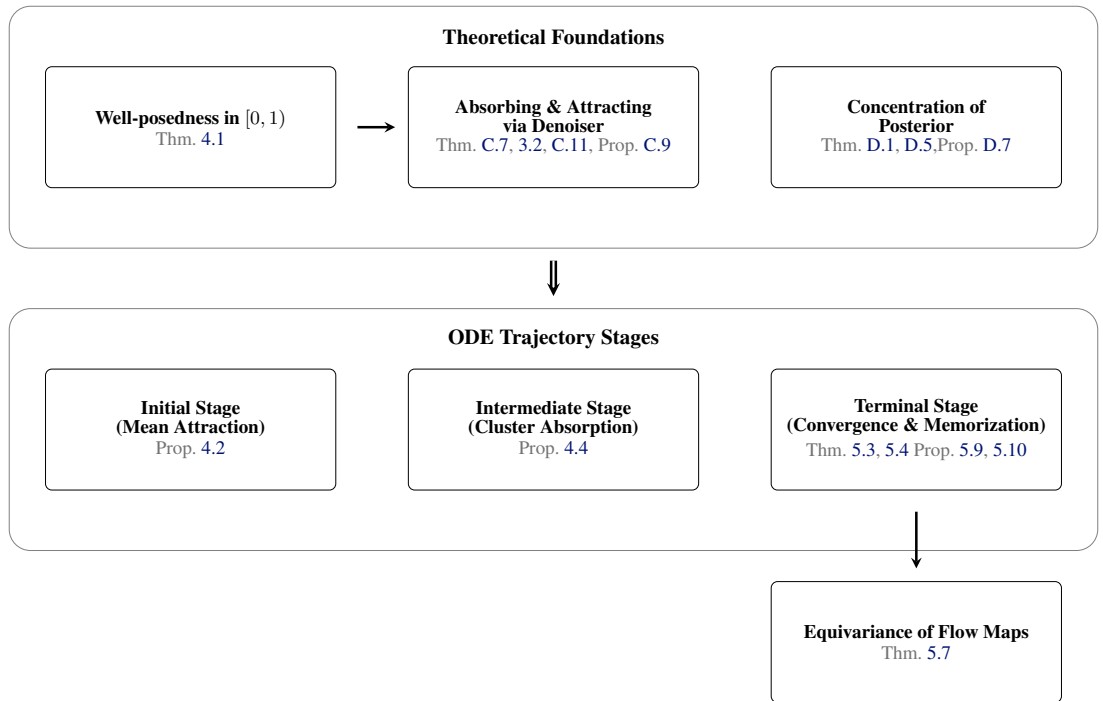

*Figure 4*. **Roadmap of theoretical results.** Our analysis builds upon three foundational components: (1) well-posedness of FM ODEs in $[0, 1)$, establishing existence and uniqueness of trajectories—setting the foundation for subsequent analysis, (2) attracting and absorbing properties derived from denoiser behavior, and (3) concentration results for posterior distributions. These foundations enable us to characterize the full evolution of FM ODE trajectories through three distinct stages: initial mean attraction, intermediate cluster absorption, and terminal convergence with memorization implications. The convergence at terminal time further allows us to establish equivariance properties of flow maps. Together, these results provide a complete geometric understanding of FM ODE dynamics.

## B. Geometric Notions and Results

In this section, we review basic concepts from convex geometry and metric geometry, and establish several results which will be used in our later proofs. These geometric results are also of independent interest and may be applicable in other contexts.

### B.1. Convex Geometry Notions and Results

In this subsection, we collect some basic notions and results in convex geometry that are used in the proofs. Our main reference is the book (Hug & Weil, 2020).

We first introduce the definition of a convex set and convex function.

**Definition B.1** (Convex set). A set $K \subset \mathbb{R}^d$ is called a *convex set* if for any $x_1, \ldots, x_n \in K$ and $0 \leq \alpha_1, \ldots, \alpha_n \leq 1$ such that $\sum_{i=1}^n \alpha_i = 1$, we have that $\sum_{i=1}^n \alpha_i x_i \in K$.

**Definition B.2** (Convex function). A function $f : \mathbb{R}^d \to \mathbb{R}$ is called a *convex function* if for any $x, y \in \mathbb{R}^d$ and $0 \leq \alpha \leq 1$, we have that $f(\alpha x + (1 - \alpha)y) \leq \alpha f(x) + (1 - \alpha)f(y)$.

An intermediate result that the sublevel sets $\{f < c\}$ or $\{f \leq c\}$ of a convex function $f$ are convex sets; see Hug & Weil (2020, Remark 2.6).

We now introduce the definition of the convex hull of a set.

**Definition B.3** (Convex hull (Hug & Weil, 2020, Definition 1.3, Theorem 1.2)). The *convex hull* of a set $\Omega \subset \mathbb{R}^d$ is the smallest convex set that contains $\Omega$ and is denoted by $\mathrm{conv}(\Omega)$. Additionally, we have that

$$\mathrm{conv}(\Omega) = \left\{ \sum_{i=1}^n \alpha_i x_i : k \in \mathbb{N}, x_i \in \Omega, \alpha_i \in [0, 1], \sum_{i=1}^n \alpha_i = 1 \right\}.$$

Let $\Omega$ be a set, and $x \in \Omega$. We say a hyperplane give by a linear function $H$ is a *supporting hyperplane* of $\Omega$ at $x$ if the following conditions hold:

1. $H(x) = 0$.

2. $H(y) \geq 0$ for all $y \in \Omega$.

For a closed convex set $\Omega$, every boundary point of $\Omega$ has a supporting hyperplane.

**Proposition B.4** (Supporting hyperplane (Hug & Weil, 2020, Theorem 1.16))**.** *Let $K$ be a closed convex set in $\mathbb{R}^d$ and $x \in \partial K$. Then, there exists a supporting hyperplane of $K$ at $x$.*

We collect some basic properties regarding a convex set $K$ and its distance function $d_X(x) := \inf_{y \in K} \|x - y\|$.

**Proposition B.5.** *Let $K$ be a convex set in $\mathbb{R}^d$ and $\Omega$ be a set in $\mathbb{R}^d$. Then, we have that*

1. *For each $x \in \mathbb{R}^d$, there exists a unique point $\mathrm{proj}_K(x) \in K$ such that $\|x - \mathrm{proj}_K(x)\| = d_K(x)$.*

2. *The distance function $d_K(x)$ is a convex function.*

3. *For any $r \geq 0$, the $r$-thickening of $K$, defined as $B_r(K) := \{x \in \mathbb{R}^d : d_K(x) < r\}$, is a convex set.*

4. *The diameter of $\Omega$ is the same as the diameter of its convex hull, that is $\mathrm{diam}(\Omega) = \mathrm{diam}(\mathrm{conv}(\Omega))$.*

5. *Let $a > 0$, then a set $\Omega$ is convex if and only if $a \cdot \Omega$ is convex.*

## B.2. Metric Geometry Notions and Results

Let $\Omega \subset \mathbb{R}^d$ be a closed subset. Recall that $\Sigma_\Omega$ denotes the medial axis of $\Omega$ and $d_\Omega : \mathbb{R}^d \to \mathbb{R}$ is defined by $d_\Omega(x) := \inf_{y \in \Omega} \|x - y\|$. We now consider certain properties of the projection function $\mathrm{proj}_\Omega : \Sigma_\Omega^c \to \Omega$, where $\Sigma_\Omega^c := \mathbb{R}^d \backslash \Sigma_\Omega$.

We first recall the definition of the local feature size in Amenta & Bern (1998) with a slight generalization that we consider all points in $\mathbb{R}^d$ instead of only points in $\Omega$.

**Definition B.6** ((Amenta & Bern, 1998))**.** *For any $x \in \mathbb{R}^d$, we define the local feature size $\mathrm{lfs}_\Omega(x)$ of $x$ as $\mathrm{lfs}_\Omega(x) := d_{\Sigma_\Omega}(x)$.*

For any $x \in \Sigma_\Omega^c$, we let $x_\Omega := \mathrm{proj}_\Omega(x)$. We consider the following set for any $x \in \Sigma_\Omega^c$:

$$T_\Omega(x) := \left\{ t \geq 0 : \mathrm{proj}_\Omega \left( x_\Omega + t \frac{x - x_\Omega}{\|x - x_\Omega\|} \right) = x_\Omega. \right\}.$$

**Lemma B.7.** *We have the following characterizations of $T_\Omega(x)$.*

- *$T_\Omega(x)$ is an interval.*

- *For any $t \in T_\Omega(x)$, we have that $x_\Omega + t \frac{x - x_\Omega}{\|x - x_\Omega\|} \notin \Sigma_\Omega$.*

- *We let $R_\Omega(x) := \sup T_\Omega(x)$. If $0 < R_\Omega(x) < \infty$, we have that $x_\Omega + R_\Omega(x) \frac{x - x_\Omega}{\|x - x_\Omega\|} \in \overline{\Sigma_\Omega}$.*

- *$[0, d_\Omega(x) + \mathrm{lfs}_\Omega(x)) \subset T_\Omega(x)$.*

*Proof of Lemma B.7.* The first three items follow from the pioneering work Federer (1959, Theorem 4.8); see also Delfour & Zolésio (2011, Theorem 6.2) for more details.

We provide a proof for the last item. First of all, it is straightforward to see that $[0, d_\Omega(x)] \subset T_\Omega(x)$. Now, we assume that $r := R_\Omega(x) \in [d_\Omega(x), d_\Omega(x) + \mathrm{lfs}_\Omega(x))$. This implies that

$$x_\Omega + r \frac{x - x_\Omega}{\|x - x_\Omega\|} = x + (r - d_\Omega(x)) \frac{x - x_\Omega}{\|x - x_\Omega\|} \in \overline{\Sigma_\Omega}$$

and hence

$$\mathrm{lfs}_\Omega(x) = d_{\Sigma_\Omega}(x) = d_{\overline{\Sigma_\Omega}}(x) \leq r - d_\Omega(x) < \mathrm{lfs}_\Omega(x).$$

This is a contradiction and hence $R_\Omega(x) \geq d_\Omega(x) + \mathrm{lfs}_\Omega(x)$. This concludes the proof. $\square$

Next, for any point $b \in \Omega$, we analyze the angle $\angle x x_\Omega b$ for any $b \in \Omega$. The following lemma is a slight variant of Federer (1959, Theorem 4.8 (7)).

**Lemma B.8.** *For any $x \in \Sigma_\Omega^c$ and any $t > 0$ such that $t \in T_\Omega(x)$, the following holds for any $b \in \Omega$:*

$$\langle x - x_\Omega, x_\Omega - b \rangle \geq -\frac{\|x_\Omega - b\|^2 \|x - x_\Omega\|}{2t}.$$

*Proof.* By definition of $T_\Omega(x)$, we have that $\mathrm{proj}_\Omega \left( x_\Omega + t \frac{x - x_\Omega}{\|x - x_\Omega\|} \right) = x_\Omega$. Therefore, we have that

$$\left\| x_\Omega + t \frac{x - x_\Omega}{\|x - x_\Omega\|} - b \right\|^2 \geq d_\Omega^2 \left( x_\Omega + t \frac{x - x_\Omega}{\|x - x_\Omega\|} \right) = t^2$$

$$\|x_\Omega - b\|^2 + 2t \left\langle x_\Omega - b, \frac{x - x_\Omega}{\|x - x_\Omega\|} \right\rangle + t^2 \geq t^2$$

$$2t \langle x_\Omega - b, x - x_\Omega \rangle \geq -\|x_\Omega - b\|^2 \|x - x_\Omega\|$$

$$\langle x - x_\Omega, x_\Omega - b \rangle \geq -\frac{\|x_\Omega - b\|^2 \|x - x_\Omega\|}{2t}$$

$\square$

When $\Omega$ is convex, then $T_\Omega(x) = [0, \infty)$. In this way, we have the following corollary.

**Corollary B.9.** *If $\Omega$ is convex, then for any $b \in \Omega$ and any $x \in \mathbb{R}^d$ we have that $\langle x - x_\Omega, x_\Omega - b \rangle \geq 0$.*

This control of the angle $\angle x x_\Omega b$ allows us to derive the following result that bounds the distance between $b \in \Omega$ and the projection $x_\Omega = \mathrm{proj}_\Omega(x)$ in terms of the distance between $b$ and $x$.

**Lemma B.10.** *For any $x \in \Sigma_\Omega^c$ and any $t > 0$ such that $t \in T_\Omega(x)$, we have that for any $b \in \Omega$,*

$$\|x - b\|^2 \geq d_\Omega(x)^2 + \|b - x_\Omega\|^2 \left( 1 - \frac{d_\Omega(x)}{t} \right).$$

*Proof.* The case when $x \in \Omega$ trivially holds. Below we consider the case when $x \in \Sigma_\Omega^c \cap \Omega^c$. In this case, $d_\Omega(x) > 0$ as $\Omega$ is a closed set.

By the law of cosines, we have that

$$\cos(\angle x x_\Omega b) = \frac{d_\Omega(x)^2 + \|b - x_\Omega\|^2 - \|x - b\|^2}{2 d_\Omega(x) \|b - x_\Omega\|}$$

Suppose $\|x - b\|^2 < d_\Omega(x)^2 + \|b - x_\Omega\|^2 (1 - \frac{d_\Omega(x)}{t})$, then we have that

$$\cos(\angle x x_\Omega b) = \frac{d_\Omega(x)^2 + \|b - x_\Omega\|^2 - \|x - b\|^2}{2 d_\Omega(x) \|b - x_\Omega\|}$$

$$> \frac{\|d_\Omega(x)^2 + \|b - x_\Omega\|^2 - d_\Omega(x)^2 - \|b - x_\Omega\|^2 + \|b - x_\Omega\|^2 \frac{d_\Omega(x)}{t}}{2 d_\Omega(x) \|b - x_\Omega\|}$$

$$= \frac{\|b - x_\Omega\|^2}{2t}.$$

By applying Lemma B.8 to $b$ and $x$, we have that

$$\langle x - x_\Omega, x_\Omega - b \rangle \geq -\frac{\|x_\Omega - b\|^2 \|x - x_\Omega\|}{2t}.$$

This implies the following estimate for the cosine of the angle $\angle x x_\Omega b$:

$$\cos(\angle x x_\Omega b) = \frac{\langle x - x_\Omega, b - x_\Omega \rangle}{\|x - x_\Omega\| \|b - x_\Omega\|} \leq \frac{\|x_\Omega - b\|^2}{2t}.$$

This contradicts the inequality above and hence we must have that

$$\|x - b\|^2 \geq d_\Omega(x)^2 + \|b - x_\Omega\|^2 \left(1 - \frac{d_\Omega(x)}{t}\right). \qquad \square$$

We then have the following corollary which will be used in the proof of Theorem D.1.

**Corollary B.11.** *Fix any $x \in \Sigma_\Omega^c$, any $t \in [d_\Omega(x), d_\Omega(x) + \mathrm{lfs}_\Omega(x))$ and any $\epsilon > 0$. Then, we have that*

$$B_{\sqrt{d_\Omega(x)^2 + \epsilon^2(1 - d_\Omega(x)/t)}}(x) \cap \Omega \subseteq B_\epsilon(x_\Omega) \cap \Omega.$$

*Proof.* For any $b \in B_{\sqrt{d_\Omega(x)^2 + \epsilon^2(1 - d_\Omega(x)/t)}}(x) \cap \Omega$, we have that

$$\|x - b\|^2 < d_\Omega(x)^2 + \epsilon^2 \left(1 - \frac{d_\Omega(x)}{t}\right).$$

By Lemma B.10, we have that

$$\|x - b\|^2 \geq d_\Omega(x)^2 + \|b - x_\Omega\|^2 \left(1 - \frac{d_\Omega(x)}{t}\right).$$

Combining the two inequalities, we have that $\|b - x_\Omega\|^2 < \epsilon^2$ and hence $b \in B_\epsilon(x_\Omega) \cap \Omega$. $\qquad \square$

Finally, we derive the local Lipschitz continuity of the projection function.

**Lemma B.12** (Local Lipschitz continuity of the projection). *For any $\epsilon > 0$ and for any $x, y \in \mathbb{R}^d$ such that $\mathrm{lfs}_\Omega(x) > \epsilon$ and $\mathrm{lfs}_\Omega(y) > \epsilon$, we have that*

$$\|x_\Omega - y_\Omega\| \leq \left(\frac{\max\{d_\Omega(x), d_\Omega(y)\}}{\epsilon} + 1\right) \|x - y\|.$$

*Proof.* Since $\mathrm{lfs}(x) > \epsilon$ and $\mathrm{lfs}_\Omega(y) > \epsilon$, by Lemma B.7 we have that

$$\mathrm{proj}_\Omega \left(x_\Omega + (\epsilon + d_\Omega(x))\frac{x - x_\Omega}{\|x - x_\Omega\|}\right) = x_\Omega$$

and

$$\mathrm{proj}_\Omega \left(y_\Omega + (\epsilon + d_\Omega(y))\frac{y - y_\Omega}{\|y - y_\Omega\|}\right) = y_\Omega.$$

By applying Lemma B.8 to $x, x_\Omega, y_\Omega$ and separately to $y, y_\Omega, x_\Omega$, we have that

$$\langle x_\Omega - y_\Omega, x - x_\Omega \rangle \geq -\frac{\|x_\Omega - y_\Omega\|^2}{2(\epsilon + d_\Omega(x))}d_\Omega(x),$$

$$\langle y_\Omega - x_\Omega, y - y_\Omega \rangle \geq -\frac{\|x_\Omega - y_\Omega\|^2}{2(\epsilon + d_\Omega(y))}d_\Omega(y).$$

By adding the two inequalities about the inner products, we have that

$$\langle x_\Omega - y_\Omega, x - x_\Omega - y + y_\Omega \rangle \geq -\frac{\|x_\Omega - y_\Omega\|^2}{2(\epsilon + d_\Omega(x))}d_\Omega(x) - \frac{\|x_\Omega - y_\Omega\|^2}{2(\epsilon + d_\Omega(y))}d_\Omega(y),$$

$$\langle x_\Omega - y_\Omega, x - y \rangle - \|x_\Omega - y_\Omega\|^2 \geq -\frac{\|x_\Omega - y_\Omega\|^2}{2(\epsilon + d_\Omega(x))}d_\Omega(x) - \frac{\|x_\Omega - y_\Omega\|^2}{2(\epsilon + d_\Omega(y))}d_\Omega(y),$$

$$\langle x_\Omega - y_\Omega, x - y \rangle \geq \|x_\Omega - y_\Omega\|^2 - \frac{\|x_\Omega - y_\Omega\|^2}{2(\epsilon + d_\Omega(x))}d_\Omega(x) - \frac{\|x_\Omega - y_\Omega\|^2}{2(\epsilon + d_\Omega(y))}d_\Omega(y).$$

Let $m = \max\{d_\Omega(x), d_\Omega(y)\}$, then we have that

$$0 < 1 - \frac{m}{m + \epsilon} \leq \left(1 - \frac{d_\Omega(x)}{2\left(\epsilon + d_\Omega(x)\right)} - \frac{d_\Omega(y)}{2\left(\epsilon + d_\Omega(y)\right)}\right).$$

Therefore, by Cauchy-Schwarz inequality, we have that

$$\|x_\Omega - y_\Omega\|^2 \left(1 - \frac{m}{m + \epsilon}\right) \leq \langle x_\Omega - y_\Omega, x - y\rangle \leq \|x_\Omega - y_\Omega\|\|x - y\|.$$

This implies $\|x_\Omega - y_\Omega\| \leq \left(\frac{m+\epsilon}{\epsilon}\right)\|x - y\|$. $\qquad\square$

### B.2.1. DIFFERENTIABILITY OF THE DISTANCE FUNCTION

Let $\Omega \subset \mathbb{R}^d$ be any closed subset. For any $x \in \mathbb{R}^d$, we let $P_\Omega(x) := \{y \in \Omega : \|x - y\| = d_\Omega(x)\}$ denote the set of points in $\Omega$ that achieve the infimum. When $x$ is not in the medial axis of $\Omega$, the set $P_\Omega(x)$ is the singleton set $\{\text{proj}_\Omega(x)\}$.

Interestingly, there is the following result result showing the existence of one sided directional derivatives for $d_\Omega$ by de Mises (1937) (see also Białożyt (2023) for a more recent English treatment).

**Lemma B.13** ((de Mises, 1937)). *For any vector $v \in \mathbb{R}^d$, the one-sided directional derivative of $d_\Omega$ at $x \in \mathbb{R}^d\backslash\Omega$ in the direction $v$ exists and is given by*

$$D_v d_\Omega(x) = \inf\left\{-\frac{\langle v, y - x\rangle}{\|y - x\|} : y \in P_\Omega(x)\right\}.$$

Motivated by the this result, we mimick the proof and establish the following result for the squared distance function $d_\Omega^2$. Notice that, we can remove the constraint for $x \notin \Omega$.

**Lemma B.14.** *For any $x \in \mathbb{R}^d\backslash\Sigma_\Omega$, the directional derivative of $d_\Omega^2$ at $x$ exists and is given by*

$$D_v d_\Omega^2(x) = -2\langle v, \text{proj}_\Omega(x) - x\rangle.$$

*Proof.* Without loss of generality, we assume that $x = 0$ is the origin and $v = (c, 0, \ldots, 0)$ for some $c > 0$ (one can achieve these by applying rigid transformations).

We let $x_\Omega = \text{proj}_\Omega(x)$, $x_t = tv$ and $x_t^* = \text{proj}_\Omega(x_t)$. Since $\|x_t^* - x_t\| \leq \|x_\Omega - x_t\|$, we have that

$$\|x_t^*\|^2 - \|x_\Omega\|^2 \leq 2ct(x_t^{*,(1)} - x_0^{*,(1)}),$$

where $x_t^{*,(1)}$ denotes the first coordinate of $x_t^*$. Note that $\|x_t^*\| = \|x_t^* - x\| \geq \|x_\Omega - x\| = \|x_\Omega\|$. So we have that

$$0 \leq x_t^{*,(1)} - x_0^{*,(1)}.$$

Now we have that by the cosine rule of the triangle formed by $x_t = tv$, $x = 0$ and $x_t^*$, we have that

$$d_\Omega^2(x_t) = \|x_t^*\|^2 + t^2c^2 - 2t\langle v, x_t^* - x\rangle.$$

Therefore, we have that

$$\frac{d_\Omega^2(x_t) - d_\Omega^2(x)}{t} = \frac{\|x_t^*\|^2 - \|x_\Omega\|^2}{t} - 2\langle v, x_t^* - x\rangle.$$

As discussed above, we have that

$$0 \leq \frac{\|x_t^*\|^2 - \|x_\Omega\|^2}{t} \leq 2c(x_t^{*,(1)} - x_0^{*,(1)}).$$

By continuity of $\text{proj}_\Omega$ outside $\Sigma_\Omega$, we have that $x_t^* \to x^*$ as $t \to 0$. Therefore, we have that

$$\lim_{t\to 0}\frac{d_\Omega^2(x_t) - d_\Omega^2(x)}{t} = -2\langle v, x^* - x\rangle = -2\langle v, \text{proj}_\Omega(x) - x\rangle. \qquad\square$$

**Corollary B.15.** *Let $(x_t)_t$ be a differentiable curve in $x \in \mathbb{R}^d\backslash\Sigma_\Omega$, then we have that $d_\Omega^2(x_t)$ is differentiable with respect to $t$ and we have that*

$$\frac{d}{dt}d_\Omega^2(x_t) = D_{\dot{x}_t}d_\Omega^2(x_t) = -2\langle \dot{x}_t, \text{proj}_\Omega(x_t) - x_t\rangle.$$

# C. Denoiser and FM ODE: Singularity, Attracting and Absorbing

## C.1. Basics About the Denoiser

**Well-definedness of the denoiser.** Although denoisers have been widely used in various works, their well-definedness has not been rigorously established in the literature—specifically, whether the integral defining the conditional mean $\mathbb{E}[\boldsymbol{X}|\boldsymbol{X}_t = x]$ exists. We address this gap in the following proposition.

**Proposition C.1.** *If $p$ has a finite 1-moment, then $p(\cdot|\boldsymbol{X}_t = x)$ also has a finite 1-moment for any $t \in [0, 1)$, making $m_t(x) := \mathbb{E}[\boldsymbol{X}|\boldsymbol{X}_t = x]$ well-defined. The same applies to $m_\sigma$ for $\sigma \in (0, \infty)$.*

*Proof of Proposition C.1.* For the normalizing factor $Z = \int \exp\left(-\frac{\|x - \alpha_t y'\|^2}{2\beta_t^2}\right) p(dy')$, we note that $0 < \exp\left(-\frac{\|x - \alpha_t y'\|^2}{2\beta_t^2}\right) \leq 1$. Hence, the factor $Z$ must be positive and bounded.

Now, we consider the following integral

$$\int \exp\left(-\frac{\|x - \alpha_t y\|^2}{2\beta_t^2}\right) \|y\| p(dy) \leq \int \|y\| p(dy) < \infty.$$

The last inequality follows from the fact that $p$ has a finite 1-moment. Hence, the posterior distribution $p(\cdot|\boldsymbol{X}_t = x)$ has a finite 1-moment and the denoiser $m_t$ is well-defined. $\qquad\square$

**An alternative parametrization for $\sigma$.** The backward integration w.r.t. $\sigma$ in Equation (7) might be cumbersome in analysis and we alternatively use the parameter $\lambda := -\log(\sigma)$. We let $\sigma(\lambda)$ denote the inverse function. Then, when $\sigma$ changes from $\infty$ to 0, $\lambda$ changes from $-\infty$ to $\infty$. For an ODE trajectory $(x_\sigma)_{\sigma \in (0,\infty)}$, we define $(x_\lambda := x_{\sigma(\lambda)})_{\lambda \in (-\infty,\infty)}$. For any $x \in \mathbb{R}^d$, we define $m_\lambda(x) := m_{\sigma(\lambda)}(x)$. Then, the ODE in $\lambda$ has a concise form:

$$\frac{dx_\lambda}{d\lambda} = m_\lambda(x_\lambda) - x_\lambda. \tag{16}$$

*Proof of Equation (16).* Since $\lambda = -\log \sigma$, we have that

$$\frac{d\lambda}{d\sigma} = -\frac{1}{\sigma}$$

and thus the ODE equation Equation (11) becomes

$$\begin{aligned}
\frac{dx_\lambda}{d\lambda} &= \frac{dx_{\sigma(\lambda)}}{d\sigma}\frac{d\sigma}{d\lambda} = -\sigma\nabla\log q_\sigma(x_\sigma)\cdot(-\sigma) \\
&= -x_\sigma + \frac{\int y\exp\left(-\frac{\|x_\sigma - y\|^2}{2\sigma^2}\right)p(dy)}{\int \exp\left(-\frac{\|x_\sigma - y'\|^2}{2\sigma^2}\right)p(dy')} \\
&= m_\lambda(x_\lambda) - x_\lambda. \qquad\square
\end{aligned}$$

**Jacobians of the denoiser and data covariance.** We point out that the denoiser, under some mild condition on the data distribution $p$, is differentiable and its Jacobian is inherently connected with the covariance matrix of the posterior distribution $p(\cdot|\boldsymbol{X}_t = x)$ (or $p(\cdot|\boldsymbol{X}_\sigma = x)$). Similar formulas for computing the Jacobian have been utilized before for various purposes; see, for example, Zhang et al. (2024, Lemma B.2.1), Gao et al. (2024, Lemma 4.1), Ben-Hamu et al. (2024, Proposition 4.1) and Rissanen et al. (2025, Equation (8)). Moreover, the covariance formula in Proposition C.2 is a direct consequence of higher order generalization of Tweedie's formula, which has been studied in previous works (see, e.g., Efron (2011), Meng et al. (2021)).

**Proposition C.2.** *Assume that $p$ has a finite 2-moment. For any $t \in [0, 1)$, we have that $m_t$ is differentiable. In particular, the Jacobian $\nabla_x m_t$ can be explicitly expressed as follows for any $x \in \mathbb{R}^d$:*

$$\begin{aligned}
\nabla_x m_t(x) &= \frac{\alpha_t}{2\beta_t^2}\iint (z - z')(z - z')^T p(dz|\boldsymbol{X}_t = x)p(dz'|\boldsymbol{X}_t = x) \\
&= \frac{\alpha_t}{\beta_t^2}\mathrm{Cov}[\boldsymbol{X}|\boldsymbol{X}_t = x].
\end{aligned}$$

*Furthermore, if we let $\sigma = \sigma_t$, then for any $\sigma \in (0, \infty)$:*

$$\nabla_x m_\sigma(x) = \frac{1}{2\sigma^2} \iint (z - z')(z - z')^T p(dz | \boldsymbol{X}_\sigma = x) p(dz' | \boldsymbol{X}_\sigma = x)$$

$$= \frac{1}{\sigma^2} \mathrm{Cov}[\boldsymbol{X} | \boldsymbol{X}_\sigma = x].$$

*Proof of Proposition C.2.* Recall that

$$m_t(x) = \frac{\int \exp\left(-\frac{\|x - \alpha_t z\|^2}{2\beta_t^2}\right) z \, p(dz)}{\int \exp\left(-\frac{\|x - \alpha_t z\|^2}{2\beta_t^2}\right) p(dz)} \quad \text{and} \quad p(dz | \boldsymbol{X}_t = x) = \frac{\exp\left(-\frac{\|x - \alpha_t z\|^2}{2\beta_t^2}\right) p(dz)}{\int \exp\left(-\frac{\|x - \alpha_t z\|^2}{2\beta_t^2}\right) p(dz)}.$$

We let $w_t(x, z) := \exp\left(-\frac{\|x - \alpha_t z\|^2}{2\beta_t^2}\right)$. Then,

$$\nabla_x m_t(x) = \int z \nabla_x \left(\frac{\exp\left(-\frac{\|x - \alpha_t z\|^2}{2\beta_t^2}\right)}{\int \exp\left(-\frac{\|x - \alpha_t z'\|^2}{2\beta_t^2}\right) p(dz')}\right) p(dz)$$

$$= \left(\int w_t(x, z') p(dz')\right)^{-2} \int z \left(w_t(x, z) \left(-\frac{x - \alpha_t z}{\beta_t^2}\right)^T \int w_t(x, z') p(dz')\right) p(dz)$$

$$- \left(\int w_t(x, z') p(dz')\right)^{-2} \int z \left(w_t(x, z) \int w_t(x, z') \left(-\frac{x - \alpha_t z'}{\beta_t^2}\right)^T p(dz')\right) p(dz)$$

$$= \left(\int w_t(x, z') p(dz')\right)^{-2} \iint w_t(x, z) w_t(x, z') z \left(-\frac{x - \alpha_t z}{\beta_t^2} + \frac{x - \alpha_t z'}{\beta_t^2}\right)^T p(dz) p(dz')$$

$$= \frac{\alpha_t}{\beta_t^2} \left(\int w_t(x, z') p(dz')\right)^{-2} \iint w_t(x, z) w_t(x, z') z \, (z - z')^T p(dz) p(dz')$$

$$= \frac{\alpha_t}{2\beta_t^2} \left(\int w_t(x, z') p(dz')\right)^{-2} \iint w_t(x, z) w_t(x, z') (z - z') (z - z')^T p(dz) p(dz')$$

$$= \frac{\alpha_t}{2\beta_t^2} \iint (z - z')(z - z')^T p(dz | \boldsymbol{X}_t = x) p(dz' | \boldsymbol{X}_t = x).$$

The second equation follows from a similar calculation. $\qquad \square$

*Remark* C.3. We have established in Corollary D.3 that for any $x \notin \Sigma_\Omega$ (where $\Sigma_\Omega$ denotes the medial axis of the support $\Omega$ of $p$), the denoiser satisfies

$$m_t(x) \to \mathrm{proj}_\Omega(x) \quad \text{as } t \to 1.$$

We conjecture that a similar convergence holds for the Jacobian, that is,

$$\nabla_x m_t(x) \to \nabla_x \mathrm{proj}_\Omega(x) \quad \text{as } t \to 1.$$

Note that when $\Omega$ is a discrete set, the projection $\mathrm{proj}_\Omega(x)$ is locally constant almost everywhere, so its Jacobian $\nabla_x \mathrm{proj}_\Omega(x)$ is identically zero. This suggests a potential pitfall: if $\nabla_x m_t(x)$ also collapses to zero, the model may effectively "memorize" training points. Regularizing $\nabla_x m_t(x)$ to prevent such collapse could thus help mitigate memorization.

## C.2. Denoiser and ODE Dynamics: Terminal Time Singularity

The terminal time is referred to as the time $t = 1$ (or $\sigma = 0$, $\lambda \to \infty$) in the FM model. The convergence of the ODE trajectory at the terminal time relies on the terminal time regularity of the vector field. We now elucidate two types of singularities of the vector field that arise at the terminal time in the FM model, one due to the ODE formulation and the other due to the data geometry.

**Singularity due to the ODE formulation.** Recall that the vector field $u_t$ is given by

$$u_t(x) = \frac{\dot{\beta}_t}{\beta_t}x + \frac{\dot{\alpha}_t\beta_t - \alpha_t\dot{\beta}_t}{\beta_t} \mathbb{E}[\boldsymbol{X}|\boldsymbol{X}_t = x].$$

Since the denominator $\beta_t$ approaches 0 as $t \to 1$, the vector field $u_t$ faces an issue of division by zero when approaching the terminal time. Similarly, the vector field in $\sigma$ formulation is given by $-\frac{1}{\sigma}(m_\sigma(x) - x)$, which also faces the same issue when $\sigma \to 0$. This singularity is intrinsic to the flow matching ODE formulation. In the following proposition, we show that when the data distribution $p$ is not fully supported, the limit $\lim_{t\to 1} \|u_t(x)\|$ diverges to infinity for almost all $x$ outside the support of $p$, whereas it remains bounded when $p$ is fully supported.

**Proposition C.4.** *Assume that $\alpha, \beta : [0,1] \to \mathbb{R}$ are smooth, and $\dot{\alpha}_1, \dot{\beta}_1$ exist and are non zero. Let $\Omega := \mathrm{supp}(p)$ and let $\Sigma_\Omega$ denote its medial axis. Then, we have the following properties:*

- *If $p$ is fully supported, i.e., $\Omega = \mathbb{R}^d$, and has a Lipschitz density, then for any $x \in \mathbb{R}^d$, the vector field $u_t(x)$ is uniformly bounded for all $t \in [0,1)$.*

- *If $p$ is not fully supported, i.e., $\Omega \neq \mathbb{R}^d$, then for any $x \notin \Omega \cup \Sigma_\Omega$, $\lim_{t\to 1} \|u_t(x)\| = \infty$.*

*Proof of Proposition C.4.* Let $\Omega := \mathrm{supp}(p)$. Recall that

$$u_t(x) = (\log \beta_t)'x + \beta_t (\alpha_t/\beta_t)' m_t(x)$$
$$= \dot{\beta}_t \frac{x - \alpha_t m_t(x)}{\beta_t} + \dot{\alpha}_t m_t(x).$$

Then, we have that

$$\|m_t(x) - \mathrm{proj}_\Omega(x)\| \leq \|m_{\sigma_t}(x/\alpha_t) - \mathrm{proj}_\Omega(x/\alpha_t)\| + \|\mathrm{proj}_\Omega(x/\alpha_t) - \mathrm{proj}_\Omega(x)\|.$$

By Lemma B.12, there exists a positive constant $C_x$ such that $\|\mathrm{proj}_\Omega(x) - \mathrm{proj}_\Omega(y)\| \leq C_x\|x - y\|$ for any $y$ close to $x$. Therefore,

$$\|\mathrm{proj}_\Omega(x/\alpha_t) - \mathrm{proj}_\Omega(x)\| \leq C_x \left\| \frac{x}{\alpha_t} - x \right\| = O(1 - \alpha_t).$$

Now, when $\Omega = \mathbb{R}^d$, by Corollary D.6, we have that

$$\|m_{\sigma_t}(x/\alpha_t) - \mathrm{proj}_\Omega(x/\alpha_t)\| = O(\sigma_t) = O(\beta_t).$$

In this case, $\mathrm{proj}_\Omega(x) = x$. This implies that for $t$ close to 1, we have that

$$\left\| \frac{x - \alpha_t m_t(x)}{\beta_t} \right\| \leq O\left( \frac{1 - \alpha_t}{\beta_t} \right) + O(1).$$

The right hand side is bounded since $\lim_{t\to 1} \frac{1-\alpha_t}{\beta_t} = \lim_{t\to 1} \frac{-\dot{\alpha}_t}{\dot{\beta}_t} = \frac{-\dot{\alpha}_1}{\dot{\beta}_1}$ exists. Therefore, the vector field $u_t(x)$ is uniformly bounded for all $t \in [0,1)$.

If $\Omega \neq \mathbb{R}^d$ and $x \notin \Omega \cup \Sigma_\Omega$, then $\mathrm{proj}_\Omega(x) \neq x$. By Corollary D.3, we have that $m_t(x) \to \mathrm{proj}_\Omega(x)$ as $t \to 1$. This implies that $\|x - \alpha_t m_t(x)\| \to \|x - \mathrm{proj}_\Omega(x)\| > 0$. Then, as the denominator $\beta_t \to 0$, we have that $\lim_{t\to 1} \|u_t(x)\| = \infty$. $\square$

Another way to interpret the singularity in the ODE formulation is through the transformation of the ODE in terms of $\sigma$ in Equation (16), where the singularity arises due to the presence of the $1/\sigma$ term.

A seemingly straightforward approach to addressing this blowup is to reformulate the ODE in terms of $\lambda$ as given in Equation (16):

$$\frac{dx_\lambda}{d\lambda} = m_\lambda(x_\lambda) - x_\lambda, \quad \lambda \in (-\infty, \infty),$$

with the terminal time being $\lambda \to \infty$. In this formulation, there is no denominator approaching zero, seemingly eliminating the singularity. However, for the ODE trajectory to converge as $\lambda \to \infty$, a necessary condition is that $\lim_{\lambda \to \infty} \| m_\lambda(x_\lambda) - x_\lambda \| = 0$. This is precisely what we establish in proving our convergence result in Theorem 5.3.

**Singularity due to the data geometry.** When $p$ is not fully supported, the medial axis $\Sigma_\Omega$ of the data support plays a crucial role in the singularity of the denoiser $m_\sigma$ which results in discontinuity of the limit $\lim_{\sigma \to 0} m_\sigma(x)$. In this case, when the ODE is transformed into the $\lambda$, the vector field $u_\lambda$ does not have a uniform Lipschitz bound for all $\lambda \in (a, \infty)$ and hence the typical ODE theory such as Picard-Lindelöf theorem can not be directly applied to analyze the flow matching ODEs.

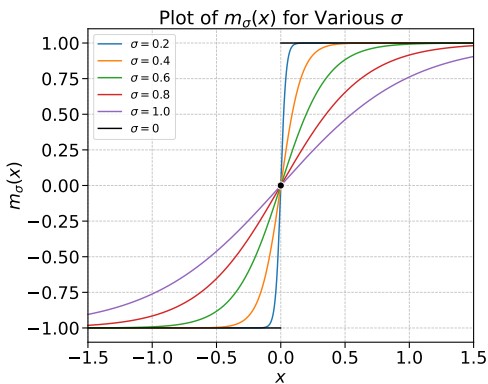

Figure 5. **The denoiser $m_\sigma(x)$ for the two point example with various $\sigma$.**

The discontinuity behavior can be illustrated by the following simple example of a two-point data distribution which can be easily extended to higher dimensions.

*Example* C.5. Let $p = \frac{1}{2}\delta_{-1} + \frac{1}{2}\delta_1$ be a probability measure on $\mathbb{R}^1$. Then, the support $\Omega := \mathrm{supp}(p) = \{-1, 1\}$ is just a two-point set. The medial axis is the singleton $\Sigma_\Omega = \{0\}$ whose distance to either point is 1. Now, we can explicitly write down the denoiser $m_\sigma$ as follows:

$$m_\sigma(x) = \frac{-\exp\left(-\frac{(x+1)^2}{2\sigma^2}\right) + \exp\left(-\frac{(x-1)^2}{2\sigma^2}\right)}{\exp\left(-\frac{(x+1)^2}{2\sigma^2}\right) + \exp\left(-\frac{(x-1)^2}{2\sigma^2}\right)}. \tag{17}$$

Notice that when $\sigma$ approaches $0$,

- The denoiser $m_\sigma$ is converging to the function $f : \mathbb{R}^1 \to \{-1, 0, 1\}$ with $f(x) = \begin{cases} 1 & x > 0 \\ 0 & x = 0 \\ -1 & x < 0 \end{cases}$.

- A singularity of $m_\sigma$ is emerging at $\Sigma_\Omega = \{0\}$: the derivative $\frac{dm_\sigma(0)}{dx}$ is blowing up when $\sigma \to 0$.

A full characterization of the limit $\lim_{\sigma \to 0} m_\sigma(x)$ for discrete data distribution will be given in Appendix D where the discontinuity often arises at the medial axis of the data support. All these singularities pose challenges in theoretical analysis of the flow matching ODEs and particularly in the convergence of the ODE trajectory when approaching the terminal time. The data geometry singularity is more challenging to handle, especially the discontinuity behavior of the limit of the denoiser near the medial axis.

Our resolution of the convergence of FM ODE trajectory (Theorem 5.3) will be based on the attracting and absorbing property of the ODE dynamics so that the trajectory will avoid the singularities and converge to the data support.

### C.3. Denoiser and ODE Dynamics: Attracting and Absorbing

The following result on the convergence of ODE under asymptotically vanishing perturbation will be used often in the proofs of this section.

**Lemma C.6.** *Let $(y_t \geq 0)_{t \in [t_1, \infty)}$ be a differentiable trajectory of non-negative real numbers satisfying the following differential inequality*

$$\frac{d\,y_t}{dt} \leq -k\,y_t + \phi(t),$$

*where $k > 0$ and $\phi : \mathbb{R} \to \mathbb{R}$. Then, we have:*

1. *For any $t_2 > t_1$ we have that*

$$y_{t_2} \leq e^{-k(t_2-t_1)} y_{t_1} + \int_{t_1}^{t_2} e^{-k(t_2-t)} \phi(t) dt.$$

2. *If $\lim_{t \to \infty} \phi(t) = 0$, then $\lim_{t \to \infty} y_t = 0$;*

*Proof of Lemma C.6.* By multiplying the integrating factor $e^{kt}$, we have that

$$\frac{d\,e^{kt} y_t}{dt} = e^{kt} \frac{d\,y_t}{dt} + k e^{kt} y_t \leq \phi(t) e^{kt}.$$

Then for all $t_2 > t_1$, we have

$$e^{kt_2} y_{t_2} \leq e^{kt_1} y_{t_1} + \int_{t_1}^{t_2} e^{kt} \phi(t) dt,$$

$$y_{t_2} \leq e^{-k(t_2-t_1)} y_{t_1} + \int_{t_1}^{t_2} e^{-k(t_2-t)} \phi(t) dt.$$

This proves Item 1.

For Item 2, we will first show that $y_t$ is bounded. As $\phi(t)$ decays to zero as $t$ goes to infinity, so will be $|\phi(t)|$, and hence there exists some constant $C > 0$ such that $|\phi(t)| \leq C$ for all $t \geq t_1$. Then, we have that

$$y_{t_2} \leq e^{-k(t_2-t_1)} y_{t_1} + C \int_{t_1}^{t_2} e^{-k(t_2-t)} dt,$$

$$\leq e^{-k(t_2-t_1)} y_{t_1} + \frac{C}{k}(1 - e^{-k(t_2-t_1)}),$$

$$\leq e^{-k(t_2-t_1)} y_{t_1} + \frac{C}{k}.$$

Hence, $y_t$ is bounded for all $t \geq t_1$, and we denote by $C_y > 0$ any bound for $y_t$.

Next, we show that $y_t$ converges to zero as $t$ goes to infinity. For any $\epsilon > 0$, there is a sufficient large $t_\epsilon > t_1$ such that

1. $e^{-kt_\epsilon} C_y < \epsilon/2$.

2. $|\phi(t)| < \epsilon/2$ for all $t \geq t_\epsilon$.

Then for all $t_2 > 2t_\epsilon$, by integrating the inequality from $t_\epsilon$ to $t_2$, we have that

$$y_{t_2} \leq e^{-k(t_2-t_\epsilon)} y_{t_\epsilon} + \int_{t_\epsilon}^{t_2} e^{-k(t_2-t)} \phi(t) dt,$$

$$\leq e^{-k(t_\epsilon)} C_y + \epsilon/2 \int_{t_\epsilon}^{t_2} e^{-k(t_2-t)} dt,$$

$$\leq \epsilon/2 + \epsilon/2(1 - e^{-k(t_2-t_\epsilon)}),$$

$$\leq \epsilon.$$

This implies that $y_t$ converges to zero as $t$ goes to infinity. $\qquad\square$

**Attracting toward sets.** Let $\Omega$ be a closed set in $\mathbb{R}^d$. We want to examine the distance to $\Omega$ along the ODE trajectory $(x_\sigma)_{\sigma \in (\sigma_2, \sigma_1]}$ with some $\sigma_1 > \sigma_2$. Assume that the trajectory avoids the medial axis of $\Omega$ then the distance of $x_\sigma$ to $\Omega$ will decrease as $\sigma$ decreases if the trajectory direction $\frac{1}{\sigma} (m_\sigma(x_\sigma) - x_\sigma)$ forms an acute angle with the direction pointing toward $\Omega$, that is

$$\langle m_\sigma(x_\sigma) - x_\sigma, \operatorname{proj}_\Omega(x_\sigma) - x_\sigma \rangle > 0.$$

Notice that one has

$$\langle m_\sigma(x_\sigma) - x_\sigma, \operatorname{proj}_\Omega(x_\sigma) - x_\sigma \rangle = \langle m_\sigma(x_\sigma) - \operatorname{proj}_\Omega(x_\sigma), \operatorname{proj}_\Omega(x_\sigma) - x_\sigma \rangle + \|\operatorname{proj}_\Omega(x_\sigma) - x_\sigma\|^2. \tag{18}$$

Hence, whenever $x_\sigma \notin \Omega$, $\|\operatorname{proj}_\Omega(x_\sigma) - x_\sigma\| > 0$ and the acute angle condition will be satisfied if the term $\langle m_\sigma(x_\sigma) - \operatorname{proj}_\Omega(x_\sigma), \operatorname{proj}_\Omega(x_\sigma) - x_\sigma \rangle$ is not too negative. This intuition is formalized in the following theorem, which is the formal version of Theorem 3.1.

**Theorem C.7** (Attracting toward sets). *Let $(x_\sigma)_{\sigma \in (\sigma_2, \sigma_1]}$ be an ODE trajectory of Equation (11) starting from some $x_{\sigma_1}$. Assume that the trajectory avoids the medial axis of a closed $\Omega$ then we have the following results.*

1. *If $\langle m_\sigma(x_\sigma) - \operatorname{proj}_\Omega(x_\sigma), \operatorname{proj}_\Omega(x_\sigma) - x_\sigma \rangle \leq \zeta \|x_\sigma - \operatorname{proj}_\Omega(x_\sigma)\|^2$ for some $0 \leq \zeta < 1$ along the trajectory, then $d_\Omega(x_\sigma)$ decreases along the trajectory with rate:*

$$d_\Omega(x_\sigma) \leq \frac{\sigma^{1-\zeta}}{\sigma_1^{1-\zeta}} d_\Omega(x_{\sigma_1}),$$

   *In particular, if $\sigma_2 = 0$, then $d_\Omega(x_\sigma)$ is guaranteed to converge to zero as $\sigma \to 0$.*

2. *If $\sigma_2 = 0$ and $|\langle m_\sigma(x_\sigma) - \operatorname{proj}_\Omega(x_\sigma), \operatorname{proj}_\Omega(x_\sigma) - x_\sigma \rangle| \leq \phi(\sigma)$ for some function $\phi(\sigma)$ along the trajectory with $\lim_{\sigma \to 0} \phi(\sigma) = 0$, then*

$$\lim_{\sigma \to 0} d_\Omega(x_\sigma) = 0.$$

*Remark* C.8. In fact, when considering the parameter $\lambda = -\log(\sigma)$ and the trajectory $z_\lambda := x_{\sigma(\lambda)}$, we obtain the following convergence rate for Item 2 in the above theorem:

$$d_\Omega(z_\lambda) \leq e^{-(\lambda - \lambda_1)} d_\Omega(z_{\lambda_1}) + e^{-\lambda} \sqrt{\int_{\lambda_1}^{\lambda} 2e^{2t} \phi(e^{-t}) dt}.$$

*Proof of Theorem C.7 and Remark C.8.* We consider the change of variable $\lambda = -\log(\sigma)$. We let $z_\lambda := x_\sigma(\lambda)$ for all $\lambda \in [\lambda_1 := \lambda(\sigma_1), \lambda_2 := \lambda(\sigma_2)]$. Then, $(z_\lambda)_{\lambda \in [\lambda_1, \lambda_2]}$ satisfies the nice ODE as in Equation (16). By assumption we have that $z_{\lambda_1}$ is outside the convex set $\Omega$. For any $\lambda \in [\lambda_1, \lambda_2]$, by Corollary B.15 we have that

$$\frac{d}{d\lambda} d_\Omega^2(z_\lambda) = -2\langle z_\lambda - \operatorname{proj}_\Omega(z_\lambda), z_\lambda - m_\lambda(z_\lambda) \rangle$$
$$= -2 \left( \langle z_\lambda - \operatorname{proj}_\Omega(z_\lambda), z_\lambda - \operatorname{proj}_\Omega(z_\lambda) \rangle + \langle z_\lambda - \operatorname{proj}_\Omega(z_\lambda), \operatorname{proj}_\Omega(z_\lambda) - m_\lambda(z_\lambda) \rangle \right).$$

For the first item, we have that $\frac{d}{d\lambda} d_\Omega^2(z_\lambda) \leq -2d_\Omega^2(z_\lambda)$. Multiplying with the exponential integrator $e^{2(1-\zeta)\lambda}$, we have that

$$\frac{d}{d\lambda} (e^{2(1-\zeta)\lambda} d_\Omega^2(z_\lambda)) \leq 0.$$

Then for any $\lambda \in [\lambda_1, \lambda_2]$, we have that

$$d_\Omega^2(z_\lambda) \leq e^{-2(1-\zeta)(\lambda - \lambda_1)} d_\Omega^2(z_{\lambda_1}).$$

Using the change of variable, we have that

$$d_\Omega(z_\lambda) \leq \frac{\sigma^{1-\zeta}}{\sigma_1^{1-\zeta}} d_\Omega(z_{\lambda_1}).$$

For the second item, we have that

$$\frac{d}{d\lambda}d_\Omega^2(z_\lambda) \leq -2d_\Omega^2(z_\lambda) + 2\phi(e^{-\lambda}).$$

Then we can apply Item 2 of Lemma C.6 to obtain $\lim_{\lambda\to\infty} d_\Omega^2(z_\lambda) = 0$. This implies that $\lim_{\sigma\to 0} d_\Omega^2(x_\sigma) = 0$.

We then apply Item 1 of Lemma C.6 to obtain that for any $\lambda \geq \lambda_1$:

$$d_\Omega^2(z_\lambda) \leq e^{-2(\lambda-\lambda_1)}d_\Omega^2(z_{\lambda_1}) + 2\int_{\lambda_1}^\lambda e^{-2(\lambda-t)}\phi(e^{-t})dt.$$

Using the fact that $\sqrt{a+b} \leq \sqrt{a} + \sqrt{b}$ for any $a, b \geq 0$, we have that

$$d_\Omega(z_\lambda) \leq e^{-(\lambda-\lambda_1)}d_\Omega(z_{\lambda_1}) + e^{-\lambda}\sqrt{\int_{\lambda_1}^\lambda 2e^{2t}\phi(e^{-t})dt}.$$

This proves Remark C.8 □

**Absorbing by sets.** Now, we prove the absorbing Theorem 3.2 below.

*Proof of Theorem 3.2.* We will utilize the parameter $\lambda$ for this proof with which we consider the ODE trajectory $(z_\lambda := x_{\sigma(\lambda)})_{\lambda\in[\lambda_1,\lambda_2]}$ of Equation (16) with $\lambda_1 = -\log(\sigma_1)$ and $\lambda_2 = -\log(\sigma_2)$.

For Item 1, we will show that the trajectory $z_\lambda$ must stay inside $B_r(\Omega)$ for all $\lambda \in [\lambda_1, \lambda_2]$ whenever $z_{\lambda_1} \in B_r(\Omega)$. Suppose otherwise then there exists some first time $\lambda_o > \lambda_1$ such that $z_{\lambda_o} \in \partial B_r(\Omega)$. By the assumption that $\overline{B_r(\Omega)} \cap \Sigma_\Omega = \emptyset$, we can apply Lemma B.14 to obtain the derivative of the squared distance function along the trajectory for any $\lambda \in [\lambda_1, \lambda_o]$:

$$\frac{d}{d\lambda}d_{B_r(\Omega)}^2(z_\lambda) = -2\langle m_\lambda(z_\lambda) - z_\lambda, \mathrm{proj}_\Omega(z_\lambda) - z_\lambda\rangle.$$

This implies that $\frac{d}{d\lambda}d_{B_r(\Omega)}^2(z_{\lambda_o}) < 0$. By the continuity of the derivative above, we have that $\frac{d}{d\lambda}d_{B_r(\Omega)}^2(z_\lambda) < 0$ for all $\lambda \in [\lambda_o - \epsilon, \lambda_o]$ for some $\epsilon > 0$. Note that $d_\Omega^2(z_{\lambda_o}) = r^2$ and $d_\Omega^2(z_{\lambda_o-\epsilon}) < r^2$. Then by the mean value theorem, there exists some $\lambda_i \in (\lambda_o - \epsilon, \lambda_o)$ such that $\frac{d}{d\lambda}d_{B_r(\Omega)}^2(z_{\lambda_i}) > 0$. This leads to a contradiction and hence the trajectory $z_\lambda$ must stay inside $B_r(\Omega)$ for all $\lambda \in [\lambda_1, \lambda_2]$, which concludes the proof.

The second part of the theorem follows straightforwardly from the fact that $\Omega$ is a closed set. If a trajectory starts from $\Omega$ but leaves $\Omega$ at some point, it must also leave a neighborhood of $\Omega$, which would contradict the given assumption. This completes the proof. □

We now describe how we will use the above absorbing property for convex sets which will be used often in results in Sections 4 and 5.2, and how its generalization will be used to analyze the convergence of the flow matching ODEs in Section 5.1.

For a convex set $K$, its the medial axis $\Sigma_K$ is empty (see e.g. Item 1 of Proposition B.5) and if we assume that the denoiser $m_\sigma$ lies in $K$ for any $x \in \partial B_r(K)$ then any neighborhood of $K$ will be absorbing for the ODE trajectory. We also obtain an stronger result regarding when the set $K$ itself is absorbing.

**Proposition C.9** (Absorbing of convex sets). *Let $K$ be a closed convex set in $\mathbb{R}^d$. Let $(x_\sigma)_{\sigma\in(\sigma_2,\sigma_1]}$ be an ODE trajectory of Equation* (11). *Then, we have the following results.*

1. *For any $r > 0$, if $m_\sigma(x) \in K$ for any $x \in \partial B_r(K)$ and any $\sigma \in (\sigma_2, \sigma_1]$, then $B_r(K)$ is absorbing for $(x_\sigma)_{\sigma\in(\sigma_2,\sigma_1]}$.*

2. *If the interior $K^\circ$ of $K$ is not empty and $m_\sigma(x) \in K^\circ$ for any $x \in \partial K$ and any $\sigma \in (\sigma_2, \sigma_1]$, then $K$ is absorbing for $(x_\sigma)_{\sigma\in(\sigma_2,\sigma_1]}$.*

*Proof of Proposition C.9.* For the first item, for any $x \in \partial B_r(K)$, since $m_\sigma(x)$ lies in the convex set $K$, we can apply Item 2 of Corollary B.9 to conclude

$$\langle m_\sigma(x) - \text{proj}_K(x), \text{proj}_K(x) - x \rangle \geq 0.$$

Then there is

$$\langle m_\sigma(x) - x, \text{proj}_K(x) - x \rangle = \langle m_\sigma(x) - \text{proj}_K(x), \text{proj}_K(x) - x \rangle + \|\text{proj}_K(x) - x\|^2 \geq r^2 > 0.$$

We then apply Theorem 3.2 to conclude that $B_r(K)$ is absorbing for $(x_\sigma)_{\sigma \in (\sigma_2, \sigma_1]}$.

For the second item, the distance function $d_K^2(x)$ does not distinguish between the interior and the boundary of $K$ and we utilize the supporting hyperplane function instead. Assume the trajectory $(x_\sigma)_{t \in (\sigma_2, \sigma_1]}$ leaves $K$ at some first time $\sigma_o \in (\sigma_2, \sigma_1)$. Then, in particular, $x_\sigma \in K$ for all $\sigma \in [\sigma_o, \sigma_1]$ and $x_{\sigma_o} \in \partial K$. In terms of the parameter $\lambda = -\log(\sigma)$, we have $z_\lambda \in [\lambda_1 := -\log(\sigma_1), \lambda_o := -\log(\sigma_o)]$ and $z_{\lambda_o} \in \partial K$. Since $K$ is a closed convex set, there exist a supporting hyperplane $H$ at $z_{\lambda_o}$ such that $H(z_{\lambda_o}) = 0$ and $H(y) \geq 0$ for all $y \in K$ (see Proposition B.4). In particular, we can write $H(y) = \langle y - z_{\lambda_o}, n \rangle$, where $n$ is the unit normal vector of the hyperplane. Since $z_\lambda \in K$ for all $\lambda \in [\lambda_o, \lambda_1]$ and $H(z_{\lambda_o}) \leq H(z_\lambda)$ for all $\lambda \in [\lambda_o, \lambda_1]$, we must have that $\frac{dH(z_\lambda)}{d\lambda}\big|_{\lambda = \lambda_o^-} \leq 0$. Therefore, we have that

$$\frac{dH(z_\lambda)}{d\lambda}\Big|_{\lambda = \lambda_o^-} = \left\langle \frac{dz_\lambda}{d\lambda}\big|_{\lambda = \lambda_o}, n \right\rangle = \langle m_{\lambda_o}(z_{\lambda_o}) - z_{\lambda_o}, n \rangle \leq 0.$$

Since $m_{\lambda_o}(z_{\lambda_o})$ lies in the interior of $K$, we must have that $\langle m_{\lambda_o}(z_{\lambda_o}) - z_{\lambda_o}, n \rangle > 0$. This leads to a contradiction and hence the trajectory $z_\lambda$ must stay inside $K$ for all $\lambda \in [\lambda_1, \lambda_2)$. $\square$

*Remark* C.10. When $\Omega$ is not convex, a typical way to show the acute angle condition is by requiring $\|m_\sigma(x) - \text{proj}_\Omega(x)\|$ to be small enough on $\partial B_r(\Omega)$ for all $\sigma \in (\sigma_2, \sigma_1]$. This can be seen by the following computation:

$$\langle m_\sigma(x_\sigma) - x_\sigma, \text{proj}_\Omega(x_\sigma) - x_\sigma \rangle = \langle m_\sigma(x_\sigma) - \text{proj}_\Omega(x_\sigma), \text{proj}_\Omega(x_\sigma) - x_\sigma \rangle + \|\text{proj}_\Omega(x_\sigma) - x_\sigma\|^2 \tag{19}$$

$$\geq -\|m_\sigma(x_\sigma) - \text{proj}_\Omega(x_\sigma)\|\|\text{proj}_\Omega(x_\sigma) - x_\sigma\| + \|\text{proj}_\Omega(x_\sigma) - x_\sigma\|^2. \tag{20}$$

Furthermore, once the absorbing property is established, it guarantees that $d_\Omega(x_\sigma) = \|\text{proj}_\Omega(x_\sigma) - x_\sigma\|$ to be bounded for a trajectory in consideration. In this case, the condition in Item 2 of the attracting theorem Theorem C.7 can be derived from the decay of $\|m_\sigma(x_\sigma) - \text{proj}_\Omega(x_\sigma)\|$ as

$$|\langle m_\sigma(x_\sigma) - \text{proj}_\Omega(x_\sigma), \text{proj}_\Omega(x_\sigma) - x_\sigma \rangle| \leq \|m_\sigma(x_\sigma) - \text{proj}_\Omega(x_\sigma)\|\|\text{proj}_\Omega(x_\sigma) - x_\sigma\|.$$

These type of arguments will be utilize in Section 4.3 and Section 5.2 when discussing the ODE dynamics of the flow matching ODEs.

When $\Omega$ is unbounded, e.g the support of a general distribution, it would require much more assumptions for us to control the term $\|m_\sigma(x_\sigma) - \text{proj}_\Omega(x_\sigma)\|$ uniformly on the boundary. As one way to circumvent this issue, we consider the intersection of $B_r(\Omega)$ with a bounded set and establish the absorbing property of the bounded subset. This is formalized in the following result.

**Theorem C.11** (Absorbing of data support). *Fix any small $0 < \delta < \tau_\Omega/4$ and any $0 < \zeta < 1$. Assume that there exists a constant $\sigma_\Omega > 0$ such that for any $R > \frac{1}{2}\tau_\Omega$ and for any $z \in B_R(0) \cap B_{\tau_\Omega/2}(\Omega)$, one has that*

$$\|m_\sigma(z) - \text{proj}_\Omega(z)\| \leq C_{\zeta, \tau, R} \cdot \sigma^\zeta, \text{ for all } 0 < \sigma < \sigma_\Omega.$$

*where $C_{\zeta, \tau, R}$ is a constant depending only on $\zeta$ and $\tau$ and $R$.*

*Then, there exists $\sigma_\delta \leq \sigma_\Omega$ dependent on $\delta, \zeta$ and $C_{\zeta, \tau, R}$ satisfying the following property for any $R > 2\delta$: The trajectory $(x_\sigma)_{\sigma \in (0, \sigma_\delta]}$ starting at any initial point $x_{\sigma_\delta} \in B_R(0) \cap B_\delta(\Omega)$ of the ODE in Equation (11) will be absorbed in a slightly larger space: for any $\sigma \leq \sigma_\delta$: $x_\sigma \in B_{2R}(0) \cap B_{2\delta}(\Omega)$.*

Note that this absorbing result is slightly different from Theorem 3.2, where the neighborhood of the data support $\Omega$ must be enlarged from $\delta$ to $2\delta$ (and $R$ to $2R$) to guarantee the absorbing property. This subtle difference arises from the additional treatment in the proof to account for the bounded ball $B_R(0)$ in the above theorem.

*Proof of Theorem C.11.* Similarly as in the proofs of other theorems in this section, we consider the change of variable $\lambda = -\log(\sigma)$ and $z_\lambda := x_\sigma(\lambda)$.

We let $C := C_{\zeta,\tau,2R}$ and define (one can see how the definition is motivated from the following proof)

$$\lambda_\delta := \max\left\{-\frac{1}{\zeta}\cdot\min\left\{\log\left(\frac{\delta\zeta}{2C}\right), 2\log\left(\frac{\zeta}{8}\sqrt{\frac{(2-\zeta)\delta}{C}}\right), \log\left(\frac{\delta(2-\zeta)}{4C}\right)\right\}, \Lambda\right\}.$$

Then, $\sigma_\delta := e^{-\lambda_\delta}$.

By continuity of the ODE path, there exists a maximal interval $I = [\lambda_\delta, \Lambda_\delta]$ so that for any $\lambda \in I$, $z_\lambda \in \overline{B_{2R}(0)} \cap \overline{B_{2\delta}(\Omega)}$ where the overline indicates the closure of the underlying set. Now, we show that $\Lambda_\delta$ must be infinity by showing that otherwise $z_{\Lambda_\delta}$ lies in the interior, i.e., $z_{\Lambda_\delta} \in B_{2R}(0) \cap B_{2\delta}(\Omega)$ and hence the trajectory will be able to extend within $B_{2R}(0) \cap B_{2\delta}(\Omega)$ to $\Lambda_\delta + \epsilon$ for some small $\epsilon > 0$. In this way, we can extend the interval $I$ to $[\lambda_\delta, \Lambda_\delta + \epsilon]$ which contradicts the maximality of $I$.

Now, assume that $\Lambda_\delta < \infty$. Then, by Corollary B.15, we have that for any $\lambda \in I$,

$$\frac{d(d_\Omega^2(z_\lambda))}{d\lambda} = -2\langle z_\lambda - \text{proj}_\Omega(z_\lambda), z_\lambda - m_\lambda(z_\lambda)\rangle$$

$$= -2\left(\langle z_\lambda - \text{proj}_\Omega(z_\lambda), z_\lambda - \text{proj}_\Omega(z_\lambda)\rangle + \langle z_\lambda - \text{proj}_\Omega(z_\lambda), \text{proj}_\Omega(z_\lambda) - m_\lambda(z_\lambda)\rangle\right)$$

$$\leq -2d_\Omega^2(z_\lambda) + 2d_\Omega(z_\lambda)\|m_\lambda(z_\lambda) - \text{proj}_\Omega(z_\lambda)\|.$$

By our assumption on $z_\lambda$, we have that $d_\Omega(z_\lambda) < 2\delta$ and for any $\lambda > \Lambda$, $\|m_\lambda(z_\lambda) - \text{proj}_\Omega(z_\lambda)\| \leq C \cdot e^{-\zeta\lambda}$. Then, by Remark C.8, we have that

$$d_\Omega(z_\lambda) \leq e^{-(\lambda-\lambda_\delta)}d_\Omega(z_{\lambda_\delta}) + e^{-\lambda}\sqrt{\int_{\lambda_\delta}^\lambda 4C\delta e^{2t}e^{-\zeta t}dt}$$

$$\leq e^{-(\lambda-\lambda_\delta)}d_\Omega(z_{\lambda_\delta}) + \underbrace{\sqrt{\frac{4\delta C}{2-\zeta}\left(e^{-\zeta\lambda} - e^{(2-\zeta)\lambda_\delta - 2\lambda}\right)}}_{\leq\sqrt{\frac{4\delta C}{2-\zeta}e^{-\zeta\lambda}}\leq\delta} < 2\delta,$$

where the inequality under the brace bracket follows from the definition of $\lambda_\delta$. Hence, $z_{\Lambda_\delta} \in B_{2\delta}(\Omega)$.

Now we examine $\|z_\lambda\|$ along the integral path. We have that

$$\|z_s - z_{\lambda_\delta}\| \leq \int_{\lambda_\delta}^s \|m_\lambda(z_\lambda) - z_\lambda\|d\lambda$$

$$\leq \int_{\lambda_\delta}^s \|m_\lambda(z_\lambda) - \text{proj}_\Omega(z_\lambda)\| + \|\text{proj}_\Omega(z_\lambda) - z_\lambda\|d\lambda$$

$$\leq \int_{\lambda_\delta}^s Ce^{-\zeta\lambda} + d_\Omega(z_\lambda)d\lambda.$$

Now, for the integral of $d_\Omega(z_\lambda)$, we have that

$$\int_{\lambda_\delta}^s d_\Omega(z_\lambda)d\lambda \leq \int_{\lambda_\delta}^s e^{-(\lambda-\lambda_\delta)}d_\Omega(z_{\lambda_\delta}) + \sqrt{\frac{4\delta C}{2-\zeta}\left(e^{-\zeta\lambda} - e^{(2-\zeta)\lambda_\delta - 2\lambda}\right)}d\lambda$$

$$\leq \int_{\lambda_\delta}^s e^{-(\lambda-\lambda_\delta)}\delta + \sqrt{\frac{4\delta C}{2-\zeta}e^{-\zeta\lambda}}d\lambda$$

$$= \delta(1 - e^{-s+\lambda_\delta}) + \sqrt{\frac{4\delta C}{2-\zeta}}\cdot\frac{2}{\zeta}\left(e^{-\zeta\lambda_\delta/2} - e^{-\zeta s/2}\right).$$

Therefore,

$$\|z_s - z_{\lambda_\delta}\| \leq \underbrace{\frac{C}{\zeta}(-e^{-\zeta s} + e^{-\zeta\lambda_\delta})}_{\leq \frac{C}{\zeta}e^{-\zeta\lambda_\delta} \leq \delta/2} + \delta(1 - e^{-s+\lambda_\delta}) + \underbrace{\sqrt{\frac{4\delta C}{2-\zeta} \cdot \frac{2}{\zeta}\left(e^{-\zeta\lambda_\delta/2} - e^{-\zeta s/2}\right)}}_{\leq \sqrt{\frac{4\delta C}{2-\zeta} \cdot \frac{2}{\zeta}}e^{-\zeta\lambda_\delta/2} \leq \delta/2} \leq 2\delta \tag{21}$$

where we used the definition of $\lambda_\delta$ again to control all the exponential terms. This implies that $\|z_s\| \leq R + 2\delta < 2R$ for all $s \in [\lambda_\delta, \Lambda_\delta]$ (recall that we assumed that $R > 2\delta$) and hence $z_{\Lambda_\delta} \in B_{2R}(0)$. This concludes the proof. $\qquad\square$

### C.3.1. ABSORBING AND ATTRACTING TO THE CONVEX HULL OF THE DATA SUPPORT

We prove Proposition 3.4 below.

*Proof of Proposition 3.4.* The key observation comes from that, the posterior distribution $p(\cdot|\boldsymbol{X}_{\boldsymbol{\sigma}} = x)$ is also supported on supp$(p)$ and hence its expectation, the denoiser $m_\sigma(x)$ must lie in the convex hull conv(supp$(p)$).

For the first part, the first Item in Proposition C.9 applies to conv(supp$(p)$) and hence any neighborhood of conv(supp$(p)$) is absorbing $(x_\sigma)_{\sigma \in (\sqrt{\sigma_{\text{init}}(\Omega,\zeta,R_0)^2},\sigma_1]}$. We then obtain the desired result by Item 2 in Theorem 3.2.

For the second part, we use Item 2 of Corollary B.9 to obtain

$$\left\langle m_\sigma(x) - \text{proj}_{\text{conv}(\text{supp}(p))}(x), \text{proj}_{\text{conv}(\text{supp}(p))}(x) - x \right\rangle \leq 0$$

for all $x \in \mathbb{R}^d$ and all $\sigma$. Then we can apply Item 1 of the meta attracting result Theorem C.7 with $\zeta = 0$ to conclude the proof. $\qquad\square$

The above propositions requires the data distribution to have a bounded support. We now extend the above results to a more general setting where the data distribution is $p = p_b * \mathcal{N}(0, \delta^2 I)$, i.e., the convolution of a bounded support distribution $p_b$ and a Gaussian distribution. This is done by noting that the ODE trajectory with data distribution $p$ can be derived from the ODE trajectory with data distribution $p_b$ by early stopping the trajectory. We formalize this in the following lemma.

**Lemma C.12.** *Let $p_b$ be a distribution with bounded support and let $p := p_b * \mathcal{N}(0, \delta^2 I)$ for any $\delta \geq 0$. Let $(y_{\sigma_b})_{\sigma_b \in (0,\infty)}$ be an ODE trajectory of Equation (7) with data distribution $p_b$. We define $(x_\sigma := y_{\sigma_b = \sqrt{\sigma^2+\delta^2}})_{\sigma \in (0,\infty)}$. Then, $(x_\sigma)$ is an ODE trajectory of Equation (7) with data distribution $p$.*

*Proof of Lemma C.12.* Let $q_{\sigma_b} := p_b * \mathcal{N}(0, \sigma_b^2 I)$ be the probability path with data distribution $p_b$. Then, we have that $y_{\sigma_b}$ satisfies

$$\frac{d\,y_{\sigma_b}}{d\sigma_b} = -\sigma_b \nabla \log q_{\sigma_b}(y_{\sigma_b}),$$

With the change of variable $\sigma_b = \sqrt{\sigma^2 + \delta^2}$, we have that

$$\begin{aligned}
\frac{d\,y_{\sigma_b}}{d\sigma} &= -\frac{d\,\sigma_b}{d\,\sigma}\sigma_b \nabla \log q_{\sigma_b}(y_{\sigma_b}), \\
&= -\frac{\sigma}{\sigma_b}\sigma_b \nabla \log q_{\sigma_b}(y_{\sigma_b}), \\
&= -\sigma \nabla \log q_{\sigma_b}(y_{\sigma_b}),
\end{aligned}$$

One has that

$$\begin{aligned}
q_{\sigma_b}(y_{\sigma_b}) &= \frac{1}{(2\pi\sigma_b^2)^{d/2}} \int \exp\left(-\frac{\|y_{\sigma_b} - x\|^2}{2\sigma_b^2}\right) p_b(dx) \\
&= \frac{1}{(2\pi(\sigma^2 + \delta^2))^{d/2}} \int \exp\left(-\frac{\|x_\sigma - x\|^2}{2(\delta^2 + \sigma^2)}\right) p_b(dx) \\
&= q_\sigma(x_\sigma),
\end{aligned}$$

where $q_\sigma(x)$ denotes the density of $q_\sigma := p * \mathcal{N}(0, \sigma^2 I)$. Therefore, the trajectory $x_\sigma := y_{\sigma_b}$ satisfies

$$\frac{d\,x_\sigma}{d\sigma} = -\sigma \nabla \log q_\sigma(x_\sigma). \qquad \square$$

**Corollary C.13.** *Given the data distribution $p$ defined as in Lemma C.12, for any $\sigma_1 > 0$, let $x_\sigma$ be an ODE trajectory of Equation (11) from $\sigma = \sigma_1$ to $\sigma = 0$. Then, we have that*

1. *If $x_{\sigma_1} \in \mathrm{conv}(\mathrm{supp}(p_b))$, then $x_\sigma \in \mathrm{conv}(\mathrm{supp}(p_b))$ for any $\sigma \in (0, \sigma_1]$;*

2. *If $x_{\sigma_1} \notin \mathrm{conv}(\mathrm{supp}(p_b))$, then $x_\sigma$ moves toward $\mathrm{conv}(\mathrm{supp}(p_b))$ with the following decay guarantee:*

$$d_{\mathrm{conv}(\mathrm{supp}(p_b))}(x_\sigma) \leq \frac{\sqrt{\sigma^2 + \delta^2}}{\sqrt{\sigma_1^2 + \delta^2}} d_{\mathrm{conv}(\mathrm{supp}(p_b))}(x_{\sigma_1}),$$

*for any $\sigma \in (0, \sigma_1]$.*

*Proof of Corollary C.13.* This is a direct consequence of Proposition 3.4 and Lemma C.12 $\qquad \square$

## D. Denoiser Analysis: Concentration and Convergence of the Posterior Distribution

In this section, we analyze the denoiser through the concentration and convergence of the posterior distribution, i.e., Theorem D.1, and its variants under different assumptions. Our proof strategy is similar to that used in Stanczuk et al. (2024, Theorem 4.1), where the integral under consideration is split into two parts to analyze the concentration. However, our results are more general as we do not require the data distribution to lie on a manifold or have points be sufficiently near the data support. We additionally provide rate control of the convergence result which is crucial for the terminal behavior analysis in Section 5.1.

**Theorem D.1.** *Let $\Omega := \mathrm{supp}(p)$. Assume that $p$ has a finite 2-moment. For all $x \in \mathbb{R}^d \backslash \Sigma_\Omega$, we have that*

$$\lim_{\sigma \to 0} d_{\mathrm{W},2}\left(p(\cdot|\boldsymbol{X}_\sigma = x), \delta_{\mathrm{proj}_\Omega(x)}\right) = 0.$$

*Proof of Theorem D.1.* We let $\Phi_x := d_{\Sigma_\Omega}(x)/2 > 0$ and $\Delta_x := d_\Omega(x) \geq 0$. We let $x_\Omega := \mathrm{proj}_\Omega(x)$. According to Lemma B.10, for any $\epsilon > 0$, if we define the radius

$$r_{x,\epsilon} := \sqrt{\Delta_x^2 + \epsilon^2 \left(1 - \frac{\Delta_x}{\Delta_x + \Phi_x}\right)},$$

then we have the following inclusion

$$B_{r_{x,\epsilon}}(x) \cap \Omega \subseteq B_\epsilon(x_\Omega) \cap \Omega.$$

For the other direction, we have that

$$B_{r_{x,\epsilon}^*}(x_\Omega) \cap \Omega \subseteq B_{r_{x,\epsilon}}(x) \cap \Omega$$

where

$$r_{x,\epsilon}^* := r_{x,\epsilon} - \Delta_x = \frac{\Phi_x}{(\Phi_x + \Delta_x)(r_{x,\epsilon} + \Delta_x)} \epsilon^2.$$

Here $r_{x,\epsilon}^*$ satisfies that as $d_\Omega(x) \to 0$, $r_{x,\epsilon}^* \to \epsilon$. So $B_{r_{x,\epsilon}^*}(x_\Omega)$ can be thought of as an approximation of $B_\epsilon(x_\Omega)$.

Then, we estimate the two terms below separately:

$$d_{\mathrm{W},2}(p(\cdot|\boldsymbol{X}_\sigma = x), \delta_{x_\Omega})^2 = \int_\Omega \|x_0 - x_\Omega\|^2 \frac{\exp\left(-\frac{\|x-x_0\|^2}{2\sigma^2}\right)}{\int_\Omega \exp\left(-\frac{\|x-x_0'\|^2}{2\sigma^2}\right) p(dx_0')} p(dx_0)$$

$$= \underbrace{\int_{B_{r_{x,\epsilon}}(x)\cap\Omega} \|x_0 - x_\Omega\|^2 \frac{\exp\left(-\frac{\|x-x_0\|^2}{2\sigma^2}\right)}{\int_\Omega \exp\left(-\frac{\|x-x_0'\|^2}{2\sigma^2}\right) p(dx_0')} p(dx_0)}_{I_1}$$

$$+ \underbrace{\int_{B_{r_{x,\epsilon}}(x)^c(y)\cap\Omega} \|x_0 - x_\Omega\|^2 \frac{\exp\left(-\frac{\|x-x_0\|^2}{2\sigma^2}\right)}{\int_\Omega \exp\left(-\frac{\|x-x_0'\|^2}{2\sigma^2}\right) p(dx_0')} p(dx_0)}_{I_2}$$

For $I_1$, we have the following estimate:

$$I_1 \leq \int_{B_\epsilon(x_\Omega)\cap\Omega} \|x_0 - x_\Omega\|^2 \frac{\exp\left(-\frac{\|x-x_0\|^2}{2\sigma^2}\right)}{\int_\Omega \exp\left(-\frac{\|x-x_0'\|^2}{2\sigma^2}\right) p(dx_0')} p(dx_0) \leq \epsilon^2$$

For the second term $I_2$, we have the following estimate:

$$I_2 \leq \int_{B_{r_{x,\epsilon}}^c(y)\cap\Omega} (2\|x_0\|^2 + 2\|x_\Omega\|^2) \frac{\exp\left(-\frac{\|x-x_0\|^2}{2\sigma^2}\right)}{\int_\Omega \exp\left(-\frac{\|x-x_0'\|^2}{2\sigma^2}\right) p(dx_0')} p(dx_0)$$

$$= \int_{B_{r_{x,\epsilon}}^c(y)\cap\Omega} \frac{2\|x_0\|^2 + 2\|x_\Omega\|^2}{\int_\Omega \exp\left(\frac{\|x-x_0\|^2}{2\sigma^2} - \frac{\|x-x_0'\|^2}{2\sigma^2}\right) p(dx_0')} p(dx_0)$$

$$\leq \int_{B_{r_{x,\epsilon}\cap\Omega}^c(y)\cap\Omega} \frac{2\|x_0\|^2 + 2\|x_\Omega\|^2}{\int_{B_{r_{x,\epsilon/\sqrt{2}}}(y)\cap\Omega} \exp\left(\frac{\|x-x_0\|^2}{2\sigma^2} - \frac{\|x-x_0'\|^2}{2\sigma^2}\right) p(dx_0')} p(dx_0)$$

$$\leq \int_{B_{r_{x,\epsilon}\cap\Omega}^c(y)\cap\Omega} \frac{2\|x_0\|^2 + 2\|x_\Omega\|^2}{\int_{B_{r_{x,\epsilon/\sqrt{2}}}(y)\cap\Omega} \exp\left(\frac{\epsilon^2}{4\sigma^2}\left(1 - \frac{\Delta_x}{\Delta_x + \Phi_x}\right)\right) p(dx_0')} p(dx_0)$$

$$= \exp\left(-\frac{\epsilon^2}{4\sigma^2}\left(1 - \frac{\Delta_x}{\Delta_x + \Phi_x}\right)\right) \frac{2\|x_\Omega\|^2 + 2\mathsf{M}_2(p)}{p\left(B_{r_{x,\epsilon/\sqrt{2}}}(y)\cap\Omega\right)}$$

$$\leq \exp\left(-\frac{\epsilon^2}{4\sigma^2}\left(1 - \frac{\Delta_x}{\Delta_x + \Phi_x}\right)\right) \frac{2\|x_\Omega\|^2 + 2\mathsf{M}_2(p)}{p\left(B_{r_{x,\epsilon/\sqrt{2}}^*}(x_\Omega)\right)}.$$

Since $\Omega$ is the support of $p$, we have that $p\left(B_{r_{x,\epsilon/\sqrt{2}}^*}(x_\Omega)\right) > 0$. Hence, for any $\epsilon > 0$, we have

$$I_1 + I_2 \leq \epsilon^2 + \frac{2\|x_\Omega\|^2 + 2\mathsf{M}_2(p)}{p\left(B_{r_{x,\epsilon/\sqrt{2}}^*}(x_\Omega)\right)} \exp\left(-\frac{\epsilon^2}{4\sigma^2}\left(1 - \frac{\Delta_x}{\Delta_x + \Phi_x}\right)\right).$$

By letting $\sigma \to 0$, we have that $I_1 + I_2 \leq \epsilon$. Therefore, we have that

$$\lim_{\sigma \to 0} d_{\mathrm{W},2}(p(\cdot|\boldsymbol{X}_\sigma = x), \delta_{x_\Omega}) = 0.$$

$\square$

As a direct consequence of the convergence of the posterior distribution, we have the convergence of the denoiser stated in Theorem 5.1. Here we provide a simple proof.

*Proof of Theorem 5.1.* This is a direct consequence of the following property of the Wasserstein distance where the case $s = 1$ was proved in Rubner et al. (1998) and the other cases follow from the fact that $d_{W,s}$ is increasing with respect to $s$ (Villani, 2009, Remark 6.6).

**Lemma D.2.** *For any probability measures $\nu_1, \nu_2$ on $\mathbb{R}^d$ and any $1 \leq s \leq \infty$, we have that*

$$d_{W,s}(\nu_1, \nu_2) \geq \|\text{mean}(\nu_1) - \text{mean}(\nu_2)\|.$$

$\square$

When we turn back to the parameter $t$, we have the following corollary. The proof turns out to be rather technical instead of being a direct consequence of Theorem 5.1. The main difficulty lies in the scaling $\alpha_t$ within the exponential term. This is another example that the parameter $\sigma$ is more convenient for theoretical analysis.

**Corollary D.3.** *Let $\Omega := \text{supp}(p)$. Assume that $p$ has a finite 2-moment. For all $x \in \mathbb{R}^d \backslash \Sigma_\Omega$, we let $x_\Omega := \text{proj}_\Omega(x)$. Then, we have that*

$$\lim_{t \to 1} m_t(x) = \text{proj}_\Omega(x).$$

*Proof of Corollary D.3.* In the proof of Theorem D.1, we end up with

$$d_{W,2}(p(\cdot|\boldsymbol{X}_\sigma = x), \delta_{x_\Omega})^2 \leq \epsilon^2 + \exp\left(-\frac{\epsilon^2}{4\sigma^2}\left(1 - \frac{\Delta_x}{\Delta_x + \Phi_x}\right)\right) \frac{2\|x_\Omega\|^2 + 2M_2(p)}{p\left(B_{r^*_{x,\epsilon/\sqrt{2}}}(x_\Omega)\right)}$$

for any arbitrarily chosen small $\epsilon$.

By Lemma B.12, there exists a small $\xi > 0$ and a constant $C_\xi > 0$ such that for any $\|z - x\| < \xi$, one has $\|z_\Omega - x_\Omega\| < C_\xi \|z - x\|$. As both $\Delta_x$ and $\Phi_x$ are continuous in a local neighborhood of $x$, there exists a small $\xi_1 > 0$ such that for any $z$ with $\|z - x\| < \xi_1$, one has

- $r^*_{z,\epsilon/\sqrt{2}} > \frac{1}{2} r^*_{x,\epsilon/\sqrt{2}}$;

- $1 - \frac{\Delta_x}{\Delta_x + \Phi_x} > \frac{1}{2}\left(1 - \frac{\Delta_z}{\Delta_z + \delta_z}\right)$;

- $\|z_\Omega\| \leq 2\|x_\Omega\|$.

Now, take $\Xi := \min\left\{\xi, \xi_1, \frac{1}{2C_\xi} r^*_{z,\epsilon/\sqrt{2}}\right\}$. Then, it is easy to check that for any $z$ such that $\|z - x\| < \Xi$, one has

$$B_{\frac{1}{4} r^*_{x,\epsilon/\sqrt{2}}}(x_\Omega) \subset B_{r^*_{z,\epsilon/\sqrt{2}}}(z_\Omega).$$

This implies that for any $z$ such that $\|z - x\| < \Xi$, one has

$$\|m_\sigma(z) - z_\Omega\|^2 \leq d_{W,2}(p(\cdot|\boldsymbol{X}_\sigma = z), \delta_{z_\Omega})^2 \leq \epsilon^2 + \exp\left(-\frac{\epsilon^2}{8\sigma^2}\left(1 - \frac{\Delta_x}{\Delta_x + \Phi_x}\right)\right) \frac{4\|x_\Omega\|^2 + 2M_2(p)}{p\left(B_{r^*_{x,\frac{1}{4}\epsilon/\sqrt{2}}}(x_\Omega)\right)}.$$

Hence, when $\sigma$ is small enough (dependent on $\epsilon$), we have that $\|m_\sigma(z) - z_\Omega\| \leq 2\epsilon$ for any $z$ such that $\|z - x\| < \Xi$.

Now, we have that

$$\|m_t(x) - x_\Omega\| \leq \|m_{\sigma_t}(x/\alpha_t) - \text{proj}_\Omega(x/\alpha_t)\| + \|\text{proj}_\Omega(x/\alpha_t) - \text{proj}_\Omega(x)\|.$$

As $\alpha_t \to 1$, there exists $t_0$ such that when $t > t_0$, one has that $z = x/\alpha_t \in B_\Xi(x)$ and $\|\text{proj}_\Omega(x/\alpha_t) - \text{proj}_\Omega(x)\| \leq \epsilon$. By the analysis above, we can enlarge $t$ so that $\sigma_t$ is small enough so that $\|m_{\sigma_t}(z) - z_\Omega\| \leq 2\epsilon$. Therefore, for any $t > t_0$, we have that $\|m_t(x) - x_\Omega\| \leq 3\epsilon$. Since $\epsilon$ is arbitrary, we have that $\lim_{t \to 1} m_t(x) = x_\Omega$. $\square$

### D.1. Convergence Rate

We establish the following convergence rate for the posterior distribution when more assumptions are made on $p$.

**Theorem D.4.** *Assume that the reach $\tau = \tau_\Omega > 0$ is positive. Consider any $x \in \mathbb{R}^d$. We assume that $d_\Omega(x) < \frac{1}{2}\tau$ and that there exists $g \geq 0$ such that there exist constants $C, c > 0$ so that for any small radius $0 < r < c$, one has $p(B_r(x_\Omega)) \geq Cr^k$. Then, for any $0 < \zeta < 1$ we have the following convergence rate for any $0 < \sigma < c^{1/\zeta}$:*

$$d_{\mathrm{W},2}\left(p(\cdot|\boldsymbol{X}_\sigma = x), \delta_{x_\Omega}\right) \leq \sqrt{\sigma^{2\zeta} + \frac{10^k(2\mathsf{M}_2(p) + \|x_\Omega\|^2)\max\{\tau^k, \sigma^{k\zeta}\}}{C\sigma^{2k\zeta}}\exp\left(-\frac{1}{8}\sigma^{2(\zeta-1)}\right)} \leq C_{\zeta,\tau}\sigma^\zeta,$$

*where $C_{\zeta,\tau}$ is a constant depending only on $\zeta$ and $\tau_\Omega$.*

*Proof of Theorem D.4.* Recall from the proof of Theorem D.1 that for any $\epsilon > 0$, we have that

$$I_1 + I_2 \leq \epsilon^2 + \frac{2\|x_\Omega\|^2 + 2\mathsf{M}_2(p)}{p\left(B_{r_{x,\epsilon/\sqrt{2}}^*}(x_\Omega)\right)}\exp\left(-\frac{\epsilon^2}{4\sigma^2}\left(1 - \frac{\Delta_x}{\Delta_x + \Phi_x}\right)\right).$$

Then, we have that $2\Phi_x \geq \tau_\Omega - \Delta_x$. So

$$r_{x,\epsilon}^* = \frac{\Phi_x}{(\Phi_x + \Delta_x)(r_{x,\epsilon} + \Delta_x)}\epsilon^2 \tag{22}$$

$$\geq \frac{\tau_\Omega - \Delta_x}{2\tau_\Omega} \cdot \frac{\epsilon^2}{\Delta_x + \sqrt{\Delta_y^2 + \epsilon^2}} \tag{23}$$

$$\geq \frac{1}{4} \cdot \min\left\{\frac{\epsilon}{(\sqrt{2}+1)\Delta_x}, \frac{1}{\sqrt{2}+1}\right\}\epsilon \tag{24}$$

$$\geq \frac{\epsilon}{10}\min\left\{\frac{\epsilon}{\tau_\Omega}, 1\right\}. \tag{25}$$

On the other hand, we have that

$$r_{x,\epsilon}^* \leq \frac{\Phi_x}{(\Phi_x + \Delta_x)\epsilon\sqrt{\frac{\Phi_x}{\Delta_x + \Phi_x}}}\epsilon^2$$

$$= \epsilon\sqrt{\frac{\Phi_x}{\Delta_x + \Phi_x}} \leq \epsilon.$$

Therefore, if one let $\epsilon = \sigma^\zeta$, when $\sigma < c^{1/\zeta}$ we have that $r_{x,\epsilon}^* < c$ and then

$$p\left(B_{r_{x,\epsilon/\sqrt{2}}^*}(x_\Omega)\right) \geq C_\Omega \cdot \left(\frac{\epsilon}{10}\min\left\{\frac{\epsilon}{\tau_\Omega}, 1\right\}\right)^k$$

$$= \frac{C_\Omega\sigma^{k\zeta}}{10^k}\min\left\{\frac{\sigma^{k\zeta}}{\tau_\Omega^k}, 1\right\}.$$

Notice that $1 - \frac{\Delta_x}{\Delta_x + \Phi_x} \geq \frac{1}{2}$, we have that eventually

$$I_1 + I_2 \leq \sigma^{2\zeta} + \frac{2\mathsf{M}_2(\rho) + 2\|x_\Omega\|^2}{\frac{C_\Omega\sigma^{k\zeta}}{10^k}\min\left\{\frac{\sigma^{k\zeta}}{\tau_\Omega^k}, 1\right\}}\exp\left(-\frac{1}{8\sigma^{2(1-\zeta)}}\right)$$

$$= \sigma^{2\zeta} + \frac{10^k(2\mathsf{M}_2(\rho) + 2\|x_\Omega\|^2)\max\{\tau_\Omega^k, \sigma^{k\zeta}\}}{C_\Omega\sigma^{2k\zeta}}\exp\left(-\frac{1}{8}\sigma^{2(\zeta-1)}\right).$$

The rightmost inequality in the theorem follows from the fact that the right most summand in the above equation has an exponential decay which is way faster than any polynomial decay. $\square$

Now, we improve the convergence rate when $p$ is either supported on a submanifold or a discrete set.

**Convergence rates for the manifold case.** We first consider the case when $p$ is supported on a submanifold $M \subset \mathbb{R}^d$. Note that this does not exclude the case when $M = \mathbb{R}^d$. Under some mild conditions, we have the following convergence rate for the posterior distribution.

**Theorem D.5.** *Let $M \subset \mathbb{R}^d$ be a $m$ dimensional closed submanifold (without self-intersection) with a positive reach $\tau_M$. Assume that $p(dx) = \rho(x)\mathrm{vol}_M(dx)$ has a smooth non-vanishing density $\rho : M \to \mathbb{R}$. For any $x \in \mathbb{R}^d$ and $\sigma \in (0, \infty)$, we let $x_M := \mathrm{proj}_M(x)$. If $x \in \mathbb{R}^d$ satisfies that $d_M(x) < \frac{1}{2}\tau_M$, then we have that*

$$d_{\mathrm{W},2}(p(\cdot|\boldsymbol{X}_\sigma = x), \delta_{x_M}) = \sqrt{m}\sigma + O(\sigma^2).$$

The proof is very lengthy and we postpone it to the end of this section.

Note that the leading-order term in the convergence rate, $\sqrt{m}\sigma$, depends solely on the intrinsic dimension $m$ of the submanifold $M$. The higher-order term $O(\sigma^2)$ depends on finer geometric properties of $M$, such as curvature and reach. A detailed characterization of these higher-order contributions would be an interesting direction for future work.

As a direct consequence of the convergence of the posterior distribution, we have the following convergence of the denoiser.

**Corollary D.6.** *Under the same assumptions as in Theorem D.5, for any $x \in \mathbb{R}^d$ satisfying $d_M(x) < \frac{1}{2}\tau_M$, we have that*

$$\|m_\sigma(x) - x_M\| = O(\sigma).$$

**Convergence rates for the discrete case.** Let the data distribution $p = \sum_{i=1}^N a_i\,\delta_{x_i}$ be a general discrete distribution with $x_1, \ldots, x_N \in \mathbb{R}^d$ and $a_1, \ldots, a_N > 0$. We use $\Omega = \{x_1, \ldots, x_N\}$ to denote the support of $p$. We study the concentration and convergence of the posterior measure $p(\cdot|\boldsymbol{X}_\sigma = x)$ for each $x \in \mathbb{R}^d$, including those $x$ on $\Sigma_\Omega$, the medial axis of $\Omega$. To this end, we introduce the following notations.

For each point $x \in \mathbb{R}^d$, we denote the set of distance values from $x$ to each point in $\Omega$ as follows:

$$\mathrm{DV}_\Omega(x) := \{\|x - x_i\| : x_i \in \Omega\}. \tag{26}$$

We use $d_\Omega(x; i)$ to denote the $i$-th smallest distance value in $\mathrm{DV}_\Omega(x)$. A useful geometric notion will be the gap between the squares of the two smallest distances which we denote by

$$\Delta_\Omega(x) := d_\Omega^2(x; 2) - d_\Omega^2(x; 1). \tag{27}$$

We further let

$$\mathrm{NN}_\Omega(x) := \left\{ x_i \in \Omega : \|x - x_i\| = d_\Omega(x; 1) = \min_{x_j \in \Omega} \|x - x_j\| \right\}.$$

We use the notation $\widehat{p}_{\mathrm{NN}(x)}$ to denote the normalized measure restricted to the points in $\Omega$ that are closest to $x$:

$$\widehat{p}_{\mathrm{NN}(x)} := \frac{1}{\sum_{x_i \in \mathrm{NN}_\Omega(x)} a_i} \sum_{x_i \in \mathrm{NN}_\Omega(x)} a_i\,\delta_{x_i}. \tag{28}$$

Whenever $x$ is not on $\Sigma_\Omega$, we have $\mathrm{NN}_\Omega(x) = \{\mathrm{proj}_\Omega(x)\}$ and $\widehat{p}_{\mathrm{NN}(x)} = \delta_{\mathrm{proj}_\Omega(x)}$. With the above notation, we have the following convergence result for the posterior measure $p(\cdot|\boldsymbol{X}_\sigma = x)$ as $\sigma \to 0$.

**Theorem D.7.** *Let $p = \sum_{i=1}^N a_i\,\delta_{x_i}$ be a discrete distribution. For any $x \in \mathbb{R}^d$, we have the following convergence of the posterior measure $p(\cdot|\boldsymbol{X}_\sigma = x)$ toward $\widehat{p}_{\mathrm{NN}(x)}$:*

$$d_{\mathrm{W},2}\left(p(\cdot|\boldsymbol{X}_\sigma = x), \widehat{p}_{\mathrm{NN}(x)}\right) \le \mathrm{diam}(\Omega)\sqrt{\frac{1 - p(\mathrm{NN}_\Omega(x))}{p(\mathrm{NN}_\Omega(x))}} \exp\left(-\frac{\Delta_\Omega(x)}{4\sigma^2}\right).$$

*Proof of Theorem D.7.* When $\Delta_\Omega(x) = 0$, $\mathrm{NN}_\Omega(x) = \Omega$ and $p(\cdot|\boldsymbol{X}_\sigma = x) = p$ and then the Wasserstein distance becomes 0 which proves the statement.

Now, we will focus on the case when $\Delta_\Omega(x) > 0$. The posterior measure $p(\cdot|\boldsymbol{X}_\sigma = x)$ is given by

$$p(\cdot|\boldsymbol{X}_\sigma = x) = \sum_{i=1}^{N} \frac{a_i \exp\left(-\frac{1}{2\sigma^2}\|x_i - x\|^2\right)}{\sum_{j=1}^{N} a_j \exp\left(-\frac{1}{2\sigma^2}\|x_j - x\|^2\right)} \delta_{x_i}.$$

For the ease of notation, we use $B_\sigma := \sum_{j=1}^{N} a_j \exp\left(-\frac{1}{2\sigma^2}\|x_j - x\|^2\right)$ to denote the normalization constant in $p(\cdot|\boldsymbol{X}_\sigma = x)$, $B_{\sigma,x_i} := a_i \exp\left(-\frac{1}{2\sigma^2}\|x_i - x\|^2\right)$ to denote the $i$-th term in $B_\sigma$. We use $A = p(\mathrm{NN}_\Omega(x))$ to denote the normalization constant in $\widehat{p}_{\mathrm{NN}(x)}$.

Observe that for any $x_i, x_j \in \mathrm{NN}_\Omega(x)$, one has

$$p(x_i|\boldsymbol{X}_\sigma = x)/p(x_j|\boldsymbol{X}_\sigma = x) = \widehat{p}_{\mathrm{NN}(x)}(x_i)/\widehat{p}_{\mathrm{NN}(x)}(x_j).$$

This motivates the following construction of a coupling $\mu$ between $p(\cdot|\boldsymbol{X}_\sigma = x)$ and $p_{\mathrm{NN}(y)}$:

$$\mu = \sum_{x_j \in \mathrm{NN}_\Omega(x)} \frac{B_{\sigma,x_j}}{B_\sigma} \cdot \sum_{x_i \in \mathrm{NN}_\Omega(x)} \frac{a_i}{A} \delta_{(x_i, x_i)} + \sum_{x_i \in \mathrm{NN}_\Omega(x), x_j \in \Omega \setminus \mathrm{NN}_\Omega(x)} \frac{B_{\lambda,x_j}}{B_\sigma} \frac{a_i}{A} \delta_{(x_i, x_j)}.$$

Then, we bound the Wasserstein distance between $p(\cdot|\boldsymbol{X}_\sigma = x)$ and $\widehat{p}_{\mathrm{NN}(x)}$ as follows:

$$\begin{aligned}
d_{\mathrm{W},2}(p(\cdot|\boldsymbol{X}_\sigma = x), \widehat{p}_{\mathrm{NN}(x)})^2 &\leq \int \|x - y\|^2 \mu(dx, dy), \\
&= \sum_{x_i \in \mathrm{NN}_\Omega(x), x_j \in \Omega \setminus \mathrm{NN}_\Omega(x)} \frac{B_{\lambda,x_j}}{B_\sigma} \frac{a_i}{A} \|x_i - x_j\|^2, \\
&\leq \sum_{x_i \in \mathrm{NN}_\Omega(x), x_j \in \Omega \setminus \mathrm{NN}_\Omega(x)} \frac{B_{\lambda,x_j}}{B_\sigma} \frac{a_i}{A} \mathrm{diam}(\Omega)^2, \\
&= \mathrm{diam}(\Omega)^2 \sum_{x_i \in \mathrm{NN}_\Omega(x), x_j \in \Omega \setminus \mathrm{NN}_\Omega(x)} \frac{a_j \exp\left(-\frac{1}{2\sigma^2}\|x_j - x\|^2\right)}{\sum_{j=1}^{N} a_j \exp\left(-\frac{1}{2\sigma^2}\|x_j - x\|^2\right)} \frac{a_i}{A}, \\
&\leq \mathrm{diam}(\Omega)^2 \sum_{x_i \in \mathrm{NN}_\Omega(x), x_j \in \Omega \setminus \mathrm{NN}_\Omega(x)} \frac{a_j \exp\left(-\frac{1}{2}e^{2\lambda}d_\Omega^2(x,2)\right)}{A \exp\left(-\frac{1}{2}e^{2\lambda}d_\Omega^2(x,1)\right)} \frac{a_i}{A}, \\
&= \mathrm{diam}(\Omega)^2 \frac{1-A}{A} \frac{\exp\left(-\frac{1}{2}e^{2\lambda}d_\Omega^2(x,2)\right)}{\exp\left(-\frac{1}{2}e^{2\lambda}d_\Omega^2(x,1)\right)}, \\
&= \mathrm{diam}(\Omega)^2 \frac{1-A}{A} \exp\left(-\frac{1}{2}e^{2\lambda}(d_\Omega^2(x,2) - d_\Omega^2(x,1))\right), \\
&\leq \mathrm{diam}(\Omega)^2 \frac{1-A}{A} \exp\left(-\frac{1}{2}e^{2\lambda}\Delta_\Omega(x)\right).
\end{aligned}$$

By taking the square roots on both sides, we conclude the proof. $\qquad\square$

As a corollary of Theorem D.7, we have the following convergence of the denoiser.

**Corollary D.8.** *Let $p = \sum_{i=1}^{N} a_i \delta_{x_i}$ be a discrete distribution. For any $x \in \mathbb{R}^d$, we have the following convergence of the denoiser $m_\sigma(x)$ toward the mean of $\widehat{p}_{\mathrm{NN}(x)}$:*

$$\|m_\sigma(x) - \mathrm{mean}(\widehat{p}_{\mathrm{NN}(x)})\| \leq \mathrm{diam}(\Omega)\sqrt{\frac{1 - p(\mathrm{NN}_\Omega(x))}{p(\mathrm{NN}_\Omega(x))}} \exp\left(-\frac{\Delta_\Omega(x)}{4\sigma^2}\right).$$

*In particular, when $x \notin \Sigma_\Omega$, assume that $x_i = \mathrm{proj}_\Omega(x)$ we have that*

$$\|m_\sigma(x) - x_i\| \leq \mathrm{diam}(\Omega)\sqrt{\frac{1 - a_i}{a_i}} \exp\left(-\frac{\Delta_\Omega(x)}{4\sigma^2}\right).$$

*Proof of Corollary D.8.* The result follows from Theorem D.7 and the stability of the mean operation under the Wasserstein distance Lemma D.2. □

Later in Section 5.2, we will utilize the above results to analyze memorization behavior.

**Proof of Theorem D.5** We still let $\Delta_x := d_M(x) < \frac{1}{2}\tau_M$. Then, we let $q_\sigma^x := p(\cdot|\boldsymbol{X}_\sigma = x)$ and hence,

$$q_\sigma^x(dx_1) := \frac{\exp\left(-\frac{\|x-x_1\|^2}{2\sigma^2}\right)\rho(x_1)\mathrm{vol}_M(dx_1)}{\int_M \exp\left(-\frac{\|x-x_1'\|^2}{2\sigma^2}\right)\rho(x_1')\mathrm{vol}_M(dx_1')}, \quad \text{for } x_1 \in M.$$

As $\sigma \to 0$, $q_\sigma^x$ is concentrated around $x_M = \mathrm{proj}_M(x)$. For any $r_0 > 0$, we have that

$$
\begin{aligned}
d_{\mathrm{W},2}(q_\sigma^x, \delta_{x_M})^2 &= \int_M \|x_1 - x_M\|^2 q_\sigma^x(dx_1) \\
&= \underbrace{\int_{B_{r_0}(x_M)} \|x_1 - x_M\|^2 q_\sigma^x(dx_1)}_{I_1} + \underbrace{\int_{M\backslash B_{r_0}(x_M)} \|x_1 - x_M\|^2 q_\sigma^x(dx_1)}_{I_2}.
\end{aligned}
$$

We first consider the term $I_2$.

$$I_2 \le \int_{M\backslash B_{r_0}(x_M)} (2\|x_1\|^2 + 2\|x_M\|^2)q_\sigma^x(dx_1).$$

As we have shown already in the proof of Theorem D.1, we have that $B_{r_{x,r_0}}(x) \cap M \subset B_{r_0}(x_M) \cap M$. So, we have that

$$I_2 \le \int_{M\backslash B_{r_{x,r_0}}(x)} (2\|x_1\|^2 + 2\|x_M\|^2)q_\sigma^x(dx_1) \tag{29}$$

$$\le \exp\left(-\frac{r_0^2}{4\sigma^2}\left(1 - \frac{\Delta_x}{\Delta_x + \Phi_x}\right)\right)\frac{2\mathsf{M}_2(p) + 2\|x_M\|^2}{p\left(B_{r_{x,r_0/\sqrt{2}}^*}(x_M)\right)} = O\left(\exp\left(-\frac{r_0^2}{8\sigma^2}\right)\right). \tag{30}$$

Now, we consider the term $I_1$. Since $M$ is a submanifold of $\mathbb{R}^d$, its tangent space $T_{x_M}M$ can be identified as a subspace of $\mathbb{R}^d$. We use $\iota : T_{x_M}M \to \mathbb{R}^d$ to denote the inclusion map. In particular, for any $u \in T_{x_M}M$, we have that $\langle x - x_M, \iota(u)\rangle = 0$. Let $\mathrm{Inj}(M)$ denote the injectivity radius of $M$. Note that, by Alexander & Bishop (2006, Corollary 1.4), $\mathrm{Inj}(M) \ge \tau_M/4$. So, the exponential map $\exp_{x_M} : T_{x_M}M \to M$ will be an diffeomorphism in a small ball $B_{\tau_M/4}^{T_{x_M}}(\mathbf{0}) \subset T_{x_M}M$. Furthermore, we have the following inclusion relation between geodesic balls and Euclidean balls.

**Lemma D.9.** *For any $0 < h \le \frac{3\tau_M}{25}$, we have that*

$$M \cap B_h(x_M) \subset \exp_{x_M}\left(B_{5h/3}^{T_{x_M}}(\mathbf{0})\right) \subset M \cap B_{5h/3}(x_M).$$

*Proof of Lemma D.9.* The proof of the left hand side follows from Lemma A.1 and Lemma A.2 (ii) of Aamari & Levrard (2019). Although Lemma A.2 (ii) stated compactness for the manifold $M$, this assumption is not needed in the proof. The right hand side follows from the fact that $\|x - y\| \le d_M(x,y)$ for any $x, y \in M$, where $d_M$ denotes the geodesic distance. □

Now, we fix some $r_0 \le 3\tau_M/25$. Then, there exists an open neighborhood $U_{r_0} \subset B_{5r_0/3}^{T_{x_M}}(\mathbf{0})$ around $\mathbf{0} \in T_{x_M}M$ so that we have the following diffeomorphism (which gives rise to the normal coordinates):

$$\exp_{x_M} : U_{r_0} \subseteq T_{x_M}M \to B_{r_0}(x_M) \cap M.$$

In particular, $x_M = \exp_{x_M}(\mathbf{0})$ and each $x_1 \in B_{r_0}(x_M) \cap M$ can be written as $x_1 = \exp_{x_M}(u)$ for some $u \in U_r$.

Let $\boldsymbol{II} : T_{x_M}M \times T_{x_M}M \to (T_{x_M}M)^\perp$ denote the second fundamental form of $M$ at $x_M$. Then, the exponential map $\exp_{x_M}$ in $U_{r_0}$ has the following Taylor expansion (Monera et al., 2014):

$$\exp_{x_M}(u) = x_M + \iota(u) + \frac{1}{2}\boldsymbol{II}(u, u) + O(\|u\|^3), \tag{31}$$

where $O(\|u\|^3) \le C(\boldsymbol{II}, \nabla\boldsymbol{II})\|u\|^3$ for some constant $C(\boldsymbol{II}, \nabla\boldsymbol{II}) > 0$ only dependent on $\boldsymbol{II}$ and its derivatives $\nabla\boldsymbol{II}$.

For any $u \in U_{r_0}$, we let $g(u)$ denote the metric tensor, then $g(\mathbf{0})$ is the identity matrix. The volume form is given by $\sqrt{\det(g(u))}du^1 \wedge \ldots \wedge du^k$ and it admits the following Taylor expansion around $\mathbf{0} \in T_{x_M}M$:

$$\sqrt{\det(g(u))} = 1 - \frac{1}{6}R_{ij}u^i u^j + O(\|u\|^3),$$

where $R_{ij}$ is the Ricci curvature tensor of $M$ at $x_M$, see e.g. Chow et al. (2023, Exercise 1.83).

We can write

$$\rho(u)\sqrt{\det(g(u))} = \rho(\mathbf{0}) + R_1(u), \quad |R_1(u)| \le C_1\|u\|, \tag{32}$$

where $C_1$ depends on Ricci curvature tensor $R_{ij}$ and the gradient $\nabla\rho(x_M)$.

Now, we define

$$f_\sigma(u) := \exp\left(-\frac{\|x - \exp_{x_M}(u)\|^2}{2\sigma^2}\right)\rho(u)\sqrt{\det(g(u))}.$$

Let $MB_{r_0} := B_{r_0}(x_M) \cap M$. Then, we have that

$$q_\sigma^x(du) = \frac{f_\sigma(u)du}{\int_{U_{r_0}} f_\sigma(u')du' + \int_{M\setminus MB_{r_0}} \exp\left(-\frac{\|x-x_1'\|^2}{2\sigma^2}\right)\rho(x_1')\mathrm{vol}_M(dx_1')}.$$

Using the same argument as the one used for controlling $I_2$ in the proof of Theorem D.1, we have

$$\frac{\int_{M\setminus MB_{r_0}} \exp\left(-\frac{\|x-x_1'\|^2}{2\sigma^2}\right)\rho(x_1')\mathrm{vol}_M(dx_1')}{\int_{MB_{r_0}} \exp\left(-\frac{\|x-x_1'\|^2}{2\sigma^2}\right)\rho(x_1')\mathrm{vol}_M(dx_1')} = O\left(\exp\left(-\frac{r_0^2}{8\sigma^2}\right)\right). \tag{33}$$

So,

$$\int_{MB_{r_0}} \|x_1 - x_M\|^2 q_\sigma^x(dx_1) = \frac{\int_{U_{r_0}} \|\exp_{x_M}(u) - x_M\|^2 f_\sigma(u)du}{\int_{U_{r_0}} f_\sigma(u')du'}\left(1 + O\left(\exp\left(-\frac{r_0^2}{8\sigma^2}\right)\right)\right). \tag{34}$$

Next, we derive a Taylor expansion of the squared distance from $x$ to $\exp_{x_M}(u)$ for $u$ around $\mathbf{0}$:

**Lemma D.10.** *Let $v := x - x_M$. Then, we have that*

$$\|x - \exp_{x_M}u\|^2 = \|v\|^2 + \|u\|^2 - \langle v, \boldsymbol{II}(u,u)\rangle + R_2(u) = \|v\|^2 + u^T(\boldsymbol{I} - \boldsymbol{II}_v)u + R_2(u)$$

*where $\boldsymbol{II}_v := (\langle v, \boldsymbol{II}_{ij}\rangle)_{i,j=1,\ldots,k}$ and $|R_2(u)| \le C_2\|u\|^3$ for some positive constant $C_2 = C_2(\boldsymbol{II}, \nabla\boldsymbol{II}) > 0$, and $\boldsymbol{I}$ is the identity matrix.*

*Proof of Lemma D.10.* We first note that

$$\|x - \exp_{x_M}u\|^2 = \|x - x_M\|^2 + \|x_M - \exp_{x_M}(u)\|^2 + 2\langle x - x_M, x_M - \exp_{x_M}(u)\rangle.$$

By equation (31), we have that

$$\begin{aligned}\|x - \exp_{x_M}u\|^2 &= \|v\|^2 + \|u\|^2 + \langle u, \boldsymbol{II}(u,u)\rangle - 2\langle v, \iota(u)\rangle - \langle v, \boldsymbol{II}(u,u)\rangle + O(\|u\|^3)\\ &= \|v\|^2 + \|u\|^2 - \langle v, \boldsymbol{II}(u,u)\rangle + O(\|u\|^3)\end{aligned}$$

where we used the fact that $v$ belongs the normal space $(T_{x_M}M)^\perp$ of $M$ at $x_M$ so that $\langle v, \iota(u)\rangle = 0$ and $\langle u, \boldsymbol{II}(u,u)\rangle = 0$. $\qquad\square$

By Lemma D.10 and Equation (34), we have that

$$\int_{U_{r_0}} \| \exp_{x_M}(u) - x_M \|^2 f_\sigma(u) du$$
$$= \exp\left(-\frac{\|v\|^2}{2\sigma^2}\right) \int_{U_{r_0}} (u^T(\boldsymbol{I} - \boldsymbol{II}_v)u + R_2(u)) \exp\left(-\frac{u^T(\boldsymbol{I} - \boldsymbol{II}_v)u + R_2(u)}{2\sigma^2}\right) (\rho(\boldsymbol{0}) + R_1(u)) du.$$

and

$$\int_{U_{r_0}} f_\sigma(u) du = \exp\left(-\frac{\|v\|^2}{2\sigma^2}\right) \int_{U_{r_0}} \exp\left(-\frac{u^T(\boldsymbol{I} - \boldsymbol{II}_v)u + R_2(u)}{2\sigma^2}\right) (\rho(\boldsymbol{0}) + R_1(u)) du.$$

We will establish the following claims:

*Claim* 1.

$$\int_{U_{r_0}} (u^T(\boldsymbol{I} - \boldsymbol{II}_v)u + R_2(u)) \exp\left(-\frac{u^T(\boldsymbol{I} - \boldsymbol{II}_v)u + R_2(u)}{2\sigma^2}\right) (\rho(\boldsymbol{0}) + R_1(u)) du$$
$$= \sigma^{m+2}\rho(\boldsymbol{0}) \int_{\mathbb{R}^m} z^T(\boldsymbol{I} - \boldsymbol{II}_v)z \exp\left(-\frac{z^T(\boldsymbol{I} - \boldsymbol{II}_v)z}{2}\right) dz + O(\sigma^{m+3}). \tag{35}$$

*Claim* 2.

$$\int_{U_{r_0}} \exp\left(-\frac{u^T(\boldsymbol{I} - \boldsymbol{II}_v)u + R_2(u)}{2\sigma^2}\right) (\rho(\boldsymbol{0}) + R_1(u)) du$$
$$= \sigma^m \rho(\boldsymbol{0}) \int_{\mathbb{R}^m} \exp\left(-\frac{z^T(\boldsymbol{I} - \boldsymbol{II}_v)z}{2}\right) dz + O(\sigma^{m+1}). \tag{36}$$

With the above two claims, there is

$$\int_{MB_{r_0}} \|x_1 - x_M\|^2 q_\sigma^x(dx_1) = \frac{\int_{U_{r_0}} \| \exp_{x_M}(u) - x_M \|^2 f_\sigma(u) du}{\int_{U_{r_0}} f_\sigma(u') du'} \left(1 + O\left(\exp\left(-\frac{r_0^2}{8\sigma^2}\right)\right)\right)$$

$$= \frac{\sigma^2 \int_{\mathbb{R}^m} z^T(\boldsymbol{I} - \boldsymbol{II}_v)z \exp\left(-\frac{z^T(\boldsymbol{I}-\boldsymbol{II}_v)z}{2}\right) dz + O(\sigma^3)}{\int_{\mathbb{R}^m} \exp\left(-\frac{z^T(\boldsymbol{I}-\boldsymbol{II}_v)z}{2}\right) dz + O(\sigma)} \left(1 + O\left(\exp\left(-\frac{r_0^2}{8\sigma^2}\right)\right)\right)$$

$$= \sigma^2 \left(\frac{\int_{\mathbb{R}^m} z^T(\boldsymbol{I} - \boldsymbol{II}_v)z \exp\left(-\frac{z^T(\boldsymbol{I}-\boldsymbol{II}_v)z}{2}\right) dz}{\int_{\mathbb{R}^m} \exp\left(-\frac{z^T(\boldsymbol{I}-\boldsymbol{II}_v)z}{2}\right) dz} + O(\sigma)\right) \left(1 + O\left(\exp\left(-\frac{r_0^2}{8\sigma^2}\right)\right)\right)$$

$$= m\sigma^2 + O(\sigma^3)$$

where in the last equality we used the fact that $\mathbb{E}[\boldsymbol{X}^T A \boldsymbol{X}] = \mathrm{tr}(A\Sigma)$ for a Gaussian random variable $\boldsymbol{X} \sim \mathcal{N}(0, \Sigma)$.

Therefore, we have that

$$d_{\mathrm{W},2}(q_\sigma^x, \delta_{x_M}) = \sqrt{m}\sigma + O(\sigma^2),$$

and concludes the proof.

Now we prove the two claims. For the first claim, note that by $\|v\| = d_M(x) < \tau_M/2$, we can use Berenfeld et al. (2022, Theorem 2.1) to conclude

$$\|\boldsymbol{II}_v\| \le \|v\| \cdot \max_{i,j} \|\boldsymbol{II}_{ij}\| < \frac{1}{2}\tau_M \cdot 1/\tau_M = \frac{1}{2}.$$

So the matrix $\boldsymbol{I} - \boldsymbol{II}_v$ is positive definite. We let $\lambda_{\min} > 0$ be the smallest eigenvalues of $\boldsymbol{I} - \boldsymbol{II}_v$. Note that $\lambda_{\min} \ge 1 - \|\boldsymbol{II}_v\| \ge \frac{1}{2}$. So, we can choose $r_0$ small enough at the beginning so that there exists some constant $C_3 > 0$ for all $u \in U_{r_0}$, we have that

$$u^T(\boldsymbol{I} - \boldsymbol{II}_v)u + R_2(u) \ge \lambda_{\min}\|u\|^2 - C_2\|u\|^3 > C_3\|u\|^2.$$

Consider the transformation $u = \sigma z$. Then, we let $\sigma' := 2r_0(C_2\sigma)^{\frac{1}{3}}$ which goes to 0 as $\sigma \to 0$ and for sufficiently small $\sigma$, we have that $\sigma' > \sigma$ hence there is

$$\frac{1}{\sigma'}U_{r_0} \subset \frac{1}{\sigma}U_{r_0}.$$

Then, we have that

$$\int_{U_{r_0}} u^T(\boldsymbol{I} - \boldsymbol{II}_v)u \exp\left(-\frac{u^T(\boldsymbol{I} - \boldsymbol{II}_v)u + R_2(u)}{2\sigma^2}\right)(\rho(\boldsymbol{0}) + R_1(u))du$$

$$= \sigma^{m+2}\int_{\frac{1}{\sigma}U_{r_0}} z^T(\boldsymbol{I} - \boldsymbol{II}_v)z \exp\left(-\frac{z^T(\boldsymbol{I} - \boldsymbol{II}_v)z}{2} - \frac{R_2(\sigma z)}{2\sigma^2}\right)(\rho(\boldsymbol{0}) + R_1(\sigma z))dz$$

$$= \sigma^{m+2}\underbrace{\int_{\frac{1}{\sigma'}U_{r_0}} z^T(\boldsymbol{I} - \boldsymbol{II}_v)z \exp\left(-\frac{z^T(\boldsymbol{I} - \boldsymbol{II}_v)z}{2} - \frac{R_2(\sigma z)}{2\sigma^2}\right)(\rho(\boldsymbol{0}) + R_1(\sigma z))dz}_{J_1}$$

$$+ \sigma^{m+2}\underbrace{\int_{\frac{1}{\sigma}U_{r_0}\setminus\frac{1}{\sigma'}U_{r_0}} z^T(\boldsymbol{I} - \boldsymbol{II}_v)z \exp\left(-\frac{z^T(\boldsymbol{I} - \boldsymbol{II}_v)z}{2} - \frac{R_2(\sigma z)}{2\sigma^2}\right)(\rho(\boldsymbol{0}) + R_1(\sigma z))dz}_{J_2}.$$

Now, for $J_1$, recall that $U_{r_0} \subset B_{5r_0/3}^{T_{x_M}M}(\boldsymbol{0})$. Hence, for any $z \in \frac{1}{\sigma'}U_{r_0}$, we have that $\|z\| \leq \frac{1}{\sigma'}5r_0/3 < \frac{1}{\sigma'}2r_0$ and hence

$$\frac{|R_2(\sigma z)|}{2\sigma^2} \leq \frac{C_2(\sigma\|z\|)^3}{2\sigma^2} < \frac{C_2\sigma^3(2r_0)^3}{2\sigma^2\cdot(2r_0)^3C_2\sigma} = \frac{1}{2}.$$

This implies that there is expansion $\exp\left(-\frac{R_2(\sigma z)}{2\sigma^2}\right) = 1 + O(\sigma\|z\|^3)$. Therefore, we have that

$$J_1 = \int_{\frac{1}{\sigma'}U_{r_0}} z^T(\boldsymbol{I} - \boldsymbol{II}_v)z \exp\left(-\frac{z^T(\boldsymbol{I} - \boldsymbol{II}_v)z}{2}\right)(1 + O(\sigma\|z\|^3))(\rho(\boldsymbol{0}) + O(\|\sigma z\|))dz$$

$$= \rho(\boldsymbol{0})\int_{\frac{1}{\sigma'}U_{r_0}} z^T(\boldsymbol{I} - \boldsymbol{II}_v)z \exp\left(-\frac{z^T(\boldsymbol{I} - \boldsymbol{II}_v)z}{2}\right)dz + O(\sigma)$$

$$= \rho(\boldsymbol{0})\int_{\mathbb{R}^m} z^T(\boldsymbol{I} - \boldsymbol{II}_v)z \exp\left(-\frac{z^T(\boldsymbol{I} - \boldsymbol{II}_v)z}{2}\right)dz + O(\sigma).$$

where in the last part we used the fact that the the decay rate of such an integral as $\sigma$ goes to 0 is exponential.

For $J_2$, we similarly have that

$$|J_2| \leq \int_{\frac{1}{\sigma}U_{r_0}\setminus\frac{1}{\sigma'}U_{r_0}} z^T(\boldsymbol{I} - \boldsymbol{II}_v)z \exp\left(-C_3\|z\|^2\right)(\rho(\boldsymbol{0}) + O(\|\sigma z\|))dz$$

$$= O(\sigma).$$

Therefore,

$$\int_{U_{r_0}} u^T(\boldsymbol{I} - \boldsymbol{II}_v)u \exp\left(-\frac{u^T(\boldsymbol{I} - \boldsymbol{II}_v)u + R_2(u)}{2\sigma^2}\right)(\rho(\boldsymbol{0}) + R_1(u))du$$

$$= \sigma^{m+2}\rho(\boldsymbol{0})\int_{\mathbb{R}^m} z^T(\boldsymbol{I} - \boldsymbol{II}_v)z \exp\left(-\frac{z^T(\boldsymbol{I} - \boldsymbol{II}_v)z}{2}\right)dz + O(\sigma^{m+3}). \tag{37}$$

Similarly, we have that

$$\int_{U_{r_0}} \exp\left(-\frac{u^T(\boldsymbol{I} - \boldsymbol{II}_v)u + R_2(u)}{2\sigma^2}\right)(\rho(\boldsymbol{0}) + R_1(u))du$$

$$= \sigma^m \rho(\boldsymbol{0}) \int_{\mathbb{R}^m} \exp\left(-\frac{z^T(\boldsymbol{I} - \boldsymbol{II}_v)z}{2}\right)dz + O(\sigma^{m+1}). \tag{38}$$

Additionally, by $|R_2(u)| \leq C_2\|u\|^3$, we have that

$$\int_{U_{r_0}} R_2(u) \exp\left(-\frac{u^T(\boldsymbol{I} - \boldsymbol{II}_v)u + R_2(u)}{2\sigma^2}\right)(\rho(\boldsymbol{0}) + R_1(u))du = O(\sigma^{m+3}). \tag{39}$$

Then Claim 1 follows from Equation (37) and Equation (39) and Claim 2 follows from Equation (38).

## E. Explicit Eamples of FM ODEs

The following lemma specifies the FM ODE solution when the data distribution is the standard Gaussian itself.

**Lemma E.1.** *Let the data distribution $p$ be a standard Gaussian distribution $\mathcal{N}(0, I)$ in $\mathbb{R}^d$. Then the FM ODE can be computed explicitly as*

$$\frac{dx_t}{dt} = \frac{\dot{\alpha}_t\alpha_t + \dot{\beta}_t\beta_t}{\alpha_t^2 + \beta_t^2} x_t.$$

*This implies that the following explicit formula for the ODE trajectory*

$$x_t = \sqrt{\alpha_t^2 + \beta_t^2}\, x_0$$

*for any initial point $x_0 \in \mathbb{R}^d$. In particular, the flow map $\Psi_t$ is a scaling and $\Psi_1$ is the identity map.*

*Proof of Lemma E.1.* The FM ODE is given by

$$\frac{dx_t}{dt} = \frac{\dot{\beta}_t}{\beta_t}x_t + \frac{\dot{\alpha}_t\beta_t - \alpha_t\dot{\beta}_t}{\beta_t}\int \frac{\exp\left(-\frac{\|x_t - \alpha_t x_1\|^2}{2\beta_t^2}\right)x_1}{\int \exp\left(-\frac{\|x_t - \alpha_t x_1'\|^2}{2\beta_t^2}\right)p(dx_1')}p(dx_1).$$

When plugging in the standard Gaussian density for $p$, we have the following observation:

$$\exp\left(-\frac{\|x_t - \alpha_t x_1\|^2}{2\beta_t^2}\right)\exp\left(-\frac{\|x_1\|^2}{2}\right) = \exp\left(-\frac{\|x_t - \alpha_t x_1\|^2}{\beta_t^2} - \frac{\beta_t^2\|x_1\|^2}{2\beta_t^2}\right)$$

$$= \exp\left(-\frac{1}{2\beta_t^2}\left\|\frac{\alpha_t}{\sqrt{\alpha_t^2 + \beta_t^2}}x_t - \sqrt{\alpha_t^2 + \beta_t^2}x_1\right\|^2\right)\exp\left(-\frac{1}{2(\alpha_t^2 + \beta_t^2)}\|x_t\|^2\right).$$

For brevity, we denote $B_t := \sqrt{\alpha_t^2 + \beta_t^2}$ and $C_t := \frac{\alpha_t}{\sqrt{\alpha_t^2 + \beta_t^2}}$. Then the ODE can be simplified as

$$\frac{dx_t}{dt} = \frac{\dot{\beta}_t}{\beta_t}x_t + \frac{\dot{\alpha}_t\beta_t - \alpha_t\dot{\beta}_t}{\beta_t}\int \frac{\exp\left(-\frac{\|C_t x_t - B_t x_1\|^2}{2\beta_t^2}\right)x_1 dx_1}{\int \exp\left(-\frac{\|C_t x_t - B_t x_1'\|^2}{2\beta_t^2}\right)dx_1'}.$$

$$= \frac{\dot{\beta}_t}{\beta_t}x_t + \frac{\dot{\alpha}_t\beta_t - \alpha_t\dot{\beta}_t}{\beta_t}\int \frac{\exp\left(-\frac{\|C_t/B_t x_t - x_1\|^2}{2\beta_t^2/B_t^2}\right)x_1 dx_1}{\int \exp\left(-\frac{\|C_t/B_t x_t - x_1'\|^2}{2\beta_t^2/B_t^2}\right)dx_1'}.$$

$$= \frac{\dot{\beta}_t}{\beta_t}x_t + \frac{\dot{\alpha}_t\beta_t - \alpha_t\dot{\beta}_t}{\beta_t}C_t/B_t x_t$$

$$= \frac{\dot{\beta}_t}{\beta_t}x_t + \frac{\dot{\alpha}_t\beta_t - \alpha_t\dot{\beta}_t}{\beta_t}\frac{\alpha_t}{\alpha_t^2 + \beta_t^2}x_t$$

$$= \frac{\dot{\alpha}_t\alpha_t + \dot{\beta}_t\beta_t}{\alpha_t^2 + \beta_t^2}x_t.$$

The rest of the proof follows from directly verifying that $x_t = \sqrt{\alpha_t^2 + \beta_t^2} x_0$ satisfies the ODE. $\qquad\square$

The following proposition shows that for a distribution lies in a linear subspace $\mathbb{R}^m \subset \mathbb{R}^d$, the FM ODE can be decomposed.

**Proposition E.2.** *For any $0 < m \le d$, consider the subspace $\mathbb{R}^m \subset \mathbb{R}^d$. Let $p$ be a probability measure on $\mathbb{R}^d$ supported on $\mathbb{R}^m$ and assume the FM ODE is well defined. We express any point $\boldsymbol{x} \in \mathbb{R}^d$ as $\boldsymbol{x} = (x, y)$, where $x \in \mathbb{R}^m$ and $y \in \mathbb{R}^{d-m}$. Then for any FM scheduling functions $\alpha_t, \beta_t$, we have for any initial point $\boldsymbol{x}_0 = (x_0, y_0) \in \mathbb{R}^d$, the FM ODE trajectory is given by $(\boldsymbol{x}_t = (x_t, \beta_t y_0))_{t \in [0,1]}$, where $(x_t)_{t \in [0,1]}$ is the trajectory of the FM ODE on $\mathbb{R}^m$ with initial point $x_0$, data distribution $p$ regarded as a probability measure on $\mathbb{R}^m$, and with scheduling functions $\alpha_t, \beta_t$.*

*Proof of Proposition E.2.* The FM ODE is given by

$$\frac{d\boldsymbol{x}_t}{dt} = \frac{\dot{\beta}_t}{\beta_t} \boldsymbol{x}_t + \frac{\dot{\alpha}_t \beta_t - \alpha_t \dot{\beta}_t}{\beta_t} \int \frac{\exp\left(-\frac{\|\boldsymbol{x}_t - \alpha_t \boldsymbol{x}_1\|^2}{2\beta_t^2}\right) \boldsymbol{x}_1}{\int \exp\left(-\frac{\|\boldsymbol{x}_t - \alpha_t \boldsymbol{x}_1'\|^2}{2\beta_t^2}\right) p(d\boldsymbol{x}_1')} p(d\boldsymbol{x}_1).$$

We have the following two observations:

1. For each $\boldsymbol{x}_1 \in \mathbb{R}^m$, there is $\|\boldsymbol{x}_t - \alpha_t \boldsymbol{x}_1\|^2 = \|x_t - \alpha_t x_1\|^2 + \|y_t\|^2$ where $\boldsymbol{x}_t = (x_t, y_t)$ and $\boldsymbol{x}_1 = (x_1, 0)$.

2. The denoiser $m_t$ always lies in the subspace $\mathbb{R}^m$ since the data distribution $p$ is supported on $\mathbb{R}^m$.

These two observations imply that the FM ODE can be decoupled into two ODEs:

$$\begin{cases} \dfrac{dx_t}{dt} = \dfrac{\dot{\beta}_t}{\beta_t} x_t + \dfrac{\dot{\alpha}_t \beta_t - \alpha_t \dot{\beta}_t}{\beta_t} \displaystyle\int \frac{\exp\left(-\frac{\|x_t - \alpha_t x_1\|^2}{2\beta_t^2}\right) x_1}{\int \exp\left(-\frac{\|x_t - \alpha_t x_1'\|^2}{2\beta_t^2}\right) p(dx_1')} p(dx_1), \\ \dfrac{dy_t}{dt} = \dfrac{\dot{\beta}_t}{\beta_t} y_t. \end{cases}$$

The first equation is the FM ODE on $\mathbb{R}^m$ with initial point $x_0$ and data distribution $p$ in $\mathbb{R}^m$. The second equation is the ODE for $y_t$ which is a linear ODE with solution $y_t = \beta_t y_0$. $\qquad\square$

# F. Proofs in Section 2

*Proof of Proposition 2.1.* For any $x \in \mathbb{R}^d$ and $t \in (0, 1]$, since $\alpha_t > 0$, we have that the map $A_t$ is well-defined. Furthermore, we have that

$$(A_t)_\# p_t = (A_t)_\# \left( \int \mathcal{N}(\cdot | \alpha_t x_0, \beta_t^2 I) p(dx_0) \right)$$

$$= \int \mathcal{N}(\cdot | x_0, (\beta_t/\alpha_t)^2 I) p(dx_0)$$

$$= \int \mathcal{N}(\cdot | x_0, \sigma_t^2 I) p(dx_0) = q_{\sigma_t}.$$

$\qquad\square$

*Proof of Proposition 2.2.* We first write down the density of $q_\sigma$ explicitly as follows:

$$q_\sigma(x) = \frac{1}{(2\pi\sigma^2)^{d/2}} \int \exp\left(-\frac{\|x - x_1\|^2}{2\sigma^2}\right) p(dx).$$

Then, we have that

$$\nabla \log q_\sigma(x) = -\frac{1}{\sigma^2} \left( x - \frac{\int y \exp\left(-\frac{\|x-y\|^2}{2\sigma^2}\right) p(dy)}{\int \exp\left(-\frac{\|x-y'\|^2}{2\sigma^2}\right) p(dy')} \right).$$

Here, the existence of the gradient easily follows from the dominated convergence theorem.

Note that $\frac{dt}{d\sigma} = \frac{\alpha_t^2}{\dot{\beta}_t \alpha_t - \beta_t \dot{\alpha}_t}$. Therefore with the reparametrization $x_\sigma := x_t/\alpha_t$, we have that

$$
\begin{aligned}
\frac{dx_\sigma}{d\sigma} &= \frac{dx_\sigma}{dt}\frac{dt}{d\sigma} \\
&= \frac{1}{\alpha_t^2}\left(\alpha_t\frac{dx_t}{dt} - \dot{\alpha}_t x_t\right)\frac{dt}{d\sigma} \\
&= \frac{\alpha_t}{\beta_t}\left(\frac{x_t}{\alpha_t} - \int \frac{\exp\left(-\frac{\|x_t - \alpha_t y\|^2}{2\beta_t^2}\right)y}{\int \exp\left(-\frac{\|x_t - \alpha_t y'\|^2}{2\beta_t^2}\right)p(dy')}p(dy)\right) \\
&= -\sigma\nabla\log q_\sigma(x_\sigma).
\end{aligned}
$$

$\square$

## G. Proofs in Section 3

We provide in this section a convenient pointer to the proofs in Section 3. The formal version of Theorem 3.1 is Theorem C.7 in Appendix C.3. Similarly, the proofs of Theorem 3.2 and Proposition 3.4 are also provided in Appendix C.3.

## H. Proofs in Section 4

In this section, we provide the deferred proofs in Section 4. Note that the proofs of Theorem 4.1 and Proposition 4.2 require more preparation and are organized in a subsection at the end of this section; see Appendix H.1 and Appendix H.2.

*Proof of Proposition 4.3.* For simplicity, we use $\eta_\sigma := p(\cdot|X_\sigma = x)$ to denote the posterior measure at $x$. We restrict $\eta_\sigma$ on $S$ to obtain the following probability measure

$$
\eta_\sigma^S := \frac{1}{\eta_\sigma(S)}\eta_\sigma|_S.
$$

It is easy to see that $\mathbb{E}(\eta_\sigma^S)$ lies in the convex hull of $S$. We can then bound the distance from the denoiser $m_\sigma(x)$ to the convex hull of $S$ by the Wasserstein distance between $\eta_\sigma$ and $\eta_\sigma^S$.

Similarly to the proof of Theorem D.7, we construct the following coupling $\mu$ between $\eta_\sigma$ and $\eta_\sigma^S$:

$$
\mu = \Delta_\#(\eta_\sigma|_S) + \eta_\sigma^S \otimes (\eta_\sigma|_{(\Omega\backslash S)}),
$$

where $\Delta : \mathbb{R}^d \to \mathbb{R}^d \times \mathbb{R}^d$ is the diagonal map sending $x$ to $(x, x)$.

We then have the following estimate:

$$
\begin{aligned}
d_{\mathrm{W},2}^2\left(\eta_\sigma, \eta_\sigma^S\right) &\le \iint \|y_1 - y_2\|^2 \frac{\eta_\sigma|_S}{\eta_\sigma(S)}(dy_1)\eta_\sigma|_{(\Omega\backslash S)}(dy_2), \\
&\le \eta_\sigma((\Omega\backslash S))\,(\mathrm{diam}(\Omega))^2 \\
&= (\mathrm{diam}(\Omega))^2 \frac{\int_{(\Omega\backslash S)}\exp\left(-\frac{1}{2\sigma^2}\|x - x_1\|^2\right)p(dx_1)}{\int_{S\cup(\Omega\backslash S)}\exp\left(-\frac{1}{2\sigma^2}\|x - x_1'\|^2\right)p(dx_1')}, \\
&\le (\mathrm{diam}(\Omega))^2 \frac{\int_{(\Omega\backslash S)}\exp\left(-\frac{1}{2\sigma^2}\|x - x_1\|^2\right)p(dx_1)}{\int_{S}\exp\left(-\frac{1}{2\sigma^2}\|x - x_1\|^2\right)p(dx_1)},
\end{aligned}
$$

By the assumption that $d_{\mathrm{conv}(S)}(x) \le D/2 - \epsilon$, we have that for all $x_1 \in S$

$$
\|x - x_1\| \le D/2 - \epsilon + D = 3D/2 - \epsilon,
$$

and for all $x_1 \in \Omega \backslash S$

$$\|x - x_1\| \geq 2D - (D/2 - \epsilon) = 3D/2 + \epsilon.$$

We then have that

$$
\begin{aligned}
d_{\mathrm{W},2}\left(\eta_\sigma, \eta_\sigma^S\right)^2 &\leq (\mathrm{diam}(\Omega))^2 \frac{\int_{(\Omega \backslash S)} \exp\left(-\frac{1}{2\sigma^2}(3D/2 + \epsilon)^2\right) p(dx_1)}{\int_S \exp\left(-\frac{1}{2\sigma^2}(3D/2 - \epsilon)^2\right) p(dx_1)}, \\
&= (\mathrm{diam}(\Omega))^2 \frac{1 - a_S}{a_S} \exp\left(-\frac{1}{2\sigma^2}((3D/2 + \epsilon)^2 - (3D/2 - \epsilon)^2)\right), \\
&\leq (\mathrm{diam}(\Omega))^2 \frac{1 - a_S}{a_S} \exp\left(-\frac{3D\epsilon}{\sigma^2}\right).
\end{aligned}
$$

Then by applying Lemma D.2 and the fact that $\mathbb{E}(\eta_\sigma^S) \in \mathrm{conv}(S)$, we have that

$$d_{\mathrm{conv}(S)}(m_\sigma(x)) \leq \mathrm{diam}(\Omega) \sqrt{\frac{1 - a_S}{a_S}} \exp\left(-\frac{3D\epsilon}{2\sigma^2}\right).$$

$\square$

*Proof of Proposition 4.4.* Note that, the set $B_{D/2-\epsilon}(\mathrm{conv}(S))$ is convex itself. For the first part, according to Item 2 of Proposition C.9, it suffices to show that for all $x \in \partial \overline{B_{D/2-\epsilon}(\mathrm{conv}(S))}$, the denoiser $m_\sigma(x)$ lies in $B_{D/2-\epsilon}(\mathrm{conv}(S))$ for all $\sigma < \sigma_0(S, \epsilon)$.

By Proposition 4.3, we have that

$$d_{\mathrm{conv}(S)}(m_\sigma(x)) \leq \mathrm{diam}(\mathrm{supp}(p)) \sqrt{\frac{1 - a_S}{a_S}} \exp\left(-\frac{3D\epsilon}{2\sigma^2}\right).$$

Then, it suffices to show that

$$\mathrm{diam}(\mathrm{supp}(p)) \sqrt{\frac{1 - a_S}{a_S}} \exp\left(-\frac{3D\epsilon}{2\sigma^2}\right) < D/2 - \epsilon.$$

This is equivalent to

$$\exp\left(-\frac{3D\epsilon}{2\sigma^2}\right) < C_\epsilon^S,$$

where $C_\epsilon^S = \frac{D/2 - \epsilon}{\mathrm{diam}(\mathrm{supp}(p))\sqrt{\frac{1 - a_S}{a_S}}}$. Then, when $C_\epsilon^S \geq 1$, the above inequality holds for all $\sigma < \infty$ and when $C_\epsilon^S < 1$, it is straightforward to verify the above inequality holds for all $0 < \sigma < \sigma_0(S, \epsilon) := \left(\frac{-3D\epsilon}{2\log(C_\epsilon^S)}\right)^{\frac{1}{2}}$. This concludes the proof of the first part.

For the second part, we use $\lambda = -\log \sigma$ as the reparametrization and the corresponding trajectory $z_\lambda := x_{\sigma(\lambda)}$. We want to show that the distance $d_{\mathrm{conv}(S)}(z_\lambda)$ goes to zero as $\lambda$ goes to infinity. By taking the derivative of $d_{\mathrm{conv}(S)}^2(z_\lambda)$, and use the notation

$$v := \mathbb{E}\left[X | Z_\lambda = z_\lambda \text{ and } X \in S\right] \text{ where } Z_\lambda \sim q_{\sigma(\lambda)},$$

we have

$$
\begin{aligned}
\frac{d\, d_{\mathrm{conv}(S)}^2(z_\lambda)}{d\lambda} &= -2\langle z_\lambda - \mathrm{proj}_{\mathrm{conv}(S)}(z_\lambda), z_\lambda - m_\lambda(z_\lambda)\rangle, \\
&= -2\langle z_\lambda - \mathrm{proj}_{\mathrm{conv}(S)}(z_\lambda), z_\lambda - \mathrm{proj}_{\mathrm{conv}(S)}(z_\lambda)\rangle \\
&\quad - 2\langle z_\lambda - \mathrm{proj}_{\mathrm{conv}(S)}(z_\lambda), \mathrm{proj}_{\mathrm{conv}(S)}(z_\lambda) - v\rangle \\
&\quad - 2\langle z_\lambda - \mathrm{proj}_{\mathrm{conv}(S)}(z_\lambda), v - m_\lambda(z_\lambda)\rangle, \\
&\leq -2d_{\mathrm{conv}(S)}^2(z_\lambda) + 2\|z_\lambda - v\| \|v - m_\lambda(z_\lambda)\|, \\
&\leq -2d_{\mathrm{conv}(S)}^2(z_\lambda) + 2(D/2 - \epsilon)\|v - m_\lambda(z_\lambda)\|.
\end{aligned}
$$

where in the second to last inequality we use the fact that $v \in \text{conv}(S)$ and hence

$$\langle z_\lambda - \text{proj}_{\text{conv}(S)}(z_\lambda), \text{proj}_{\text{conv}(S)}(z_\lambda) - v \rangle \geq 0.$$

Note that $v$ is exactly the quantity used in the proof of Proposition 4.3 which by the absorbing property in the first part of the proof, we can apply (in terms of $\lambda$) to $z_\lambda$ and obtain

$$\|v - m_\lambda(z_\lambda)\| \leq \text{diam}(\text{supp}(p))\sqrt{\frac{1 - a_S}{a_S}} \exp\left(-\frac{3D\epsilon}{2} e^{2\lambda}\right).$$

Consequently, we have

$$\frac{d\, d^2_{\text{conv}(S)}(z_\lambda)}{d\lambda} \leq -2d^2_{\text{conv}(S)}(z_\lambda) + 2(D/2 - \epsilon)\text{diam}(\text{supp}(p))\sqrt{\frac{1 - a_S}{a_S}} \exp\left(-\frac{3D\epsilon}{2} e^{2\lambda}\right).$$

For notation simplicity, we denote $\phi(e^{-\lambda}) = (D/2 - \epsilon)\text{diam}(\text{supp}(p))\sqrt{\frac{1 - a_S}{a_S}} \exp\left(-\frac{3D\epsilon}{2} e^{2\lambda}\right)$. Then we have

$$\frac{d\, d^2_{\text{conv}(S)}(z_\lambda)}{d\lambda} \leq -2d^2_{\text{conv}(S)} + 2\phi(e^{-\lambda})$$

By Remark C.8, we have that

$$d_{\text{conv}(S)}(z_\lambda) \leq e^{-(\lambda - \lambda_1)} d_{\text{conv}(S)}(z_{\lambda_1}) + e^{-\lambda}\sqrt{\int_{\lambda_1}^{\lambda} 2e^{2t}\phi(e^{-t})dt}.$$

Now we want to bound $2e^{2t}\phi(e^{-t})$, by introducing the constants $C_1 := (D/2 - \epsilon)\text{diam}(\text{supp}(p))\sqrt{\frac{1 - a_S}{a_S}}$ and $C_2 = \frac{3D\epsilon}{2}$, we have

$$2e^{2t}\phi(e^{-t}) = 2C_1 e^{2t} \exp\left(-C_2\, e^{2t}\right)$$
$$\leq \frac{2C_1}{\sqrt{2C_2 e}} e^t,$$

where we used the fact that $e^t \exp\left(-C_2\, e^{2t}\right)$ is maximized at $t = \frac{1}{2}\log\left(\frac{1}{2C_2}\right)$ and the maximum value is $\frac{1}{\sqrt{2C_2 e}}$. Then we have

$$d_{\text{conv}(S)}(z_\lambda) \leq e^{-(\lambda - \lambda_1)} d_{\text{conv}(S)}(z_{\lambda_1}) + e^{-\lambda}\sqrt{\int_{\lambda_1}^{\lambda} \frac{2C_1}{\sqrt{2C_2 e}} e^t dt},$$
$$\leq e^{-(\lambda - \lambda_1)} d_{\text{conv}(S)}(z_{\lambda_1}) + \sqrt{\frac{2C_1}{\sqrt{2C_2 e}}}\sqrt{e^{-\lambda} - e^{\lambda_1 - 2\lambda}}.$$

In terms of $\sigma$, we have that

$$d_{\text{conv}(S)}(x_\sigma) \leq \frac{\sigma}{\sigma_1} d_{\text{conv}(S)}(x_{\sigma_1}) + \sqrt{\frac{2(D/2 - \epsilon)\text{diam}(\text{supp}(p))\sqrt{\frac{1 - a_S}{a_S}}}{\sqrt{3D\epsilon e}}}\sqrt{\sigma(1 - \sigma/\sigma_1)}. \tag{40}$$

In particular, $d_{\text{conv}(S)}(x_\sigma) \to 0$ as $\sigma \to 0$. $\qquad\square$

We can extend Proposition 4.4 to the case where the data distribution $p$ is of the form $p = p_b * \mathcal{N}(0, \delta^2 I)$ for some $p_b$ that has a well-separated local cluster $S$ and $\delta > 0$ is some constant. We have the following result.

**Corollary H.1.** *Assume that $S$ is a local cluster of a probability measure $p_b$ satisfying the Local Cluster Assumption and $a_S := p_b(S) > 0$. Let $p = p_b * \mathcal{N}(0, \delta^2 I)$ for some $\delta > 0$ is some constant. Then for any $0 < \epsilon < D/2$, let $C_\epsilon^S := \frac{D/2 - \epsilon}{\text{diam}(\text{supp}(p))\sqrt{\frac{1 - a_S}{a_S}}}$, and define:*

$$\sigma_0(S, \epsilon) = \begin{cases} \infty & \text{if } C_\epsilon^S \geq 1, \\ \left(-\frac{3D\epsilon}{2\log(C_\epsilon^S)}\right)^{1/2} & \text{if } C_\epsilon^S < 1. \end{cases}$$

*Assume that $\delta^2 < \sigma_0(S, \epsilon)$. Then, for any $\sigma_1 < \sqrt{\sigma_0(S, \epsilon) - \delta^2}$ the set $\overline{B_{D/2 - \epsilon}}(\text{conv}(S))$ is absorbing on the interval $(0, \sigma_1]$ with respect to the distribution $p$. Moreover, for any FM ODE trajectory $(x_\sigma)_{\sigma \in (0, \sigma_1]}$ with data distribution $p$ starting from a point $x_{\sigma_1} \in \partial \overline{B_{D/2 - \epsilon}}(\text{conv}(S))$, the following estimate for the distance $d_{\text{conv}(S)}(x_\sigma)$ holds:*

$$d_{\text{conv}(S)}(x_\sigma) \leq \frac{\sqrt{\sigma^2 + \delta^2}}{\sqrt{\sigma_1^2 + \delta^2}} d_{\text{conv}(S)}(x_{\sigma_1}) + \sqrt{\frac{2(D/2 - \epsilon)\text{diam}(\text{supp}(p))\sqrt{\frac{1 - a_S}{a_S}}}{\sqrt{3D\epsilon e}}} \sqrt{\left(\sqrt{\sigma^2 + \delta^2}\right)\left(1 - \sqrt{\frac{\sigma^2 + \delta^2}{\sigma_1^2 + \delta^2}}\right)}.$$

*Proof of Corollary H.1.* Let $(x_\sigma)_{\sigma \in (0, \sigma_1]}$ be the trajectory of the FM ODE with data distribution $p$ starting at some $x_{\sigma_1}$. By Lemma C.12, we have that $x_\sigma = y_{\sqrt{\sigma^2 + \delta^2}}$ for all $\sigma \in (0, \sigma_1]$ where $(y_{\sigma_b})_{\sigma_b \in (0, \sqrt{\sigma_1^2 + \delta^2}]}$ is the FM ODE trajectory with data distribution $p_b$ starting from $y_{\sqrt{\sigma_1^2 + \delta^2}} = x_{\sigma_1}$.

Since $\sigma_1 < \sqrt{\sigma_0(S, \epsilon) - \delta^2}$, we have that $\sqrt{\sigma_1^2 + \delta^2} < \sigma_0(S, \epsilon)$. By Proposition 4.4 we have that $\overline{B_{D/2 - \epsilon}}(\text{conv}(S))$ is absorbing in $(0, \sqrt{\sigma_1^2 + \delta^2}]$ for the distribution $p_b$ and hence absorbing in $(0, \sigma_1]$ for the distribution $p$.

For the second part, we apply Equation (40) to the trajectory $x_\sigma = y_{\sqrt{\sigma^2 + \delta^2}}$ and obtain the desired result. □

## H.1. Proof of Theorem 4.1

We are going to apply the theory of continuity equations to show that the flow ODE is well-posed. In particular, we need the following result which is a direct consequence of (Ambrosio et al., 2008, Lemma 8.1.4, Proposition 8.1.8) and (Ambrosio & Crippa, 2014, Remark 7).

**Lemma H.2.** *Let $(q_t)_{t \in [0,1)}$ be a narrowly continuous family of probability measures on $\mathbb{R}^d$ with densities solving the continuity equation below w.r.t. a smooth vector field $v_t$:*

$$\partial_t q_t + \nabla \cdot (v_t q_t) = 0, t \in [0, 1).$$

*We further assume that*

1. *$\int_0^1 \int_{\mathbb{R}^d} \|v_t(x)\| q_t(dx) dt < \infty$;*

2. *for any $t_1 \in [0, 1)$ and any compact set $K \subset \mathbb{R}^d$,*

$$\int_0^{t_1} \left(\sup_K \|v_t\| + \text{Lip}(v_t, K)\right) < \infty,$$

   *where $\text{Lip}(v_t, K)$ is the Lipschitz constant of $v_t$ on $K$;*

3. *for any $0 < t_0 < t_1 < 1$, there exists a constant $C_{t_0, t_1}$ such that*

$$\|v_t(x)\| \leq C_{t_0, t_1}(1 + \|x\|), \text{ for all } t \in [t_0, t_1].$$

*Then, the ODE*

$$\frac{dx}{dt} = u_t(x), x(0) = x_0$$

*has a unique solution $x(t)$ for all $t \in [0, 1)$. Furthermore, the flow map $\Psi_t : \mathbb{R}^d \to \mathbb{R}^d$ is continuous and satisfies that $(\Psi_t)_\# q_0 = q_t$ for all $t \in [0, 1)$.*

*Proof of Lemma H.2.* By (Ambrosio et al., 2008, Proposition 8.1.8), for any $t_1 \in [0, 1)$, we have that the flow map $(\Psi_t)_{t \in [0, t_1]}$ exists for $q_0$-a.e. $x \in \mathbb{R}^d$ and satisfies that $(\Psi_t)_{\#} q_0 = q_t$. We can take a sequence of rational numbers $t_1^{(n)} \to 1$ to conclude that the flow map $(\Psi_t)_{t \in [0, 1)}$ exists for $q_0$-a.e. $x \in \mathbb{R}^d$ and satisfies that $(\Psi_t)_{\#} q_0 = q_t$.

Now, we need to upgrade the $q_0$-a.e. conclusion to all points $x$. By (Ambrosio et al., 2008, Lemma 8.1.4), for each initial point $x_0 \in \mathbb{R}^d$, the ODE admits a unique (right) maximal solution defined in a maximal interval $I(x_0) = [0, \tau(x_0))$. Pick any $t_0 \in (0, \tau(x_0))$ and any $t_1 \in (\tau(x_0), 1)$. Then, the ODE starting at $x_{t_0}$ at time $t_0$ has a unique (right) maximal solution in the interval $[t_0, \tau(x_0))$. Suppose on the contrary that $\tau(x_0) < 1$. By item 3 in the lemma, we have that the solution $(x(t))_{t \in [t_0, \tau(x_0))}$ such that $x(t_0) = x_{t_0}$ must be uniformly bounded in $t$ because $\|v_t\|$ has a linear growth rate for all $t \in [t_0, t_1]$. This can be seen easily by applying Grönwall's inequality to the differential inequality below:

$$\frac{d\|x_t\|^2}{dt} = 2\langle x_t, v_t(x_t) \rangle \leq 2C_{t_0, t_1} \|x_t\| (1 + \|x_t\|) \leq \begin{cases} 4C_{t_0, t_1}, & \text{if } \|x_t\| \geq 1 \\ 4C_{t_0, t_1} \|x_t\|^2, & \text{if } \|x_t\| > 1 \end{cases}.$$

By (Ambrosio et al., 2008, Lemma 8.1.4) again, $\tau(x_0) \geq t_1$, contradiction. Therefore, $\tau(x_0) = 1$ and this proves that the flow map $\Psi_t$ is well-defined for all $t \in [0, 1)$ and all $x_0 \in \mathbb{R}^d$.

Finally, since the solution exists and is unique, and the vector field is locally Lipschitz, it follows that the flow map $\Psi_t$ is continuous; see, for example, (Khalil & Grizzle, 2002, Theorem 3.5). $\square$

Now, we verify that the assumptions in Lemma H.2 are satisfied by the vector field $u_t$ defined in Equation (8) and the probability measure $p_t$ defined in Equation (1). First of all, for any $p$ with a finite 2-moment, the path $(p_t)_{t \in [0,1]}$ is obviously narrowly continuous. Furthermore, each $p_t$ is absolutely continuous w.r.t. Lebesgue measure for any $t \in [0, 1)$ with a density function given by Equation (1).

Gao et al. (2024, Theorem 3.1) shows that as long as $p$ has a finite 2-moment and absolutely continuous w.r.t. Lebesgue measure, the density function of $(p_t)_{t \in [0,1]}$ satisfies the continuity equation

$$\partial_t p_t + \nabla \cdot (u_t p_t) = 0, t \in [0, 1], \tag{41}$$

where $u_t$ is defined in Equation (8).

Note that our assumption only requires $p$ to have a finite 2-moment. In this case, although $p$ may not have a density function, for all $t \in [0, 1)$, the probability measure $p_t$ is absolutely continuous w.r.t. Lebesgue measure. We point out that the same proof of (Gao et al., 2024, Theorem 3.1) is valid under this more general assumption that $p$ has a finite 2-moment, and hence the continuity equation Equation (41) still holds when we restrict to $t \in [0, 1)$ which is the case we are interested in this theorem.

Now we verify the integrality of $u_t$ to show that item 1 in Lemma H.2 is satisfied.

$$\int_0^1 \int \|u_t(x)\| \, p_t(dx) dt$$
$$\leq \int_0^1 \int \int \|u_t(x|x_1)\| p_t(dx|\boldsymbol{X} = x_1) \, p(dx_1) dt$$
$$= \int_0^1 \int \int \left\| \frac{\dot{\beta}_t}{\beta_t} x + \frac{\dot{\alpha}_t \beta_t - \alpha_t \dot{\beta}_t}{\beta_t} x_1 \right\| \frac{1}{(2\pi \beta_t^2)^{d/2}} \exp\left( -\frac{\|x - \alpha_t x_1\|^2}{2\beta_t^2} \right) dx \, p(dx_1) dt$$
$$\leq \int_0^1 \int \int \left( \left\| \frac{\dot{\beta}_t}{\beta_t} x - \frac{\alpha_t \dot{\beta}_t}{\beta_t} x_1 \right\| + \|\dot{\alpha}_t x_1\| \right) \frac{1}{(2\pi \beta_t^2)^{d/2}} \exp\left( -\frac{\|x - \alpha_t x_1\|^2}{2\beta_t^2} \right) dx \, p(dx_1) dt.$$

We split the integral by the two terms according to the summation $\|\frac{\dot{\beta}_t}{\beta_t} x - \frac{\alpha_t \dot{\beta}_t}{\beta_t} x_1\| + \|\dot{\alpha}_t x_1\|$. For the first term, we use

$\tilde{x} = x - \alpha_t x_1$ and the fact about the expected norm of a Gaussian random variable with variance $\sigma^2$ is $\sigma \sqrt{\frac{\pi}{2}} \frac{\Gamma((n+1)/2)}{\Gamma(n/2)}$.

$$\int_0^1 \int \int \left|\frac{\dot{\beta}_t}{\beta_t}\right| \|x - \alpha_t x_1\| \frac{1}{(2\pi\beta_t^2)^{d/2}} \exp\left(-\frac{\|x - \alpha_t x_1\|^2}{2\beta_t^2}\right) dx \, p(dx_1) dt$$

$$= \int_0^1 \int \left|\frac{\dot{\beta}_t}{\beta_t}\right| \beta_t \sqrt{\frac{\pi}{2}} \frac{\Gamma((d+1)/2)}{\Gamma(d/2)} p(dx_1) dt$$

$$= \sqrt{\frac{\pi}{2}} \frac{\Gamma((d+1)/2)}{\Gamma(d/2)} \int_0^1 -\dot{\beta}_t dt$$

$$= \sqrt{\frac{\pi}{2}} \frac{\Gamma((d+1)/2)}{\Gamma(d/2)},$$

where we use the assumption that $\beta_t$ is a non-increasing function of $t$ and hence $\dot{\beta}_t \leq 0$.

For the second term, we have that

$$\int_0^1 \int \int \|\dot{\alpha}_t x_1\| \frac{1}{(2\pi\beta_t^2)^{d/2}} \exp\left(-\frac{\|x - \alpha_t x_1\|^2}{2\beta_t^2}\right) dx \, p(dx_1) dt$$

$$\leq \int_0^1 \int \dot{\alpha}_t \|x_1\| p(dx_1) dt$$

$$\leq \int \|x_1\| p(dx_1) < \infty.$$

The last step follows from the fact that $p$ has a finite 2-moment and, hence, a finite first moment. We also use the assumption that $\alpha_t$ is a non-decreasing function of $t$ and hence $\dot{\alpha}_t \geq 0$.

Now we verify that $u_t$ satisfies item 2 in Lemma H.2. Recall $u_t(x) = \dot{\beta}_t/\beta_t \cdot x + (\dot{\alpha}_t\beta_t - \alpha_t\dot{\beta}_t)/\beta_t \cdot \mathbb{E}[\boldsymbol{X}|\boldsymbol{X}_t = x]$ and by Proposition C.2, we have that

$$\nabla_x m_t(x) = \frac{\alpha_t}{\beta_t^2} \text{Cov}[\boldsymbol{X}|\boldsymbol{X}_t = x],$$

where $\text{Cov}[\boldsymbol{X}|\boldsymbol{X}_t = x]$ denotes the covariance matrix of the posterior distribution $p(\cdot|\boldsymbol{X}_t = x)$. For simplicity, we let $\Sigma_t(x) := \text{Cov}[\boldsymbol{X}|\boldsymbol{X}_t = x]$ below. Therefore,

$$\nabla_x u_t(x) = \frac{\dot{\beta}_t}{\beta_t} I + \frac{\alpha_t(\dot{\alpha}_t\beta_t - \alpha_t\dot{\beta}_t)}{\beta_t^3} \Sigma_t(x).$$

Recall that $\alpha_0 = 0$, $\beta_0 = 1$ and $\alpha_t, \beta_t$ are positive continuous when $t \in (0, 1)$. Also we have assumed that derivatives exist and are bounded. Therefore, for any $t_1 \in [0, 1)$, the coefficients above consisting of $\alpha_t, \beta_t$ will be uniformly bounded within $[0, t_1]$. Hence, to prove the locally Lipschitz property, it suffices to show that the covariance matrix $\Sigma_t(x)$ of the posterior distribution $p(\cdot|\boldsymbol{X}_t = x)$ is locally (w.r.t. $x$) uniformly bounded (w.r.t. $t$). We establish the following lemma for this purpose.

**Lemma H.3.** *Let $p$ be a probability measure on $\mathbb{R}^d$ with a finite 2-moment $\mathsf{M}_2(p)$. For any $x \in \mathbb{R}^d$, consider the posterior distribution:*

$$p(dz|\boldsymbol{X}_t = x) = \frac{\exp\left(-\frac{\|x - \alpha_t z\|^2}{2\beta_t^2}\right) p(dz)}{\int_\Omega \exp\left(-\frac{\|x - \alpha_t z\|^2}{2\beta_t^2}\right) p(dz)}.$$

*We let $N_t(x) := \int \exp\left(-\frac{\|x - \alpha_t z\|^2}{2\beta_t^2}\right) p(dz)$. Then, the covariance matrix $\Sigma_t(x)$ of $p(\cdot|\boldsymbol{X}_t = x)$ satisfies:*

$$\Sigma_t(x) \preceq \left(2\|m_t(x)\|^2 + \frac{2\mathsf{M}_2(p)}{N_t(x)}\right) I.$$

*Proof of Lemma H.3.* Fix a unit vector $v$. Then, we have that

$$v^\top \Sigma_t(x) v = \int \langle z - m_t(x), v \rangle^2 \frac{\exp\left(-\frac{\|x - \alpha_t z\|^2}{2\beta_t^2}\right) p(dz)}{\int \exp\left(-\frac{\|x - \alpha_t z'\|^2}{2\beta_t^2}\right) p(dz')}.$$

For $z \in \mathbb{R}^d$, we have that

$$\langle z - m_t(x), v \rangle^2 \le \|z - m_t(x)\|^2 \le 2\|z\|^2 + 2\|m_t(x)\|^2.$$

Therefore, one has that

$$v^\top \Sigma_t(x) v \le 2\|m_t(x)\|^2 + 2 \int \|z\|^2 \frac{\exp\left(-\frac{\|x - \alpha_t z\|^2}{2\beta_t^2}\right) p(dz)}{\int \exp\left(-\frac{\|x - \alpha_t z'\|^2}{2\beta_t^2}\right) p(dz')}$$

$$= 2\|m_t(x)\|^2 + \frac{2 \int \|z\|^2 p(dz)}{\int \exp\left(-\frac{\|x - \alpha_t z'\|^2}{2\beta_t^2}\right) p(dz')}$$

$$= 2\|m_t(x)\|^2 + \frac{2\mathsf{M}_2(p)}{N_t(x)}.$$

Since $v$ is arbitrary, this concludes the proof. $\qquad\square$

By dominated convergence theorem, it is straightforward to check that $N : [0, 1) \times \mathbb{R}^d \to \mathbb{R}$ is continuous w.r.t. $(t, x)$. Hence, for any $x \in \mathbb{R}^d$, $t \in [0, 1)$ and any local compact neighborhood of $x$ in $\mathbb{R}^d$, $N$ is bounded below by some positive constant. Similarly, $m_t(x)$ is continuous w.r.t. $x$, we have that $m_t(x)$ is locally uniformly bounded as well in any local compact neighborhood of $x$ in $\mathbb{R}^d$. These together with Proposition C.2 and Equation (8) imply that the vector field $u : [0, 1) \times \mathbb{R}^d \to \mathbb{R}^d$ is locally Lipschitz in $x$ for any fixed $t \in [0, 1)$. In fact, by continuity of $N_t$ and $m_t$ in $t$, we have that for any fixed $t_1 \in [0, 1)$ and any compact set $K \subset \mathbb{R}^d$,

$$\sup_{t \in [0, t_1]} \sup_K \|u_t\| + \mathrm{Lip}_K(u_t) < \infty.$$

This implies that $u_t$ satisfies item 2 in Lemma H.2.

Finally, we show that $u_t$ satisfies item 3 in Lemma H.2. This is done by establishing the following claim.

*Claim* 3. There exists a positive constant $\hat{C}_t$ continuously dependent on $t \in (0, 1)$ so that for any $x \in \mathbb{R}^d$

$$\|u_t(x)\| \le \hat{C}_t(1 + \|x\|).$$

Here the constant $\hat{C}_t$ may blow up as $t \to 0$ or $t \to 1$ but it is finite for any fixed $t \in (0, 1)$.

*Proof of Claim 3.* We first show that for any $\sigma \in (0, \infty)$, then for any $x \in \mathbb{R}^d$, we have that

$$\|m_\sigma(x)\| \le C_\sigma(1 + \|x\|).$$

where $C_\sigma$ continuously depends on $\sigma$.

By Markov's inequality, we have that for any $a > 0$,

$$p(\|\boldsymbol{X} - x\| \ge a) \le \frac{\mathbb{E}\|\boldsymbol{X} - x\|}{a} \le \frac{\mathsf{M}_1 + \|x\|}{a},$$

where $\mathsf{M}_1$ denotes the first moment of the distribution $p$. We let $A_x := \lceil 2(\mathsf{M}_1 + \|x\|) \rceil \in \mathbb{N}$. Then, we have the following

estimates:

$$\|m_\sigma(x) - x\| = \left\| \frac{\int \exp\left(\frac{-\|x-y\|^2}{2\sigma^2}\right)(y-x)p(dy)}{\int \exp\left(\frac{-\|x-y'\|^2}{2\sigma^2}\right)p(dy')} \right\|$$

$$\leq \underbrace{\sum_{n=A_x+1}^{\infty} \frac{\int_{n \leq \|y-x\| < n+1} \exp\left(\frac{-\|x-y\|^2}{2\sigma^2}\right)\|y-x\|p(dy)}{\int \exp\left(\frac{-\|x-y'\|^2}{2\sigma^2}\right)p(dy')}}_{I_1} + \underbrace{\frac{\int_{\|y-x\| < A_x+1} \exp\left(\frac{-\|x-y\|^2}{2\sigma^2}\right)\|y-x\|p(dy)}{\int \exp\left(\frac{-\|x-y'\|^2}{2\sigma^2}\right)p(dy')}}_{I_2}.$$

If we let $\bar{X} := \chi_{\|X-x\| < A_x+1} \cdot \|X - x\|$, then we have that

$$I_2 = \mathbb{E}[\bar{X}|X_\sigma = x] \leq A_x + 1.$$

Then, we pick any small $0 < \epsilon < 1$. We have that

$$I_1 \leq \sum_{n=A_x+1}^{\infty} \frac{\int_{n \leq \|y-x\| < n+1} \exp\left(\frac{-\|x-y\|^2}{2\sigma^2}\right)\|y-x\|p(dy)}{\int_{\|y-x\| < n-\epsilon} \exp\left(\frac{-\|x-y'\|^2}{2\sigma^2}\right)p(dy')}$$

$$\leq \sum_{n=A_x+1}^{\infty} \frac{(n+1)\exp\left(\frac{-n^2}{2\sigma^2}\right)p(\{y : n \leq \|y-x\|\})}{\exp\left(\frac{-(n-\epsilon)^2}{2\sigma^2}\right)p(\{y : \|y-x\| < n-\epsilon\})}$$

$$\leq \sum_{n=A_x+1}^{\infty} \frac{(n+1)\exp\left(\frac{-(2\epsilon n - \epsilon^2)}{2\sigma^2}\right)p(\{y : n - \epsilon \leq \|y-x\|\})}{p(\{y : \|y-x\| < n-\epsilon\})}.$$

We have that

$$p(\{y : n - \epsilon \leq \|y-x\|\}) = p(\|X - x\| \geq n - \epsilon) \leq p(\|X - x\| \geq A_x) \leq \frac{\mathsf{M}_1 + \|x\|}{A_x} \leq \frac{1}{2}.$$

Then, we have that

$$\frac{p(\{y : n - \epsilon \leq \|y-x\|\})}{p(\{y : \|y-x\| < n-\epsilon\})} = \frac{p(\{y : n - \epsilon \leq \|y-x\|\})}{1 - p(\{y : n - \epsilon \leq \|y-x\|\})} \leq 1.$$

In this way, we can continue to bound $I_1$ as follows:

$$I_1 \leq \sum_{n=A_x+1}^{\infty} (n+1)\exp\left(\frac{-(2\epsilon n - \epsilon^2)}{2\sigma^2}\right)$$

$$= \exp\left(\frac{\epsilon^2}{2\sigma^2}\right) \sum_{n=1}^{\infty} (n+1)\exp\left(-\frac{\epsilon}{\sigma^2}n\right) =: C_{\epsilon,\delta} < \infty.$$

The final inequality follows from the fact that $\sum_{n=1}^{\infty}(n+1)\exp\left(-\frac{\epsilon}{\sigma^2}n\right)$ is a convergent series (which can be seen easily from ratio test).

Then, we have that

$$\|m_\sigma(x)\| \leq \|x\| + I_1 + I_2 \leq \|x\| + C_{\epsilon,\delta} + A_x + 1 \leq C_{\epsilon,\delta} + 2 + 2\mathsf{M}_1 + 3\|x\| = C_\delta(1 + \|x\|),$$

where $C_\delta$ is a constant that depends on $\delta$ and the first moment of the distribution $p$ (we can simply take $\epsilon = 1/2$ to remove the dependency on $\epsilon$).

This implies that

$$\|m_t(x)\| = \|m_{\sigma_t}(x/\alpha_t)\| \le C_{\delta_t}(1 + \|x/\alpha_t\|) \le C_t(1 + \|x\|)$$

where $C_t := C_{\delta_t}/\alpha_t$ is continuously dependent on $t \in (0,1)$ (note that $t = 0$ has to be excluded otherwise $C_0 = \infty$).

Therefore, as $u_t$ is a linear combination of $x$ and $m_t$ (cf. Equation (8)), we have that

$$\|u_t(x)\| \le \hat{C}_t(1 + \|x\|)$$

for some constant $\hat{C}_t$ that is continuously dependent on $t$ for all $t \in (0,1)$. $\qquad\square$

Now, for any $0 < t_0 < t_1 < 1$, by continuity of $C_t$, we have that $C_{t_0,t_1} := \sup_{t \in [t_0,t_1]} C_t < \infty$ and hence we have that $u_t$ satisfies item 3 in Lemma H.2.

Then, by applying Lemma H.2, we conclude the proof.

### H.2. Proof of Proposition 4.2

We now prove Proposition 4.2 which is a consequence of how close the posterior distribution is to the data distribution when $\sigma$ is large. We will first only consider the case where the data distribution has bounded support and then extend the result to the case where the data distribution is a convolution of a bounded support distribution and a Gaussian distribution using Lemma C.12.

**Proposition H.4** (Initial stability of posterior measure). *Let $p$ be a probability measure on $\mathbb{R}^d$ with bounded support which is denoted as $\Omega := \mathrm{supp}(p)$. Let $x$ be a point and consider the posterior measure $p(\cdot|\boldsymbol{X}_\sigma = x)$. We then have the following Wasserstein distance bound:*

$$d_{\mathrm{W},1}(p(\cdot|\boldsymbol{X}_\sigma = x), p) < \left( \exp\left( \frac{2\|x - \mathbb{E}[\boldsymbol{X}]\|\mathrm{diam}(\Omega)}{\sigma^2} \right) - 1 \right) \mathrm{diam}(\Omega).$$

*Proof of Proposition H.4.* Let $R_1 = \|x - \mathbb{E}[\boldsymbol{X}]\|$. Consider the function $g_\sigma(y) = \exp\left(-\frac{\|x-y\|^2}{2\sigma^2}\right)$. By the fact that $\|y - \mathbb{E}[\boldsymbol{X}]\| \le \mathrm{diam}(\Omega)$ for any $y \in \Omega$, we have that:

$$\exp\left( -\frac{(R_1 + \mathrm{diam}(\Omega))^2}{2\sigma^2} \right) \le g_\sigma(y) \le \exp\left( -\frac{(R_1 - \mathrm{diam}(\Omega))^2}{2\sigma^2} \right)$$

for all $y \in \Omega$. Then for any Borel measurable set $A \subseteq \Omega$, we can bound the ratio of the posterior and the data distribution as:

$$
\begin{aligned}
\frac{p(A|\boldsymbol{X}_\sigma = x)}{p(A)} &= \frac{\int_A g_\sigma(y) p(dy)}{p(A) \int_\Omega g_\sigma(y) p(dy)} \\
&\le \frac{\exp\left( -\frac{(R_1 - \mathrm{diam}(\Omega))^2}{2\sigma^2} \right) \int_A p(dy)}{p(A) \exp\left( -\frac{(R_1 + \mathrm{diam}(\Omega))^2}{2\sigma^2} \right) \int_\Omega p(dy)} \\
&= \underbrace{\exp\left( \frac{(R_1 + \mathrm{diam}(\Omega))^2}{2\sigma^2} - \frac{(R_1 - \mathrm{diam}(\Omega))^2}{2\sigma^2} \right)}_{a}.
\end{aligned}
$$

Similarly, we can bound the ratio from below:

$$
\begin{aligned}
\frac{p(A|\boldsymbol{X}_\sigma = x)}{p(A)} &\ge \frac{\exp\left( -\frac{(R_1 + \mathrm{diam}(\Omega))^2}{2\sigma^2} \right) \int_A p(dy)}{p(A) \exp\left( -\frac{(R_1 - \mathrm{diam}(\Omega))^2}{2\sigma^2} \right) \int_\Omega p(dy)} \\
&= \exp\left( \frac{(R_1 - \mathrm{diam}(\Omega))^2}{2\sigma^2} - \frac{(R_1 + \mathrm{diam}(\Omega))^2}{2\sigma^2} \right) \\
&= 1/a.
\end{aligned}
$$

Since $a > 1$, we have $|1/a - 1| \leq a - 1$. Therefore, we have:

$$\left| \frac{p(A|\boldsymbol{X}_\sigma = x)}{p(A)} - 1 \right| \leq a - 1.$$

In particularly, there is $|p(A|\boldsymbol{X}_\sigma = x) - p(A)| \leq a - 1$ for all $A \subseteq \Omega$ which by definition bounds the total variation distance between $p(\cdot|\boldsymbol{X}_\sigma = x)$ and $p$ by $a - 1$. Then the Wasserstein distance $d_{W,1}(p(\cdot|\boldsymbol{X}_\sigma = x), p)$ can be bounded by $(a - 1)\mathrm{diam}(\Omega)$; see e.g. Villani (2003, Proposition 7.10). □

Now we are ready to prove the following special case of Proposition 4.2.

**Proposition H.5.** *Let $p$ be a probability measure on $\mathbb{R}^d$ with bounded support which is denoted as $\Omega := \mathrm{supp}(p)$. Let $x_0$ be a point and denote $\|x_0 - \mathbb{E}[\boldsymbol{X}]\| = R_0$, where $\boldsymbol{X} \sim p$. Let $\zeta$ be a parameter such that $0 < \zeta < 1$. Then the constant*

$$\sigma_{init}(\Omega, \zeta, R_0) := \sqrt{\frac{2R_0\mathrm{diam}(\Omega)}{\log\left(1 + \frac{\zeta R_0}{\mathrm{diam}(\Omega)}\right)}}$$

*satisfies that for any $\sigma_1 > \sigma_{init}(\Omega, \zeta, R_0)$, the ODE trajectory $(x_\sigma)_{\sigma \in (\sigma_{init}(\Omega, \zeta, R_0), \sigma_1]}$ starting from $x_{\sigma_1} = x_0$ will move toward the mean of the data distribution $p$ as shown in the following estimate:*

$$\|x_\sigma - \mathbb{E}[\boldsymbol{X}]\| < \frac{\sigma^{1-\zeta}}{\sigma_1^{1-\zeta}}\|x_{\sigma_1} - \mathbb{E}[\boldsymbol{X}]\|,$$

*where $\boldsymbol{X} \sim p$.*

*Proof of Proposition H.5.* By the estimate in Proposition H.4 and the fact that $f(s) = \frac{s}{\log(1+s)}$ is increasing for $s > 0$, one can check that for any $\sigma > \sigma_{init}(\Omega, \zeta, R_0)$, and for all $z$ with $\|z - \mathbb{E}[\boldsymbol{X}]\| < R_0$ where $\boldsymbol{X}_b \sim p$ there is

$$d_{W,1}(p(\cdot|\boldsymbol{Y}_\sigma = z), p) < \zeta\|z - \mathbb{E}[\boldsymbol{X}]\|,$$

where $\boldsymbol{Y}_\sigma \sim q_\sigma$. Then by Lemma D.2, we have $\|m_\sigma(z) - \mathbb{E}[\boldsymbol{X}]\| < \zeta\|z - \mathbb{E}[\boldsymbol{X}]\|$.

This estimate shows that for all $\sigma \in (\sigma_{init}(\Omega, \zeta, R_0), \sigma_1]$ and for all $z \in \partial B_{R_0}(\mathbb{E}[\boldsymbol{X}])$, the denoiser $m_\sigma(z)$ lies in the ball $B_{\zeta R_0}(\mathbb{E}[\boldsymbol{X}])$ and hence the interior of $\overline{B_{R_0}(\mathbb{E}[\boldsymbol{X}])}$. The closed ball $\overline{B_{R_0}(\mathbb{E}[\boldsymbol{X}])}$ is clearly convex and we can then apply Item 2 of Proposition C.9 to conclude that the set $\overline{B_{R_0}(\mathbb{E}[\boldsymbol{X}])}$ is absorbing for the trajectory $(x_\sigma)_{\sigma \in (\sigma_{init}(\Omega, \zeta, R_0), \sigma_1]}$.

The absorbing result ensures the estimate the estimate

$$\|m_\sigma(x_\sigma) - \mathbb{E}[\boldsymbol{X}]\| < \zeta\|x_\sigma - \mathbb{E}[\boldsymbol{X}]\|$$

holds for the entire trajectory $(x_\sigma)_{\sigma \in (\sigma_{init}(\Omega, \zeta, R_0), \sigma_1]}$. We then obtain the result by Item 1 of the meta attracting result Theorem C.7. □

*Proof of Proposition 4.2.* We obtain the result by combining the results in Proposition H.5 and Lemma C.12. □

# I. Proofs in Section 5

In this section, we provide the missing proofs of the results in Section 5. Some proofs are already provided in the previous sections:

- The proof of Theorem 5.1 is presented in Appendix D as an immediate consequence of the technical result about posterior convergence (Theorem D.4).

- The proof for Example 5.5 is provided in Appendix E.

Also, the proofs of Theorem 5.3 and Theorem 5.4 require more preparation and are organized in Appendix I.1 and Appendix I.2, respectively, at the end of this section.

*Proof of Proposition 5.6.* For any independent random variables $\boldsymbol{X} \sim p$ and any $\boldsymbol{Z} \sim N(0, \sigma^2 I)$, we have that $\boldsymbol{X} + \boldsymbol{Z} \sim q_\sigma = p * N(0, \sigma^2 I)$. Hence,

$$d_{\mathrm{W},2}(q_\sigma, p)^2 \leq \mathbb{E}\left[\|\boldsymbol{X} + \boldsymbol{Z} - \boldsymbol{X}\|^2\right] = \mathbb{E}\left[\|\boldsymbol{Z}\|^2\right] = O(\sigma^2).$$

The result follows. □

*Proof of Proposition 5.7.* For any $t \in [0, 1)$, consider $y = s_t(\boldsymbol{O}x + \alpha_t b)$. The transformed denoiser $\bar{m}_t(y)$ w.r.t. $\bar{\alpha}_t := s_t \alpha_t / \gamma$ and $\bar{\beta}_t := s_t \beta_t$, as well as $\bar{p}$ is given by

$$
\begin{aligned}
\bar{m}_t(y) &= \int \frac{\exp\left(-\frac{\|y - \bar{\alpha}_t y_1\|^2}{2\bar{\beta}_t^2}\right) y_1}{\int \exp\left(-\frac{\|y - \bar{\alpha}_t y_1'\|^2}{2\bar{\beta}_t^2}\right) \bar{p}(dy_1')} \bar{p}(dy_1) \\
&= \int \frac{\exp\left(-\frac{\|s_t(\boldsymbol{O}x + \alpha_t b) - \alpha_t s_t(\boldsymbol{O}x_1 + b)\|^2}{2 s_t^2 \beta_t^2}\right) \gamma(\boldsymbol{O}x_1 + b)}{\int \exp\left(-\frac{\|s_t(\boldsymbol{O}x + \alpha_t b) - \alpha_t s_t(\boldsymbol{O}x_1 + b)\|^2}{2 s_t^2 \beta_t^2}\right) p(dx_1')} p(dx_1) \\
&= \int \frac{\exp\left(-\frac{\|x - \alpha_t x_1\|^2}{2\beta_t^2}\right) \gamma(\boldsymbol{O}x_1 + b)}{\int \exp\left(-\frac{\|x - \alpha_t x_1'\|^2}{2\beta_t^2}\right) p(dx_1')} p(dx_1) = \gamma(\boldsymbol{O}m_t(x) + b).
\end{aligned}
\tag{42}
$$

Let $x_t$ denote an ODE path for $dx_t/dt = u_t(x_t)$. Then we consider the path $y_t = s_t(\boldsymbol{O}x_t + \alpha_t b)$. We now check that $dy_t/dt = \bar{u}_t(y_t)$ as follows:

$$
\begin{aligned}
\frac{dy_t}{dt} &= s_t'(\boldsymbol{O}x_t + \alpha_t b) + s_t\left(\boldsymbol{O}\frac{dx_t}{dt} + \alpha_t' b\right) \\
&= s_t' \boldsymbol{O}x_t + s_t' \alpha_t b + s_t((\log \beta_t)' \boldsymbol{O}x_t + \beta_t(\alpha_t/\beta_t)' \boldsymbol{O}m_t(x_t)) + s_t \alpha_t' b \\
&= (s_t'/s_t + (\log \beta_t)')s_t(\boldsymbol{O}x_t + \alpha_t b) + \beta_t(\alpha_t/\beta_t)' s_t \boldsymbol{O}m_t(x_t) + s_t \alpha_t' b - (\log \beta_t)' s_t \alpha_t b \\
&= (\log \bar{\beta}_t)' y_t + \beta_t(\alpha_t/\beta_t)' s_t(\boldsymbol{O}m_t(x_t) + b) \\
&= (\log \bar{\beta}_t)' y_t + s_t \beta_t(\frac{\alpha_t}{\gamma \beta_t})' \gamma(\boldsymbol{O}m_t(x_t) + b) \\
&= (\log \bar{\beta}_t)' y_t + \bar{\beta}_t(\frac{\bar{\alpha}_t}{\bar{\beta}_t})' \bar{m}_t(y_t) = \bar{u}_t(y_t).
\end{aligned}
$$

Note that $y_0 = \boldsymbol{O}x_0$, hence we conclude for $t \in [0, 1)$ that

$$\overline{\Psi}_t(\boldsymbol{O}x_0) = s_t(\boldsymbol{O}\Psi_t(x_0) + \alpha_t b).$$

When $\Psi_1(x_0) = \lim_{t \to 1} \Psi_t(x_0)$ exists, by continuity of $s_t$ and $\alpha_t$, we have that $\overline{\Psi}_1(\boldsymbol{O}x_0) = \lim_{t \to 1} \overline{\Psi}_t(\boldsymbol{O}x_0)$ exists and that

$$\overline{\Psi}_1(\boldsymbol{O}x_0) = \gamma(\boldsymbol{O}\Psi_1(x_0) + b).$$

□

*Proof of Proposition 5.9.* Since each $V_i^\epsilon$ is a convex set and in fact, a closure of an open convex set, by item 2 of Proposition C.9, it suffices to prove that for all $x \in \partial V_i^\epsilon$ and all $\sigma < \sigma_0(V_i^\epsilon)$, which is the boundary of $V_i^\epsilon$, one has that $m_\sigma(x) \in \mathrm{int} V_i^\epsilon$ which is the interior of $V_i^\epsilon$.

First of all, we define

$$r_{i,\epsilon} := \frac{\mathrm{sep}^2(x_i) - \epsilon^2}{2\mathrm{sep}(x_i)}.$$

It is straightforward to check that $B_{r_{i,\epsilon}}(x_i) \subseteq V_i^\epsilon$. In fact, $r_{i,\epsilon} = \mathrm{argmax}\{r > 0 : B_r(x_i) \subseteq V_i^\epsilon\}$.

Hence, by Corollary D.8, for any $x \in \partial V_i^\epsilon$ we have that

$$\|m_\sigma(x) - x_i\| \le \operatorname{diam}(\Omega)\sqrt{\frac{1-a_i}{a_i}} \exp\left(-\frac{\Delta_\Omega(x)}{4\sigma^2}\right)$$

$$\le \operatorname{diam}(\Omega)\sqrt{\frac{1-a_i}{a_i}} \exp\left(-\frac{\epsilon^2}{4\sigma^2}\right).$$

Therefore, we need to identify when the above inequality is less than $r_{i,\epsilon}$, that is,

$$\operatorname{diam}(\Omega)\sqrt{\frac{1-a_i}{a_i}} \exp\left(-\frac{\epsilon^2}{4\sigma^2}\right) < r_{i,\epsilon},$$

$$\exp\left(-\frac{\epsilon^2}{4\sigma^2}\right) < \frac{r_{i,\epsilon}}{\operatorname{diam}(\Omega)\sqrt{\frac{1-a_i}{a_i}}},$$

Recall that $C_{i,\epsilon} = \frac{r_{i,\epsilon}}{\operatorname{diam}(\Omega)}\sqrt{\frac{a_i}{1-a_i}}$. Then, it is direct to check that when $C_{i,\epsilon} \le 1$, the above inequality holds for all $0 \le \sigma < \infty$ and when $C_{i,\epsilon} > 1$, the above inequality holds for all $\sigma < \sigma_0(V_i^\epsilon) = \frac{\epsilon}{2}\left(\log(C_{i,\epsilon}^\Omega)\right)^{-1/2}$. In summary, this implies that $m_\sigma(x) \in \operatorname{int}B_{r_{i,\epsilon}}(x_i) \subset \operatorname{int}V_i^\epsilon$ when $\sigma < \sigma_0(V_i^\epsilon)$. Then as stated above, by item 2 of Proposition C.9, we have that $V_i^\epsilon$ is absorbing.

Next, we show that $d_\Omega(z_\lambda) \to 0$ as $\lambda \to \infty$. Along the trajectory $z_\lambda$, by Corollary B.15 we have

$$\frac{d\, d_\Omega^2(z_\lambda)}{d\lambda} = -2\langle z_\lambda - x_i, z_\lambda - m_\lambda(z_\lambda)\rangle,$$

$$= -2\langle z_\lambda - x_i, z_\lambda - x_i\rangle + 2\langle z_\lambda - x_i, x_i - m_\lambda(z_\lambda)\rangle,$$

$$\le -2d_\Omega^2(z_\lambda) + 2d_\Omega(z_\lambda)\|x_i - m_\lambda(z_\lambda)\|,$$

Applying Corollary D.8 to points in $V_i^\epsilon$, we have the following uniform bound for all $y \in V_i^\epsilon$,

$$\|m_\lambda(y) - x_i\| \le \operatorname{diam}(\Omega)\sqrt{\frac{1-a_i}{a_i}} \exp\left(-\frac{1}{4}e^{2\lambda}\epsilon^2\right).$$

Since $z_\lambda \in V_i^\epsilon$, we have that

$$\frac{d\, d_\Omega^2(z_\lambda)}{d\lambda} \le -2d_\Omega^2(z_\lambda) + \operatorname{sep}(x_i)\operatorname{diam}(\Omega)\sqrt{\frac{1-a_i}{a_i}} \exp\left(-\frac{1}{4}e^{2\lambda}\epsilon^2\right).$$

Then by Lemma C.6, we have that $d_\Omega(z_\lambda) \to 0$ as $\lambda \to \infty$. By the the fact that $\Omega$ is discrete and $x_i$ is the nearest point to $z_\lambda$ for all large $\lambda$, we must have $z_\lambda \to x_i$ as $\lambda \to \infty$. □

*Proof of Proposition 5.10.* The proof idea follows the same line as the proofs of Proposition 5.9. Similarly as in the case of denoisers, we can also consider the change of variable $\lambda = -\log(\sigma)$ for handling $m_\sigma^\theta$. We let $z_\lambda^\theta := x_{\sigma(\lambda)}^\theta$ and let $m_\lambda^\theta(x) := m_{\sigma(\lambda)}^\theta(x)$ for any $x \in \mathbb{R}^d$. Then, it is straightforward to check that

$$\frac{dz_\lambda^\theta}{d\lambda} = m_\lambda^\theta(z_\lambda^\theta) - z_\lambda^\theta \tag{43}$$

and converting everything back to $\sigma$ is straightforward.

We first identify the parameter $\lambda_0(V_i^\epsilon, \phi)$ such that the denoiser $m_\lambda^\theta(x)$ is will always lie in the interior of $V_i^\epsilon$ for all $x \in \partial V_i^\epsilon$ and all $\lambda > \lambda_0(V_i^\epsilon, \phi)$.

By the triangle inequality, we have that

$$\|m_\lambda^\theta(y) - x_i\| \le \|m_\lambda^\theta(y) - m_\lambda^N(y)\| + \|m_\lambda^N(y) - x_i\|,$$

$$\le \phi(\lambda) + \operatorname{diam}(\Omega)\sqrt{\frac{1-a_i}{a_i}} \exp\left(-\frac{1}{4}e^{2\lambda}\epsilon^2\right).$$

Since $\phi(\lambda)$ goes to zero as $\lambda$ goes to infinity, there exists a parameter $\lambda_0(V_i^\epsilon, \phi)$ such that for all $\lambda > \lambda_0(V_i^\epsilon, \phi)$, $\|m_\lambda^\theta(y) - x_i\| \leq \frac{\mathrm{sep}^2(x_i) - \epsilon^2}{2\mathrm{sep}(x_i)}$. Then, with the same argument as in the proof of Proposition 5.9, we can prove that the trajectory $z_\lambda^\theta$ will never leave $V_i^\epsilon$ for all $\lambda > \lambda_0(V_i^\epsilon, \phi)$.

Since the trajectory $z_\lambda^\theta$ never leaves $V_i^\epsilon$ for all $\lambda > \lambda_0(V_i^\epsilon, \phi)$, we can then apply the uniform decay of $\|m_\lambda^\theta(y) - x_i\|$ to the differential inequality of $d_\Omega^2(z_\lambda^\theta)$ as follows:

$$\frac{d\, d_\Omega^2(z_\lambda^\theta)}{d\lambda} \leq -2d_\Omega^2(z_\lambda^\theta) + 2d_\Omega(z_\lambda^\theta)\left\|x_i - m_\lambda^\theta(z_\lambda^\theta)\right\|,$$

$$\leq -2d_\Omega^2(z_\lambda^\theta) + 2d_\Omega(z_\lambda^\theta)\left(\phi(\lambda) + \mathrm{diam}(\Omega)\sqrt{\frac{1 - a_i}{a_i}}\exp\left(-\frac{1}{4}e^{2\lambda}\epsilon^2\right)\right).$$

Now, we apply Lemma C.6 again as in the proof of Proposition 5.9, we have that $d_\Omega(z_\lambda^\theta)$ goes to zero as $\lambda$ goes to infinity and hence $z_\lambda^\theta$ converges to $x_i$.

For the second part, the limits $z_\infty^\theta := \lim_{\lambda\to\infty} z_\lambda^\theta$ and $\lim_{\lambda\to\infty} m_\lambda^\theta(z_\lambda^\theta)$ are known to exist. In particular, the limit of the derivative $\lim_{\lambda\to\infty}\frac{dz_\lambda^\theta}{d\lambda} = \lim_{\lambda\to\infty} m_\lambda^\theta(z_\lambda^\theta) - z_\lambda^\theta$ exists and we will show that it has to be zero.

Suppose $\lim_{\lambda\to\infty}\frac{dz_\lambda^\theta}{d\lambda} \neq 0$, then there must exist a coordinate $j$ with nonzero limit. Assume that the limit for the $j$-th coordinate of $\frac{dz_\lambda^\theta}{d\lambda}$ is $v_j \neq 0$. Then, there exist a $T > 0$ such that the $j$-th coordinate of $\frac{dz_\lambda^\theta}{d\lambda}$ is bounded away from zero by $|v_j|/2$ for all $\lambda > T$. However, due to the convergence $z_\lambda^\theta \to z_\infty$, we can find two numbers $\lambda_1, \lambda_2 > T$ such that $|z_{\lambda_1, j}^\theta - z_{\lambda_2, j}^\theta| < \frac{1}{2}|v_j|/2$. This contradicts with the lower bound $|v_j|/2$ for the $j$-th coordinate of the derivative by mean value theorem. Therefore, the limit of the derivative $\frac{dz_\lambda^\theta}{d\lambda}$ has to be zero which implies that

$$\lim_{\lambda\to\infty}\|m_\lambda^\theta(z_\lambda^\theta) - z_\lambda^\theta\| = 0.$$

$\square$

### I.1. Proof of Theorem 5.3

We utilize the change of variable $\lambda(t) = \log\frac{\alpha_t}{\beta_t}$ for $t \in (0, 1)$. We also let $t(\lambda)$ denote the inverse function of $\lambda(t)$.

Next, we consider $z_\lambda := \frac{x_{t(\lambda)}}{\alpha_{t(\lambda)}}$. Then, we have that $z_\lambda$ satisfies ODE Equation (16): $dz_\lambda/d\lambda = m_\lambda(z_\lambda) - z_\lambda$. Recall the transformation $A_t$ sending $x$ to $x/\alpha_t$. Then, we define $q_\lambda := (A_{t(\lambda)})_\# p_{t(\lambda)} = p * \mathcal{N}(0, e^{-2\lambda(t)}I)$.

By Theorem D.4, we have the following convergence rate for $m_\lambda(z_\lambda)$:

*Claim* 4. Fix $0 < \zeta < 1$. Then, there exists $\Lambda > -\infty$ such that for any radius $R > \frac{1}{2}\tau_\Omega$ and all $z \in B_R(0) \cap B_{\frac{1}{2}\tau_\Omega}(\Omega)$, one has

$$\|m_\lambda(z) - \mathrm{proj}_\Omega(z)\| \leq C_{\zeta,\tau,R} \cdot e^{-\zeta\lambda} \text{ for all } \lambda > \Lambda$$

where $C_{\zeta,\tau,R}$ is a constant depending only on $\zeta$ and $\tau$ and $R$.

*Proof of Claim 4.* We let $z_\Omega := \mathrm{proj}_\Omega(z)$. Note that $\|z_\Omega\| \leq \|z\| + d_\Omega(z) \leq 2R$. Since $p$ satisfies Assumption 5.2, we have that $p(B_r(z_\Omega)) \geq C_{2R}r^k$ for small $0 < r < c$. Now, we let $\Lambda := -\log(c)/\zeta$. By Theorem D.4, we conclude the proof. $\square$

The following Claim establishes an absorbing property for points in $z_{\lambda_\delta} \in B_{R_\delta}(0) \cap B_\delta(\Omega)$.

*Claim* 5. Consider $\delta > 0$ small such that $\delta < \frac{\tau_\Omega}{4}$. Fix any $R_\delta > 0$ such that $R_\delta > 2\delta$. Then, there exists $\lambda_\delta \geq \Lambda$ satisfying the following property: the trajectory $(z_\lambda)_{\lambda \in [\lambda_\delta, \infty)}$ starting at any initial point $z_{\lambda_\delta} \in B_{R_\delta}(0) \cap B_\delta(\Omega)$ of the ODE in Equation (16) satisfies that for any $\lambda \geq \lambda_\delta$: $z_\lambda \in B_{2R_\delta}(0) \cap B_{2\delta}(\Omega)$.

*Proof of Claim 5.* This follows from Claim 4 and Theorem C.11 (by letting $\sigma_\Omega := e^{-\Lambda}$). $\square$

Next we establish a concentration result for $q_\lambda$ when $\lambda$ is large.

*Claim* 6. For any small $\delta > 0$, there are $R_\delta, \lambda_\delta > 0$ large enough such that for any $\lambda \geq \lambda_\delta$ and $R > R_\delta$, we have that

$$q_\lambda \left( B_R(0) \cap B_\delta(\Omega) \right) > 1 - \delta.$$

*Proof of Claim 6.* Consider the random variable $\boldsymbol{X} = \boldsymbol{Y} + e^{-\lambda}\boldsymbol{Z}$ where $\boldsymbol{Y} \sim p$ and $\boldsymbol{Z} \sim p_0 = \mathcal{N}(0, I)$ are independent but from the same probability space $(\Omega, \mathbb{P})$. Then, $\boldsymbol{X}$ has $q_\lambda$ as its law. We have that $d_\Omega(x) \leq \|\boldsymbol{Y} + e^{-\lambda}\boldsymbol{Z} - \boldsymbol{Y}\| = e^{-\lambda}\|\boldsymbol{Z}\|$. For any $R > \delta$, we have that

$$\mathbb{P}(\|\boldsymbol{X} + e^{-\lambda}\boldsymbol{Z}\| \leq 2R, e^{-\lambda}\|\boldsymbol{Z}\| \leq \delta) \geq \mathbb{P}(\|\boldsymbol{X}\| \leq R, e^{-\lambda}\|\boldsymbol{Z}\| \leq \delta)$$
$$= \mathbb{P}(\|\boldsymbol{X}\| \leq R)\mathbb{P}(\|\boldsymbol{Z}\| \leq e^\lambda \delta).$$

Since $\boldsymbol{Z}$ follows the standard Gaussian, for any $\delta$, there exists $\lambda_\delta > 0$ such that for all $\lambda \geq \lambda_\delta$, we have that

$$\mathbb{P}(\|\boldsymbol{Z}\| \leq e^\lambda \delta) \geq \mathbb{P}(\|\boldsymbol{Z}\| \leq e^{\lambda_\delta}\delta) > 1 - \frac{\delta}{2}.$$

Now, since $p$ has a finite 2-moment and hence a finite 1-moment, there exists $R_\delta > 0$ such that $\mathbb{P}(\|\boldsymbol{X}\| \leq \frac{R_\delta}{2}) > 1 - \frac{\delta}{2}$. Therefore, for all $\lambda \geq \lambda_\delta$, we have that

$$q_\lambda \left( B_{R_\delta}(0) \cap B_\delta(\Omega) \right) \geq \mathbb{P}(\|\boldsymbol{X} + e^{-\lambda}\boldsymbol{Z}\| \leq R_\delta, e^{-\lambda}\|\boldsymbol{Z}\| \leq \delta)$$
$$\geq \left(1 - \frac{\delta}{2}\right)\left(1 - \frac{\delta}{2}\right) > 1 - \delta.$$

$\square$

Now, we establish the desired convergence results for the scale of $\lambda$.

*Claim* 7. For any small $\delta > 0$, there exist large enough $\lambda_\delta$, such that with probability at least $1 - \delta$, we have that $z_{\lambda_\delta} \sim q_{\lambda_\delta}$ satisfies the following properties:

1. $z_\lambda$ converges along the ODE trajectory starting from $z_{\lambda_\delta}$ as $\lambda \to \infty$;

2. the convergence rate is given by $\|z_\lambda - z_{\lambda_\delta}\| = O(e^{-\frac{\zeta\lambda}{2}})$.

*Proof of Claim 7.* For any $\delta > 0$, by Claim 5 and Claim 6, there exist large enough $\lambda_\delta$ and $R_\delta$, such that

- with probability at least $1 - \delta$, we have that $z_{\lambda_\delta} \sim q_{\lambda_\delta}$ lies in $B_{R_\delta}(0) \cap B_\delta(\Omega)$;

- the ODE trajectory $(z_\lambda)_{\lambda \in [\lambda_\delta, \infty)}$ starting at $z_{\lambda_\delta} \in B_{R_\delta}(0) \cap B_\delta(\Omega)$ satisfies that for all $\lambda \geq \lambda_\delta$, $z_\lambda$ lies in $B_{2R_\delta}(0)$ and $d_\Omega(z_\lambda) \leq 2\delta$.

This implies that one can apply the convergence rate of denoiser in Claim 4 to the entire trajectory with $C := C_{\zeta, \tau, R}$ where $R := 2R_\delta$. Then, for any $\lambda_1 < \lambda_2$ in the interval $[\lambda_\delta, \infty)$, we have that

$$\|z_{\lambda_2} - z_{\lambda_1}\| \leq \int_{\lambda_1}^{\lambda_2} \|m_\lambda(z_\lambda) - z_\lambda\| d\lambda$$
$$\leq \int_{\lambda_1}^{\lambda_2} \|m_\lambda(z_\lambda) - \mathrm{proj}_\Omega(z_\lambda)\| + \|\mathrm{proj}_\Omega(z_\lambda) - z_\lambda\| d\lambda \tag{44}$$
$$\leq \frac{C}{\zeta}(-e^{-\zeta\lambda_2} + e^{-\zeta\lambda_1}) + \delta e^{\lambda_\delta}(e^{-\lambda_1} - e^{-\lambda_2}) + \sqrt{\frac{4\delta C}{2 - \zeta}} \cdot \frac{2}{\zeta}\left(e^{-\zeta\lambda_1/2} - e^{-\zeta\lambda_2/2}\right),$$

where the bound is obtained in a way similar to how we obtain Equation (21).

This implies that the solution $z_\lambda$ becomes a Cauchy sequence and hence converges to a limit $z_\infty$.

Now, we let $\lambda_2$ approach $\infty$ in the above inequality and and replace $\lambda_1$ with $\lambda$ to obtain the following convergence rate:

$$\|z_\infty - z_\lambda\| = O(e^{-\frac{\zeta\lambda}{2}}). \tag{45}$$

$\square$

Now, we change the coordinate back to $t \in [0, 1)$ to obtain the following result. Since $\delta > 0$ is arbitrary so one can let $\delta$ approach 0 in the result below to conclude the proof of Theorem 5.3.

*Claim 8.* For any small $\delta > 0$, one can sample $x_0 \sim p_0$ with probability at least $1 - \delta$, such that the flow map $\Psi_t(x_0)$ converges to a limit $\Psi_1(x_0)$ as $t \to 1$.

*Proof of Claim 8.* Let $\lambda_\delta$ be the one given in Claim 7. Then, we let $t_\delta := t(\lambda_\delta)$. Consider the map $A_{t_\delta} : \mathbb{R}^d \to \mathbb{R}^d$ sending $x$ to $x/\alpha_{t_\delta}$. Then, we have that

$$(A_{t_\delta})_\# p_{t_\delta} = q_{\lambda_\delta}. \tag{46}$$

Both maps $A_{t_\delta}$ and $\Psi_{t_\delta}$ (whose existence follows from Theorem 4.1) are continuous bijections. It is then easy to see that if an ODE trajectory of Equation (16) converges starting with some $z_{\lambda_\delta} \sim q_{\lambda_\delta}$ and has convergence rate $O(e^{-\frac{\zeta\lambda}{2}})$, then the corresponding trajectory of the ODE Equation (2) starting with $x_0 := (A_{t_\delta} \circ \Psi_{t_\delta})^{-1}(z_{\lambda_\delta})$ also converges to a limit as $t \to 1$ with the same convergence rate up to the change of variable $\lambda \to t$. Finally, by Equation (46) we also have that

$$p_0(x_0 \text{ with the desired properties}) = q_{\lambda_\delta}(z_{\lambda_\delta} \text{ with the desired properties}) > 1 - \delta,$$

which concludes the proof. $\square$

Item 2 in the theorem is a direct consequence of item 1. As $\Psi_1$ is the pointwise limit of continuous maps $\Psi_t$ (cf. Theorem 4.1), it is a Borel measurable map. We know that $p_t$ weakly converges to $p_1 = p$ as $t \to 1$. Now, we just verify that $(\Psi_1)_\# p_0$ is also a weak limit of $p_t = (\Psi_t)_\# p_0$ to show that $(\Psi_1)_\# p_0 = p_1$.

For any continuous and bounded function $f$, we have that

$$\lim_{t \to 1} \int f(x)(\Psi_t)_\# p_0(dx) = \lim_{t \to 1} \int f(\Psi_t(x)) p_0(dx)$$
$$= \int f(\Psi_1(x)) p_0(dx)$$

where we used the bounded convergence theorem in the last step. Therefore, we have that $(\Psi_1)_\# p_0 = p_1$ and hence $\Psi_1$ is the flow map associated with the ODE $dx_t/dt = u_t(x_t)$.

## I.2. Proof of Theorem 5.4

**Part 1.** We first establish the following volume growth condition for the manifold $M$ satisfying assumptions in the theorem.

**Lemma I.1.** *There exists $0 < r'_M < \mathrm{Inj}(M)$ sufficiently small, where $\mathrm{Inj}(M)$ denotes the injectivity radius, such that the following holds. For any $R > 0$, there exists a constant $C_R > 0$ so that for any radius $0 < r < r'_M$, and for any $x \in M \cap B_R(0)$, one has $p(B_r(x)) \geq C_R r^m$.*

*Proof.* Notice that within the compact region $M_{R+\mathrm{Inj}(M)} := \overline{B_{R+\mathrm{Inj}(M)}(0)} \cap M$, the density $\rho$ is lower bounded by a positive constant $\rho_R > 0$. Additionally, the sectional curvature of $M$ is upper bounded by some constant $\kappa > 0$ due to the boundedness of the second fundamental form. Then by Gunther's volume comparison theorem (Gallot et al., 1990, 3.101 Theorem) for $0 < r < \mathrm{Inj}(M)$, there exists constant $c_\kappa > 0$ such that

$$\mathrm{vol}_M \left( B_r^M(x) \right) \geq \mathrm{vol}_{M_\kappa^m} \left( B_r^{M_\kappa^m}(x) \right),$$

where $B_r^M(x) := \{y \in M : d_M(x, y) < r\}$, $d_M$ denotes the geodesic distance, and $M_\kappa^m$ denotes the $m$-dimensional sphere with sectional curvature $\kappa$.

We then choose $r'_M$ to be small enough such that

$$\text{vol}_{M^m_\kappa}\left(B^{M^m_\kappa}_r(x)\right) \geq c_\kappa r^m$$

for some constant $c_\kappa > 0$ and all $r < r'_M$.

Since $\|x - y\| \leq d_M(x, y)$ for any $x, y \in M$, we have that $B^M_r(x) \subset B_r(x)$. Hence, we have that for all $x \in B_R(0) \cap M$ and $r < r'_M$,

$$p(B_r(x)) \geq p(B^M_r(x)) = \int_{B^M_r(x)} \rho(y)\text{vol}_M(dy) \geq \rho_R \text{vol}_M(B^M_r(x)) \geq \rho_R c_\kappa \cdot r^m.$$

By letting $C_R := \rho_R c_\kappa$, we conclude the proof. $\qquad\square$

Then, we can replicate the proof of Theorem 5.3 with only small changes of Claim 4 as follows using $\zeta = 1$:

*Claim* 9. Under assumptions as in Theorem 5.4, there exsits $\Lambda > -\infty$ such that for any radius $R > \tau_M/2$, all $z \in B_R(0) \cap B_{\tau_M/2}(M)$, we have that

$$\|m_\lambda(z) - \text{proj}_M(z)\| \leq C_{\tau,R} \cdot e^{-\lambda}, \text{ for } \lambda > \Lambda$$

where $C_{M,R}$ is a constant depending only on $R$ and the geometry of $M$.

*Proof of Claim 9.* This can be proved by carefully examining all bounds involved in the proof of Theorem D.5.

- Equation (29): Since $z \in B_R(0) \cap B_{\tau/2}(M)$, we have that $z_M \in B_{R+\tau_M/2}(0)$ and hence $\|z_M\|$ in the numerator can be bounded by $R + \tau_M/2$. The denominator is lower bounded by some polynomial of $r_0$ by Equation (25) and the volume growth condition of $p$ established in Lemma I.1. We also notice that $r_0$ can be chosen uniformly for all $z \in B_R(0) \cap B_{\tau_M/2}(M)$ as it is completely dependent on the reach. Therefore, the big O function is bounded above by some function of $R$.

- Equation (32): $C_1$ can be bounded by $R$ and the bounds on the local Lipschitz constant of the density and the second fundamental form (which bounds the Ricci tensor).

- Equation (33): this one is bounded similarly as the one in Equation (29) by some function of $R$.

- Lemma D.10: the bound $C_2$ is bounded by the bounds on the second fundamental form and its covariant derivatives.

In conclusion,

$$\|m_\lambda(z) - \text{proj}_M(z)\| \leq d_{\text{W},2}(p(\cdot|\boldsymbol{X}_\lambda = z), \delta_{z_M}) = \sqrt{m}e^{-\lambda} + G(z)$$

where $|G(z)| \leq C'_{M,R}e^{-2\lambda}$ for all $z \in B_R(0) \cap B_{\tau_M/2}(M)$ and $C'_{M,R}$ depends only on $R$ and geometry bounds of $M$ such as reach and second fundamental form bounds. Then, by combining the two exponential terms, one concludes the proof. $\quad\square$

**Part 2.** In the discrete case, we let $\Omega := \{x_1, \ldots, x_N\}$. We can improve the convergence rate in Theorem 5.3 to be exponential by considering a direct consequence of Corollary D.8:

*Claim* 10. For all $z \in \mathbb{R}^d$ such that $d_\Omega(z) < \frac{1}{4}\text{sep}_\Omega$, we have that

$$\|m_\lambda(z) - \text{proj}_\Omega(z)\| \leq C_2 \cdot \exp(-C_1 e^{2\lambda}),$$

where $C_1, C_2$ only depends on $\text{sep}_\Omega := \min_{x_i \neq x_j \in \Omega}\|x_i - x_j\|$ is the minimal separation between the points in $\Omega$.

Therefore, when $\lambda_\delta$ is large enough, we have that for any $\lambda_2 > \lambda_1 \geq \lambda_\delta$

$$\begin{aligned}
\frac{d(e^{2\lambda}d^2_\Omega(z_\lambda))}{d\lambda} &\leq 2e^{2\lambda}d_\Omega(z_\lambda)\|m_\lambda(z_\lambda) - \text{proj}_\Omega(z_\lambda)\| \\
&\leq 2C_2 e^{2\lambda - C_1 e^{2\lambda}}d_\Omega(z_\lambda) \\
&\leq C_3 e^{-\lambda}d_\Omega(z_\lambda),
\end{aligned}$$

where $C_3$ is a constant dependent on $C_1, C_2$ and the properties of the exponential function.

Then, in particular, there is

$$e^{2\lambda}d_\Omega^2(z_\lambda) \leq e^{2\lambda_\delta}d_\Omega^2(z_{\lambda_\delta}) + C_3 \int_{\lambda_\delta}^\lambda e^{-\lambda'}d_\Omega(z_{\lambda'})d\lambda',$$

$$\leq e^{2\lambda_\delta}C_5 + C_4 \int_{\lambda_\delta}^\lambda e^{-\lambda'}d\lambda'$$

where the we use the fact that $d_\Omega(z_\lambda)$ is bounded by the absorbing result in Theorem C.11 and hence the integral is bounded by a constant $C_4$.

This implies that $d_\Omega^2(z_\lambda) \leq e^{-2(\lambda-\lambda_\delta)}C_5 + C_4 e^{-2\lambda} \leq C_6 e^{-\lambda}$ for some constant $C_6$ depending on $C_4, C_5$. Now, following the rest of the estimations in the proof of Theorem 5.3, we can replace the rate $O(e^{-\frac{\zeta\lambda}{2}})$ in Claim 7 with $O(e^{-\lambda})$ after the change above and conclude the proof.

## J. Experiments

### J.1. A Synthetic Dataset with Three Clusters

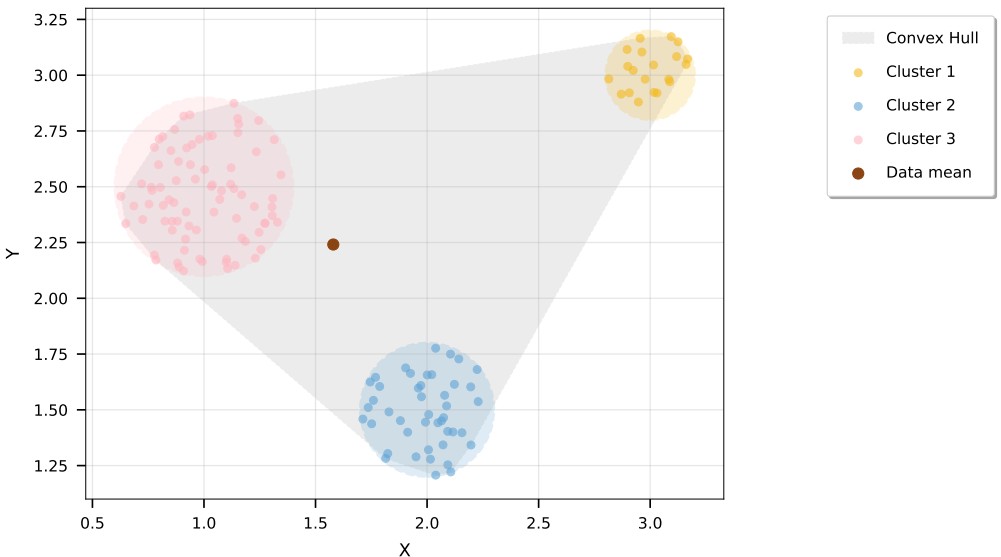

*Figure 6.* **The synthetic dataset showing three clusters with different scales and sample sizes.** The brown point marks the data mean.

We generate a synthetic dataset in $\mathbb{R}^2$ consisting of three distinct clusters shown in Figure 6:

- **Cluster 1 (pink):** 80 points drawn from the uniform distribution in the disk centered at $(1.0, 2.5)$ with radius $0.4$.

- **Cluster 2 (blue):** 44 points drawn from the uniform distribution in the disk centered at $(2.0, 1.5)$ with radius $0.3$.

- **Cluster 3 (yellow):** 20 points drawn from the uniform distribution in the disk centered at $(3.0, 3.0)$ with radius $0.2$

The total dataset contains 144 points with a diameter of 2.623. As shown in Figure 6, the clusters exhibit different scales and sample sizes. This is the same dataset used in Figure 2. We will continue to use this dataset to illustrate the various stages of a trajectory in the FM ODE with respect to the parameter $\sigma$ (cf. Equation (11)). We use the closed form optimal denoiser Equation (9) with $\alpha_t = t$ and $\beta_t = 1 - t$ (the Recitified flow scheduling) for the sampling process. This will generate a trajectory in $t$ parameter, and we transform it into $\sigma$ parameter to align with the general results in Sections 4.2, 4.3 and 5.2. To this end, we use the transformation in Proposition 2.1 with a starting $\sigma$ value $\sigma_1 = 100$.

**Single trajectory visualization.** We first examine a single trajectory in detail shown in Figure 7. The trajectory exhibits clear "turning points" suggesting transitions between stages - first approaching the data mean (brown point), then being attracted to a local cluster, and finally converging to a specific data point. We examine the quantitative results we developed in Sections 4.2, 4.3 and 5.2 on these stages.

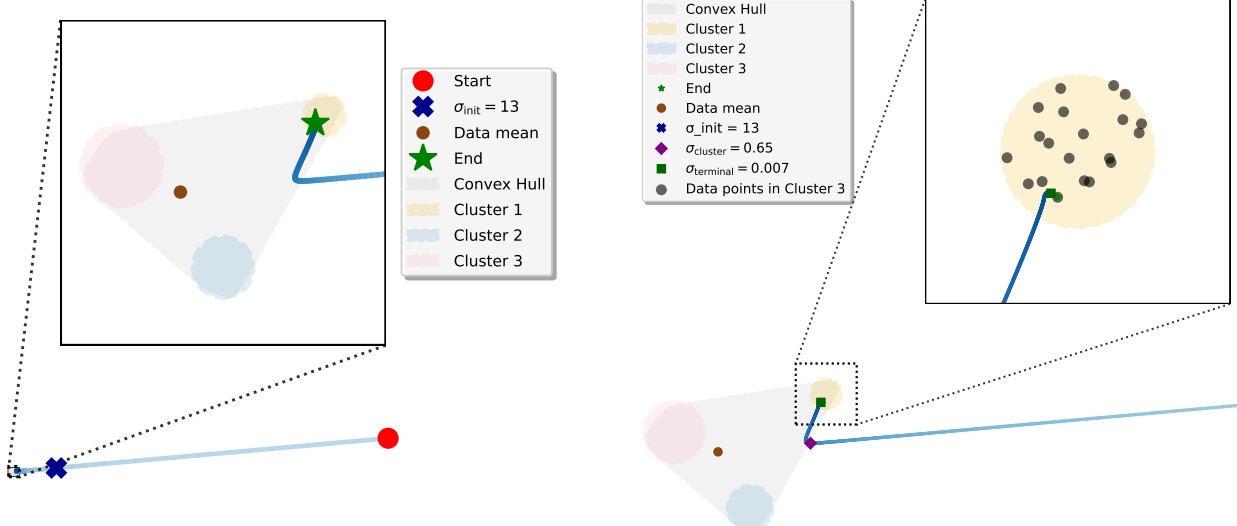

(a) Initial stage: trajectory moves toward data mean (brown point). Blue cross marks $x_{\sigma_{\text{init}}}$. Top: zoomed view near clusters. Bottom: full view showing coarse movement.

(b) Later stages: attraction to yellow cluster (purple rhombus marks $x_{\sigma_{\text{cluster}}}$) and convergence to training point (green square marks $x_{\sigma_{\text{terminal}}}$).

*Figure 7.* **Evolution of a single trajectory showing three distinct stages.**

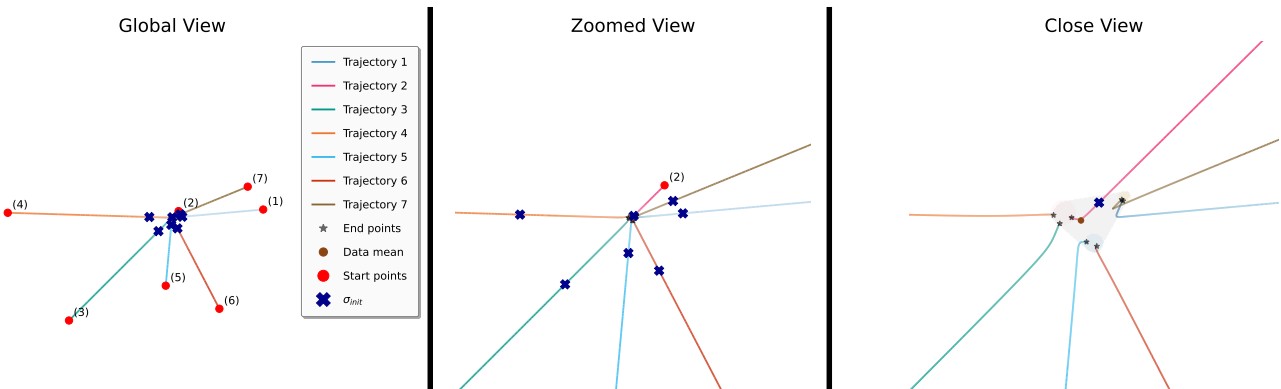

*Figure 8.* **Multiple trajectories from different initial samples.** Left: Global view showing roughly linear paths to mean. Middle: Zoomed view. Right: Close view near data support. Blue crosses mark $x_{\sigma_{\text{init}}}$ for each path.

**Initial stage:** Starting with $\sigma_1 = 100$ and initial distance to mean 122, Proposition 4.2 predicts $\sigma_{\text{init}} = 13$ with $\zeta = 0.5$. This suggests that the trajectory will approach the mean *at least* before $\sigma_{\text{init}} = 13$. We mark the location $x_{\sigma_{\text{init}}}$ (blue cross) in Figure 7a and indeed observe that the trajectory moves toward the mean prior to $\sigma_{\text{init}} = 13$.

We additionally examine multiple trajectories in Figure 8. Most initial trajectories form nearly straight lines toward the mean, with $x_{\sigma_{\text{init}}}$ (blue crosses) consistently making good predictions when the distance to the mean is monotonically decreasing. Interestingly, Trajectory 2, which starts closer to the data distribution, overshoots the mean. From the top right, it initially moves toward the mean, then continues past it, and is eventually absorbed into the pink cluster.

In Figure 9, we show the detailed view of Trajectory 1 in our running example (Figure 2) as well as Trajectory 7, which also

converges to the yellow cluster. We will use them to illustrate the intermediate and terminal stages.

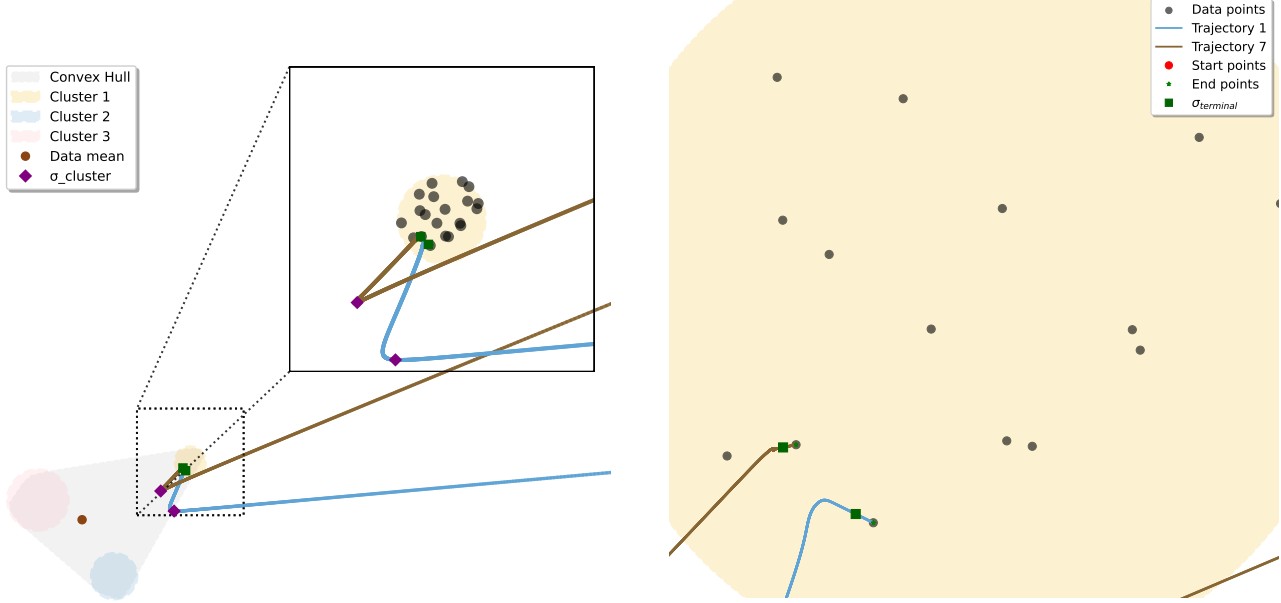

(a) Intermediate stage: Two trajectories attracted to yellow cluster. Purple rhombuses mark $x_{\sigma_{\text{cluster}}}$.

(b) Terminal stage: Final convergence to training points. Green squares mark $x_{\sigma_{\text{terminal}}}$.

*Figure 9.* **Detailed view of intermediate and terminal stages for two example trajectories.**

**Intermediate stage:** Using Proposition 4.4 with the yellow cluster's diameter of 0.364 and $\epsilon = 0.1$, we compute $\sigma_{\text{cluster}}$ ($\sigma_0(S, \epsilon)$ for yellow cluster) is 0.65. The locations $x_{\sigma_{\text{cluster}}}$ for both trajectories (purple rhombuses) align well with the points where both trajectories exhibit a clear attraction to this local cluster.

**Terminal stage:** We then examine the terminal stage in Figure 9b. First, note that the two trajectories indeed converge to two training data points, validating the terminal time convergence. We use Proposition 5.10 to compute $\sigma_0(V_i^\epsilon)$ for data points $x_i$ which we denote in this section as $\sigma_{\text{terminal}}$ ((green square) for two trajectories and see the Proposition predicts trajectory after $\sigma_{\text{terminal}}$ will converge to nearest data point. For Trajectory 1, the separation of the converged point is 0.06, and we choose $\epsilon$ as one-third of the separation. The predicted $\sigma_{\text{terminal}}$ for Trajectory 1 is 0.007 and marked on the trajectory. For Trajectory 7, the separation is 0.04, and we choose $\epsilon$ as one-third of the separation. The predicted $\sigma_{\text{terminal}}$ for Trajectory 7 is 0.004 and marked on the trajectory. In both cases, the predicted $\sigma_{\text{terminal}}$ lies in the segment where the trajectory is attracted to a specific training point, validating the prediction.

**Memorization with asymptotically optimal denoiser:** We also examine the memorization phenomenon with the asymptotically optimal denoiser $m_\sigma$ described in Proposition 5.10. We start with the same initial points as those in Figure 8 and evolve the trajectories using the asymptotically optimal denoiser given by:

$$\widetilde{m}_\sigma(x) = m_\sigma(x) + \sigma * \epsilon$$

where $\epsilon$ is a random perturbation sampled from $\mathcal{N}(0, 3^2 I)$. The trajectories are shown in Figure 10 with the perturbation vector shown in the left panel of the subfigure on the left. Comparing with Figure 8, we observe that the perturbation significantly drifts the trajectories; however, as the perturbation decays with $\sigma$, the trajectories still converge to a specific training point. This validates the memorization phenomenon with the asymptotically optimal denoiser in Proposition 5.10.

### J.2. The CIFAR-10 Dataset

The CIFAR-10 dataset (Krizhevsky, 2009) contains $50,000$ training images across 10 classes and is a popular benchmark for evaluating generative models. In this subsection, we investigate the mean attraction property of flow models and the memorization issue highlighted in our theoretical analysis utilizing the CIFAR-10 dataset. These data consist of $32 \times 32$ RGB images, each with 3 channels. Following the practice in Karras et al. (2022), we normalize the pixel values to $[-1, 1]$

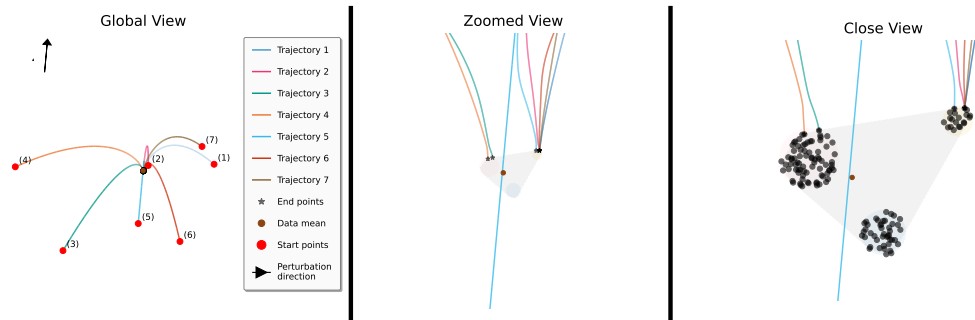

*Figure 10.* **Memorization behavior with asymptotically optimal denoiser.** Left: Global view showing trajectories under perturbation, with perturbation vector displayed. Middle: Zoomed view of trajectories. Right: Close view near data support. While perturbations initially steer trajectories (most notably in trajectory 5, which traverses across the data region), all trajectories eventually converge to data points as the perturbation strength diminishes with $\sigma$, demonstrating the robustness predicted by our theoretical results.

by the transformation $x \mapsto x/127.5 - 1$. This preprocessing results in a dataset with diameter $106.8$ and averaged norm $27.2$.

To establish a reference, we define the *empirical optimal denoiser* $m_\sigma$ based on the empirical distribution $p$ over the training images of CIFAR-10. The corresponding FM ODE trajectory is denoted by $(x_\sigma)_{\sigma \in [\sigma_2, \sigma_1]}$, as governed by Equation (11). We refer to this trajectory as the *empirical optimum*. Additionally, we consider a pre-trained denoiser $m_\sigma^{\mathrm{EDM}}$ from the EDM model (Karras et al., 2022), with its corresponding FM ODE trajectory denoted as $(x_\sigma^{\mathrm{EDM}})_{\sigma \in [\sigma_2, \sigma_1]}$ and governed by Equation (15).

To generate ODE samples, we initialize from random Gaussian noise and evolve the trajectories using the 18-step polynomial noise schedule (discretization) from EDM:

$$\sigma_n = \left(\sigma_{\mathrm{max}}^{1/\rho} + \frac{n}{N}(\sigma_{\mathrm{min}}^{1/\rho} - \sigma_{\mathrm{max}}^{1/\rho})\right)^\rho, \quad n = 0, 1, \ldots, N,$$

with parameters $\sigma_{\mathrm{max}} = 80$, $\sigma_{\mathrm{min}} = 0.002$, $\rho = 7$, and $N = 18$.

### J.2.1. INITIAL MEAN ATTRACTION

To investigate the convergence-to-mean behavior in the initial stages of sampling, we analyze trajectories generated by both the empirical optimal denoiser $m_\sigma$ and the pre-trained EDM denoiser $m_\sigma^{\mathrm{EDM}}$. Let $\mathrm{mean}$ represent the mean of the CIFAR-10 dataset.

For clarity, we use $x_\sigma^*$ to denote either $x_\sigma$ or $x_\sigma^{\mathrm{EDM}}$, and $m_\sigma^*$ to denote either $m_\sigma$ or $m_\sigma^{\mathrm{EDM}}$. At each sampling step, we evaluate two key metrics:

1. The distance $\|m_\sigma^*(x_\sigma^*) - \mathrm{mean}\|$ between the denoiser output and the dataset mean.

2. The ratio $\|m_\sigma^*(x_\sigma^*) - \mathrm{mean}\|/\|m_\sigma^*(x_\sigma^*) - x_\sigma^*\|$, which quantifies the distance between the denoiser output and the data mean relative to the trajectory direction $m_\sigma^*(x_\sigma^*) - x_\sigma^*$. This ratio can be interpreted as a relative error when approximating the denoiser output with the dataset mean.

We compute these metrics for the empirical optimal denoiser and the pre-trained EDM denoiser over 10,000 random initial seeds, with results shown in Figure 11. The figures plot mean with shaded regions indicating $\pm 1$ standard deviation. When $\sigma$ is large, both the empirical optimal and trained denoisers are close to the dataset mean, as evidenced by the small deviation to the mean in Figure 11a and the small relative errors in trajectory direction in Figure 11b. Notably, the relative error is very small for the first step and approximately below $1\%$ for the first $4$ steps, with a small variance across seeds. This suggests that the trajectory is strongly and consistently attracted to the data mean in the initial steps when $\sigma$ is large, validating the theoretical prediction in Proposition 4.2.

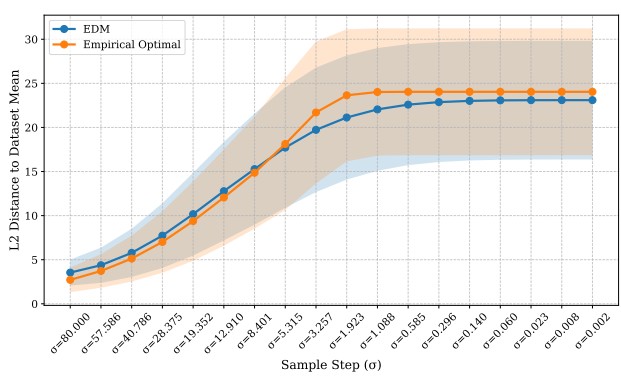
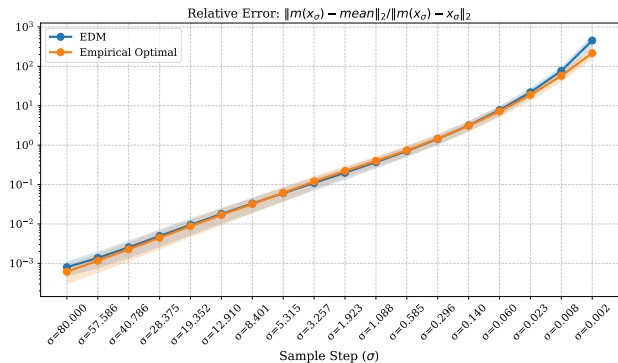

(a) Mean deviation from data mean over time, averaged across 10,000 random seeds. Shaded regions show $\pm 1$ standard deviation.

(b) Log-scale relative error between trajectory direction and denoiser-mean difference, averaged across 10,000 random seeds. Shaded regions show $\pm 1$ standard deviation.

*Figure 11.* **Initial mean attraction in CIFAR-10 trajectories.** Results averaged over 10,000 random initializations with shaded regions showing $\pm 1$ standard deviation. Both empirical optimal and trained denoisers exhibit strong initial attraction to the dataset mean when $\sigma$ is large, as evidenced by the small deviation to the mean (left) and small relative errors in trajectory direction (right, log scale), validating theoretical predictions.

### J.2.2. TERMINAL CONVERGENCE AND MEMORIZATION

First, we examine the behavior of the empirical optimal denoiser to validate Proposition 5.9 and serves as a reference for the perturbed case in Proposition 5.10. Starting from a Gaussian noise, Figure 12 illustrates the trajectory $x_\sigma$ (top) and the corresponding denoiser outputs $m_\sigma(x_\sigma)$ (bottom). As predicted by Proposition 5.9 or the general convergence result in Theorem 5.3, the unperturbed trajectory progressively refines the sample, with both the denoiser outputs showing clear image structure throughout the sampling process and ultimately converging to a training image.

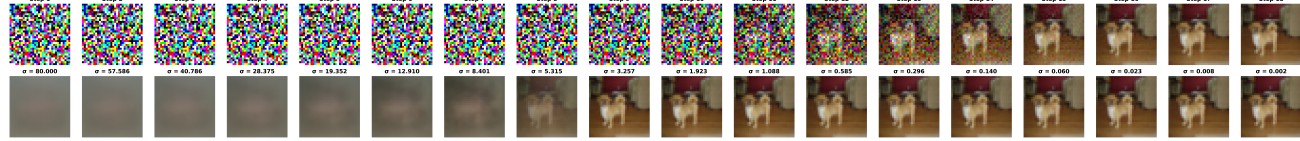

*Figure 12.* **Reference case: Sample generation with unperturbed empirically optimal denoiser.** Top: ODE trajectory $x_\sigma$. Bottom: Denoiser outputs $m_\sigma(x_\sigma)$. Note the clear image structure throughout and smooth convergence to a training image.

To validate the memorization result for an asymptotically optimal denoiser presented in Proposition 5.10, we then introduce a significant perturbation to the denoiser by adding noise that scales with $\sigma$. Specifically, we sample a fixed $\epsilon \sim \mathcal{N}(0, 10^2 I)$ and perturb the empirical optimal denoiser as follows:

$$\widetilde{m}_\sigma(x) := m_\sigma(x) + \sigma\epsilon, \ \forall x \in \mathbb{R}^d.$$

The scaling ensures the perturbed denoiser is asymptotically optimal. This perturbation dramatically affects the denoiser outputs, as shown in Figure 13. Compared to the clear outputs in the unperturbed case, the perturbed denoiser outputs (bottom row) are almost unrecognizable for all but the last four steps due to the large-scale noise. However, despite this substantial corruption of the denoiser outputs, the ODE trajectory (top row) still manages to get very close to a training image visually and through nearest neighbor search.

We also provide in Figure 14a and Figure 14b the quantitative results of the convergence and memorization with 10,000 random seeds. In the reference case with the empirical optimal denoiser, the distances to CIFAR-10 training set for both trajectories and denoiser outputs are shown in Figure 14a. The denoiser output initially is around 10 away from the CIFAR-10 dataset and then quickly converges to 0—validating the quick convergence predicted by Proposition 5.9. The distances from the trajectory to CIFAR-10 dataset smoothly converge to close to around 0.1—very small compared to the mean norm 27.2 of the dataset. In the perturbed case with the heavily corrupted denoiser shown in Figure 14b, the denoiser outputs are heavily corrupted by noise and far from the CIFAR-10 dataset until late stages. However, the trajectories still

manage to get close to training images with the distances converging to around $1.0$—still small compared to the mean norm of the dataset. Also, note that we are only using a coarse sampling schedule with $18$ steps, and the convergence can be further improved with a finer schedule.

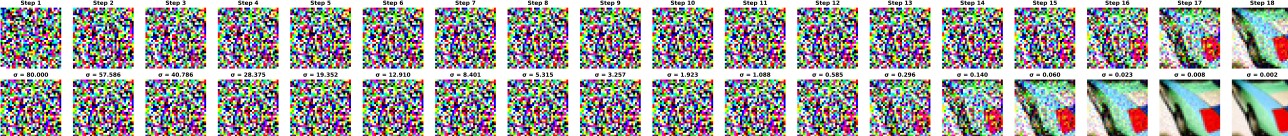

Figure 13. **Perturbed case: Sample generation with heavily perturbed but asymptotically optimal denoiser.** Despite the denoiser outputs being severely corrupted by noise (bottom) compared to the reference case, the ODE trajectory (top) still remarkably converges to a training image, validating the robustness predicted by Proposition 5.10.

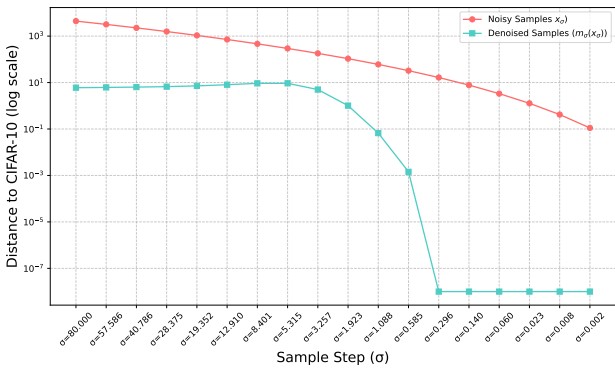
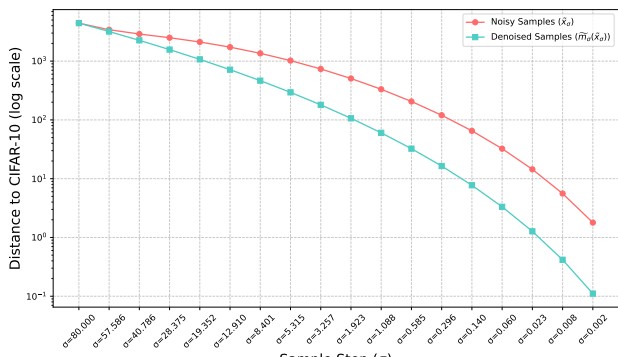

(a) Distance from trajectory points $x_\sigma$ and denoiser outputs $m_\sigma(x_\sigma)$ to nearest CIFAR-10 images for the unperturbed empirical optimal denoiser $m_\sigma$. While denoiser outputs stay close to the CIFAR-10 dataset, trajectories initially deviate but eventually converge.

(b) Distance metrics for perturbed denoiser $\widetilde{m}_\sigma$. Despite highly corrupted denoiser outputs (blue) that stay far from the CIFAR-10 dataset until the late stages, trajectories (orange) still manage to get very close to training images, validating the theoretical robustness prediction.

Figure 14. **Quantitative result of convergence and memorization.** Evolution of distances to the CIFAR-10 training set for both trajectories and denoiser outputs using the empirical optimal denoiser and its perturbed version, averaged over 10,000 random seeds. Left: Reference case with empirical optimal denoiser shows smooth convergence. Right: Despite severe perturbation corrupting intermediate denoiser outputs, trajectories still converge toward training data, demonstrating the robustness of memorization predicted by Proposition 5.10.

These experiments empirically validate our theoretical result in Proposition 5.10: even when the denoiser outputs are severely corrupted during intermediate steps, as long as the trained denoiser asymptotically approximates the empirical optimal denoiser, the FM ODE trajectory will still converge to the training data. This observation highlights the importance of carefully regularizing terminal time behavior during training to prevent memorization.

## J.3. Local Cluster Absorbing and Attracting Behavior

In this section, we provide additional experimental results to validate the local cluster absorbing and attracting behavior of the FM ODE. We use the FFHQ dataset (Karras et al., 2019) which contains high-resolution human face images. We randomly sample $10,000$ images from the FFHQ dataset and downsample them to $64 \times 64$ resolution. To visualize the distribution of facial images in the feature space, we perform t-SNE dimensionality reduction on the downsampled FFHQ dataset. As shown in Figure 15, we color code the points based on two related attributes: (1) the average RGB intensity (left) and (2) the illumination value (right). The average RGB intensity is computed as the mean of the pixel values across all three channels, while the illumination value is computed as the mean of the pixel values in the Y channel of the YCbCr color space. While the dataset does not form distinct, separated clusters as in our synthetic example in Appendix J.1, it still contains regions of varying density along the illumination spectrum. In particular, very dark faces and very bright faces concentrate at opposite ends of the feature space, creating two high-density regions. This natural organization provides an ideal setting to evaluate our theoretical results on absorption phenomena in a realistic dataset even when distinct clusters are

not present. We will demonstrate that the flow model trajectories are attracted to these high-density illumination regions, consistent with our local cluster attraction theory.

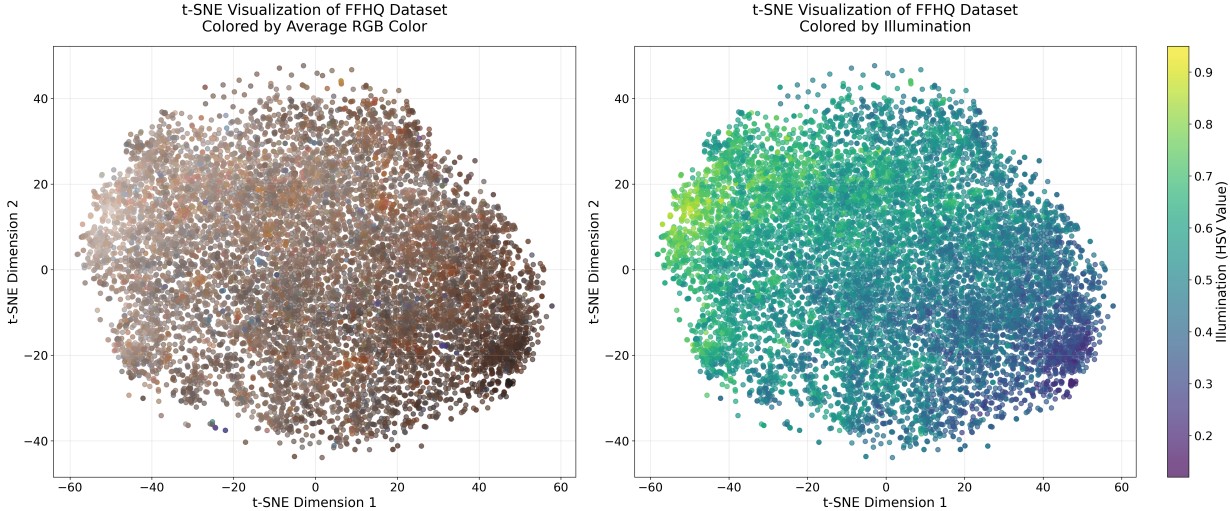

*Figure 15.* **t-SNE Visualization of FFHQ (Human Face) Dataset.** We downsample the original dataset from high-resolution (1024×1024) human face images to 64×64 resolution—aligning the training procedure in EDM and subsample 10,000 data points to perform t-SNE. The visualization shows feature embeddings colored by average RGB intensity (left) and illumination value (right). Although the data does not form distinct clusters, samples with similar illumination naturally organize into local neighborhoods in the feature space. The extremes of the illumination spectrum (very dark and very bright regions) exhibit higher local density.

In Figure 16, we show samples generated from a pretrained EDM model using three different initialization strategies: (1) random noise, (2) noise initialized near dark illumination regions, and (3) noise initialized near bright illumination regions. The results demonstrate that random initialization yields samples across the illumination spectrum, while dark and bright initializations consistently generate samples with corresponding illumination characteristics. Thus, even without explicit clusters, the flow gravitates toward locally dense regions defined by a continuous attribute—illumination which aligns with our cluster-absorption theory.

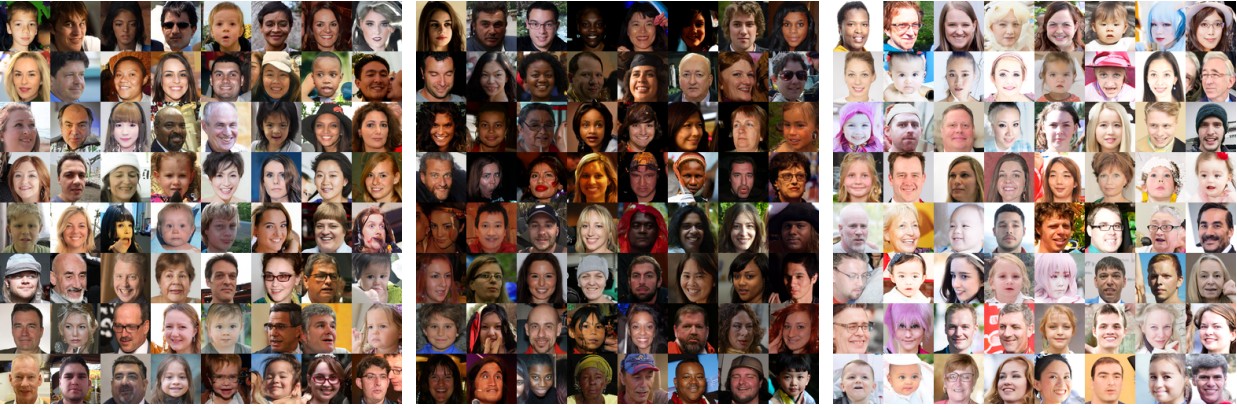

*Figure 16.* **Illumination-based Absorption Behavior.** Samples generated using three initialization strategies: random noise (left), noise near dark illumination regions (middle), and noise near bright illumination regions (right). Random initialization produces samples across the illumination spectrum, while dark and bright initializations generate samples with corresponding illumination characteristics, aligning with our cluster absorption result even for continuous attributes rather than discrete clusters.

