# OpenReview forum: "Elucidating Flow Matching ODE Dynamics via Data Geometry and Denoisers"
_ICML.cc/2025/Conference — ICML 2025 poster_

### Official Review · Reviewer_uK69 · 2025-03-06

**Overall Recommendation:** 4

**Summary:**

This paper gives a theoretical characterization of the convergence behavior of flow model ODE trajectories. The authors show that flow trajectories can be divided into three stages -- initial stage where particles are attracted towards dataset mean, intermediate stage where particles are attracted towards local clusters, and terminal stage where particles converge to data points.

**Claims And Evidence:**

All claims are supported with theoretical analysis. However, I am not familiar with the literature, and I did not check the proofs, so I could be mistaken.

**Essential References Not Discussed:**

None, to the best of my knowledge.

**Experimental Designs Or Analyses:**

Not applicable.

**Methods And Evaluation Criteria:**

Not applicable.

**Other Comments Or Suggestions:**

- At the bottom of page 2, there is a typo in the definition of medial axis

**Other Strengths And Weaknesses:**

- **Significance** : this paper provides convergence guarantees under weaker assumptions compared to previous work such as [1]. Furthermore, the paper provides some insights into the memorization behavior of flow and diffusion-based generative models. Specifically, the flow velocity could overfit to the empirical data distribution near terminal time-steps, leading to memorization.

[1] Gaussian Interpolation Flows, JMLR, 2025.

**Questions For Authors:**

None.

**Relation To Broader Scientific Literature:**

This paper gives a convergence guarantee as well as characterization of flow ODE trajectories under the assumption that the data distribution is a mixture of local clusters.

**Theoretical Claims:**

No, I did not check the proofs.

---

> ### Author Rebuttal · Authors · 2025-04-01
>
> We thank the reviewer for the positive feedback and constructive comments. Thank you for appreciating the convergence guarantees and insights into the memorization behavior of flow and diffusion-based generative models. We will fix the missing parenthesis in the definition of the medial axis to make it clearer in the revised manuscript.
>
> We would also like to point out that we have further strengthened our theoretical results with new experiments on FFHQ to show the utility of our theoretical findings (in addition to results we already have in the appendix); see response to Reviewer m3yf, as well as discussions to potential practical implications as in response to Reviewer zqMS. We will clarify these points in the revision. We believe that incorporating those changes will further strengthen our paper.

---

> > ### Comment · Reviewer_uK69 · 2025-04-08
> >
> > I have read through other Reviewer's comments and the authors' rebuttal, and I believe the additional clarifications further strengthen the paper. In particular, I find the experimental results on FFHQ interesting, and I believe the results can serve as a basis for analyses of Rectified Flows and Consistency Models. Hence, I have raised the score from **Weak Accept** to **Accept**.

---

> > > ### Author Response · Authors · 2025-04-08
> > >
> > > Thank you for your thoughtful feedback. We appreciate your support and are glad the additional clarifications and experiments helped strengthen the paper further.

---

### Official Review · Reviewer_m3yf · 2025-03-14

**Overall Recommendation:** 4

**Summary:**

This paper studies the convergence behavior of the ODE trajectories in flow-matching w.r.t training data using analytical tools from geometry. Specifically, it provides an extensive analysis of the ground-truth FM ODE trajectories under the affined Gaussian path, and shows how trajectories are shaped by data geometry in terms of “attracting” and “absorbing”. They show that trajectory evolution can be divided into ithe nitial/intermediate stage where trajectories move towards overall data support (mean) and then towards clusters of the data, and a final stage where convergence analysis is provided under much milder conditions than previous works. The work presents a lot of novel theoretical claims and findings, from well-posedness of FM ODE trajectory, per-sample level trajectory evolution patterns, convergence under a broad class of data distribution including submanifolds, equivariance, and finally provides some connection to memorization phenomena when target distributions are discrete measures.

**Claims And Evidence:**

The claims are mostly presented as propositions and theorems with substantial proofs and some empirical analysis of toy data. The findings on trajectory seem correct, although I did not have the chance to finish all the extensive proofs. Since the analysis is mostly based on a geometric point of view, it mostly talks about how a single sample is absorbed into the support of a cluster in the data and eventually converges to the target data point.

**Essential References Not Discussed:**

I am not aware of other essential references that are missing.

**Experimental Designs Or Analyses:**

There are no experimental results in the main text. There are some empirical validations of the attraction behavior on cifar10 dataset and a very simple toy data in the appendix, but these are all clustered dataset. It would be more interesting if a realistic dataset with less obvious classes/clusters such as CelebA are provided.

**Methods And Evaluation Criteria:**

The proposed method (or theoretical analysis framework) is through the lens of data geometry, which allows more general target distribution, and since it only cares about every single trajectory, I guess the framework makes sense. There is no benchmark or experimental comparison as the work is mostly theoretical and analytical.

**Other Comments Or Suggestions:**

No other comments.

**Other Strengths And Weaknesses:**

The paper is very math-heavy and packed with lots of theoretical findings. Some of them may not be very relevant to the major claims on convergence behavior (e.g., equivariance).
The paper studies the dynamic of ground-truth trajectories under gaussian path (or mostly diffusion path?), it mentioned that the findings can be used to support consistency model but did not include any results on that end. It is also unclear whether the conclusions can be generalized to non-random pairing (such as rectified flow after several rectifications), or non-Euclidean case.

**Questions For Authors:**

What is the meaning of $C_{\epsilon}^S <1$ in proposition 4.4? Does it hold for most of the clusters?  How should I interpret this proposition to make a useful design of the prior noise?

**Relation To Broader Scientific Literature:**

To my knowledge, the theoretical contribution is quite novel. As the authors claim, they are the first to individual trajectory-level behavior in flow matching, and the convergence is analysis under weaker assumptions than prior works (discussed in the related work section).

**Theoretical Claims:**

I only checked the proofs for propositions and theorems that appeared in the main text. The assumptions are clearly stated (e.g., bounded support for general attraction towards data support, the existence of FM ODE in $t\in[0,1)$,  well-separated local cluster assumption for cluster absorbance, finite 2-moment assumption for probability). I don’t think the well-separated local cluster assumption holds for any datasets, and since the clustering analysis is an important conclusion of the work, they should provide some explanations on what will happen if the data is not well-clustered.

---

> ### Author Rebuttal · Authors · 2025-04-01
>
> We appreciate the reviewer’s positive feedback and recognition of our novel theoretical contributions on the FM ODE convergence under weak assumptions, per-sample trajectory evolution patterns, and connections to memorization. We believe that the per-sample level analysis is important as it could inform data geometry-based sample step scheduling or steering strategies. Below, we respond to the comments/questions:
>
> **Cluster assumption and prior noise design:** We agree that the well-separated local cluster assumption may not hold for all datasets. Our analysis suggests similar attracting and absorbing behavior could happen for locally dense regions. For example, we can prove a particular case where data distributions obtained from well-separated clusters convolved with Gaussian noise. Specifically, Lemma C.11 identifies FM ODE trajectories with additionally convoluted Gaussian noise with early-stopped FM ODE trajectories w.r.t. the original distribution (up to parameterization), which can help extend current Proposition 4.4 to this more general setting.
>
> In practice, our analysis suggests that prior noise with a bias towards the cluster center could effectively control the sampling outcome to reflect the desired cluster characteristics. This could hold, in general, dense regions of the data distribution, as demonstrated in our new FFHQ experiments below.
>
> **Cluster experiments on FFHQ:** Following your suggestion, we conducted new experiments on the FFHQ dataset, which is similar to CelebA, with a pre-trained EDM model (Figures in PDF: https://anonymous.4open.science/r/ICML-7924/FFHQ_exp.pdf). Figure 1's t-SNE plot shows that images with similar illumination tend to cluster together, especially at brightness extremes, as RGB encoding makes brightness differences strongly influence distances in pixel space. This creates distinct dense regions for the darkest and lightest images.
>
> We performed sampling using a pre-trained EDM with three initialization strategies: random, near-light, and near-dark regions. All samples used standard EDM sampling (18 steps from noise 80 to 0.002), varying only initialization noise. The results in Figure 2 show that when initialized randomly, samples exhibit various illumination levels. When trajectories start near light regions, samples consistently display light characteristics, while dark initialization produces dark samples. These validate our theoretical findings on cluster absorption even where clustering is based on subtle, continuous features rather than discrete classes.
>
> **Non-random pairing and non-Euclidean case:** We suspect convergence still holds for non-random pairing, though general dynamics might differ significantly. For non-random pairings (as in rectified flow), the posterior distribution $p(x|x_t)$ is a Dirac delta, and the posterior mean is determined by the deterministic pairing. Hence, we suspect that the convergence in terminal time still holds in this case. But the ODE dynamics in other stages are different: for example, paths may straighten after rectification (due to convergence to optimal coupling), then there will be no travel to the data mean in the initial stage. The non-Euclidean case can be more challenging, as the transition kernels in those cases might be more complicated than Gaussian kernels. These topics are interesting and worth exploring in the future, and we will provide more discussions in the revision. We want to reiterate that our analysis of random pairing in Euclidean space is broadly applicable, as most generative models fit this setting.
>
> **Consistency model and equivariance** We agree that it would be beneficial to better connect our convergence result to consistency models and equivariance.
>
> The consistency model aims to distill the entire flow matching trajectory (from noise to data) into a single map, which necessarily requires the FM ODE to converge. Our work formally validates this convergence, ensuring consistency models are mathematically well-defined. We will add a discussion on this connection following our main convergence results.
>
> The equivariance result naturally extends from the convergence of FM ODE trajectories. Additionally, it provides another perspective on how data geometry affects the ODE trajectories: a similarity transformation results in some deterministic and explicit transformation of FM ODE trajectories.
>
> **$C_{\epsilon}^{S}<1$ in Proposition 4.4:** This condition ensures the formula is well-defined. It could fail when the weight $a_S$ on a cluster is large, but this is not problematic. In fact, when $C_{\epsilon}^{S}\geq 1$, the cluster exhibits a stronger attracting force and $\sigma_0(S, \epsilon)$ equals infinity, meaning that Proposition 4.4 would apply to any initialization time not just after time $\sigma_0(S, \epsilon)$. We will clarify this in the revision.
>
> We will incorporate these changes in the revision. Thank you for your feedback; we hope the revision addresses your concerns.

---

### Official Review · Reviewer_zqMS · 2025-03-14

**Overall Recommendation:** 3

**Summary:**

This paper presents a theoretical analysis of Flow Matching (FM) models addressing how data geometry influences the dynamics of the ODE trajectories used in FM-based generative models. They show that the denoiser guides the ODE dynamics through attracting and absorbing behaviors. They identify three stages of ODE evolution and establish the convergence of FM ODE trajectories under weak assumptions. The authors also provide insights into the memorization phenomenon and equivariance properties of FM ODEs.

**Claims And Evidence:**

The claims in the submission generally seem to be supported by the theoretical evidence. The authors provide mathematical proofs to support their claims about the convergence of FM ODE trajectories, the role of the denoiser in guiding these trajectories, and the influence of data geometry. However, the paper lacks empirical evidence to demonstrate the practical implications of the theoretical results. Adding experimental validation would strengthen the claims, particularly the stages on sampling efficiency and model performance.

**Essential References Not Discussed:**

The Jacobian of the denoiser was mentioned as being related to the covariance. The following works also mention this.

Ben-Hamu, Heli, et al. "D-Flow: Differentiating through Flows for Controlled Generation." Forty-first International Conference on Machine Learning.
Rissanen, Severi, Markus Heinonen, and Arno Solin. "Free Hunch: Denoiser Covariance Estimation for Diffusion Models Without Extra Costs." arXiv preprint arXiv:2410.11149 (2024).

It would be nice to have more discussion about how the Jacobian/covariance ties into the geometric insights of this paper.

**Experimental Designs Or Analyses:**

The paper lacks experimental validation to support its theoretical claims. While the analysis is thorough, the absence of experiments on toy problems or benchmark datasets limits the ability to assess the practicality.

**Methods And Evaluation Criteria:**

The paper seems theoretically sound and provides insights into the dynamics of FM ODE models, particularly in the role of the denoiser. However, the lack of practical evaluation, e.g. experiments on toy problems or benchmark datasets, is a significant gap. While the theoretical analysis is extensive, demonstrating these concepts on a simple synthetic or toy dataset would have helped validate the claims and made the results more tangible.

**Other Comments Or Suggestions:**

- Some assumptions, e.g. data lying on submanifolds with positive reach, could be explicitly discussed in terms of their limitations and applicability to real-world datasets.
- The paper would benefit from a small empirical demonstration, even on a toy dataset, to illustrate theoretical results and make them more tangible.

**Other Strengths And Weaknesses:**

Strengths:
- Provides theoretical insights, with convergence results for FM ODE dynamics.
- Analyses connecting data geometry with FM model trajectories.
- Extends prior theoretical analyses on flow matching and diffusion models.

Weaknesses:
- No experiments or demonstrations on toy problems or datasets.
- Sometimes difficult-to-follow notation and dense mathematical content reduce accessibility.
- Conditions (e.g., data lying on submanifolds with positive reach) might not apply broadly, potentially limiting practical relevance.
- Does not seem to link theoretical results to empirical findings in related literature.

**Questions For Authors:**

1. Could you clarify how restrictive the assumption of data being supported on submanifolds with positive reach is in practical scenarios?
2. How do you envision your theoretical results guiding practical improvements in the design or training of FM models? Could you provide examples?

**Relation To Broader Scientific Literature:**

I believe that the results are insightful in relation to the surrounding scientific literature, but I feel that the related work section could use more substance to contextualize the contributions.

**Theoretical Claims:**

The theoretical claims in the seem sound, to my knowledge, e.g. the existence and convergence of FM ODE trajectories (Theorems 4.1 and 5.3) and the attracting/absorbing dynamics (Theorems 3.1 and 3.2). But, the paper's assumptions (e.g., data on submanifolds with positive reach) and occasionally difficult-to-follow notation may limit the generality and accessibility of the results. Is the hashtag in the medial axis definition a typo? It would help to elaborate more on concepts like "medial axis" and "reach".

---

> ### Author Rebuttal · Authors · 2025-04-01
>
> We thank the reviewer for the constructive feedback and for recognizing the theoretical importance of our work on the convergence of FM ODE dynamics, the analyses connecting data geometry with FM model trajectories, and the memorization phenomenon. Below, we address the main comments/questions:
>
> **Experimental validation:** We conducted experiments on synthetic data and CIFAR-10, presented in the Supplement due to space constraints. Specifically, we demonstrated the emergence of the three stages of ODE evolution on synthetic data (Figures 7- 10)  and the convergence to data mean (Figure 11) and memorization phenomenon on CIFAR-10 (Figures 13, 14), aligning with our theoretical findings in sections 4 and 5. Following Reviewer m3yf's suggestion, we have now also conducted experiments on the human face dataset (FFHQ). Please see the response to Reviewer m3yf for more details.
>
> **Assumption on positive reach:** First, we note that this assumption is only used for our convergence results (Thm 5.3, 5.4) and equivariance (Prop. 5.7), not elsewhere in the paper. Also, the positive reach assumption is common in manifold learning literature (e.g., Fefferman et al., 2016) and is not overly restrictive. Specifically, the medial axis $\Sigma_{\Omega}$ of a manifold $\Omega$ is the set of points in the ambient space having more than one closest point on $\Omega$. The local feature size $lfs(x)$ at a point $x \in \Omega$ is the distance from $x$ to the medial axis $\Sigma_{\Omega}$. The reach of $\Omega$ is defined as $\min_{x\in \Omega} lfs(x)$. The local feature size of any submanifold embedded in Euclidean space is positive everywhere. Consequently, all compact submanifolds have positive reach, which includes most real-world datasets under the manifold hypothesis (e.g., image patches, audio).
>
> Moreover, our core analysis (e.g., Theorem 4.1) is local, meaning the positive reach assumption could potentially be relaxed to only require positive **local feature size**. This relaxation would extend our results to an even wider class of data distributions, including singular manifolds (e.g., sheets of manifolds glued together with singularities).
>
> Fefferman et al. "Testing the manifold hypothesis." JAMS  29.4 (2016): 983-1049.
>
> **Practical implications of our theory:** We believe that understanding the behavior of per-sample ODE trajectory is critical, as that is the trajectory followed during the inference stage. Our theoretical analysis has several direct practical implications for FM model design and training:
> 1. Memorization mitigation strategies: Our analysis in Section 5.2 reveals how the terminal stage of ODE evolution influences memorization phenomena, suggesting potential regularization approaches based on the denoiser near the terminal time–e.g. regularize the Jacobian of denoiser.
> 2. Latent space design principles: Our analysis reveals how data geometry directly shapes ODE trajectories, with important implications for latent space design. By leveraging the data's clustering structure, one can design more effective latent representations that facilitate both efficient sampling and precise feature formation. Our equivariance result suggests that preserving key equivariance properties can lead to more robust, generalizable, and interpretable models.
> 3. Theoretical foundation for one-step distillation method: Our convergence results provide rigorous mathematical justification for one-step distillation approaches like consistency models. These methods rely on the assumption that FM ODEs converge to stable, well-defined mappings, which our results formally validate.
> 4. Optimized sampling strategies: Our characterization of the three-stage ODE evolution suggests adaptive sampling approaches that allocate fewer integration points to the initial and final stages while concentrating computational resources on the intermediate feature formation stage. This provides theoretical grounding for empirically successful non-uniform time discretization schemes and explains why the practice of starting at moderately large noise levels (e.g., 80 in EDM) is successful, although infinite initial noise is required theoretically.
>
> **Other comments:** We will improve the paper's clarity in the revision to make the notation and concepts more accessible. For example, the hashtag in the medial axis definition is a shorthand notation for the cardinality of a set, and we will explain this explicitly in our future revision. Regarding the Jacobian of the denoiser, we will update its discussion and include the suggested references. For its relation to the data geometry, as the denoiser converges to projection onto the data manifold, the rank of the Jacobian of the denoiser can reflect the local dimensionality of the data, which could help indicate if the trained model is memorization- when the rank is very low.
>
> We will incorporate these changes in the revision. Thank you for your feedback and we hope the revision will address your concerns.

---

### Decision · Program_Chairs · 2025-05-01

**Decision:**

Accept (poster)

**Comment:**

The paper presents novel and theoretically backed up insights in the trajectories of Flow Matching models. The reviewers are in agreement that this is an important contribution and meets the criteria for publication.